# Systematic underestimation of type-specific ecosystem process variability in the Community Land Model v5 over Europe

5 **Christian Poppe Terán[1,2,3], Bibi S. Naz[1,3], Harry Vereecken[1,3], Roland Baatz[4], Rosie Fisher[5], Harrie-Jan Hendricks Franssen[1,3]**

*Correspondence to:* Christian Poppe Terán *(*c.poppe@fz-juelich.de)

[1]Institute of Bio and Geosciences – Agrosphere (IBG-3), Research Centre Jülich, Jülich, 52428, DE

10 [2]Faculty of Georesources and Materials Engineering, RWTH Aachen, Aachen, 52062, DE

[3]HPSC TerrSys, Geoverbund ABC/J, Jülich, 52428, DE

[4]Research Platform Data Analysis & Simulation, Leibniz Centre for Agricultural Landscape Research (ZALF), Müncheberg, 15374, DE

[5]CICERO Center for International Climate Research, Oslo, 0349, NO

**Abstract**

Evapotranspiration (ET) and gross primary production (GPP) are critical fluxes contributing to the energy, water, and carbon exchanges between the atmosphere and the land surface. Land surface models such as the Community Land Model v5 (CLM5) quantify these fluxes, estimate the state of carbon budgets and water resources, and contribute to a better understanding of climate change's impact on ecosystems. Past studies have shown the ability of CLM5 to model ET and GPP magnitudes well but emphasized systematic underestimations and lower variability than in the observations.

Here, we evaluated CLM5's predictions of water and energy fluxes using observations from eddy covariance stations from the Integrated Carbon Observation System (ICOS), remote sensing, and reanalysis data sets. We assess simulated ET and GPP from the grid scale ($CLM5_{grid}$) and the plant functional type (PFT) scale ($CLM5_{PFT}$). $CLM5_{PFT}$ exhibited a low systematic error in simulating the ET at the ICOS sites (average bias of -4.68%), indicating that PFT-specific ET closely matches the observations' magnitude. GPP was underestimated by $CLM5_{PFT}$, especially in deciduous forests (bias of -43.76%). The results showed an underestimation of spatiotemporal variability of simulated ET- and GPP distribution moments across PFTs for both CLM setups compared to reanalysis data and remote sensing products. These findings provide essential insights for improving land surface models, highlighting the need to enhance CLM5's ability to capture spatiotemporal variability in ET and GPP simulations across PFTs.

## 1. Introduction

Ecosystem processes, such as evapotranspiration (ET) and gross primary production (GPP), play an important role in cycling water, carbon, and energy between ecosystems and the atmosphere. Changes in the magnitude and variability of these fluxes can indicate ecosystems' inhibited performance due to changing environments (Kühn et al., 2021; Migliavacca et al., 2021). These changes can lead to short-term alterations and long-term trends in water resources and carbon pools in the atmosphere and the land surface. Thus, the accurate quantification of the variability of ecosystem processes is pivotal for developing climate change projections and formulating effective mitigation policies (Friedlingstein et al., 2023; Graf et al., 2023).

Notably, an accurate, functional understanding of land surface processes is essential to identify threatened ecosystems in the present and the future and facilitate carbon budget calculations. Land surface models (LSMs) serve as deterministic and process-based simulators of ecosystems, capturing energy, water, and carbon fluxes while considering their interactions and the heterogeneity of the land surface (Fisher and Koven, 2020). LSMs can complement point-scale observations from in-situ research infrastructures by providing spatiotemporally uniform and extensive high-resolution outputs. Their high-resolution process-based simulations contrast the often coarsely resolved remote sensing data. Hence, LSMs are frequently used tools for investigating and projecting the current understanding of ecosystem processes, such as GPP and ET, on various scales. However, there is uncertainty in the LSM structure, the parameters, the input data, and the initial conditions, which carry over to the simulated variables. Therefore, assessing how well the general simulated ET and GPP variability compares to the observations is crucial. Such evaluations deliver essential context on LSM biases and form a basis for analyses of more complex ecosystem responses. Recent studies already found discrepancies between LSM simulations of ET and GPP and observations collected in the field and from remote sensing. For instance, these discrepancies are evident in their magnitude and variability (De Pue et al., 2023; Boas et al., 2023; Cheng et al., 2021; Strebel et al., 2023) and their response to drought (Ukkola et al., 2016; Wu et al., 2020; Green et al., 2024). Therefore, assessing the accuracy of LSMs in representing observed GPP and ET fluxes is crucial to test and improve our current understanding of ecosystem process variability and identify the limitations of state-of-the-art LSMs.

Current land surface models, e.g., the Joint UK Land Environment Simulator (JULES), the Community Land Model 5 (CLM5), or the Community Atmosphere Biosphere Land Exchange Model (CABLE), employ a tiling system within the grid cell to account for functional differences of distinct patches on the land surface. The natural

and crop vegetation is grouped into plant functional types (PFT), the entities for which ecosystem process calculations are resolved (Fisher and Koven, 2020; Bonan et al., 2002; Solomon and Shugart, 1993). Typically, PFTs are defined based on morphological and phenological characteristics of the vegetation (e.g., leaf type and leaf longevity) and climate (Bonan et al., 2002). However, the usefulness of this PFT definition, or at least its current

coarsely resolved implementation, is a subject of debate (Caldararu et al., 2015; Van Bodegom et al., 2012). The primary argument against it is that observed plant traits implemented as PFT-related parameters vary to some extent in space and time in response to a changing environment. This spatiotemporal dependence of PFT traits is only marginally represented in LSMs. On top of that, most research assessing LSMs only used a handful of observation sites and did not analyze aggregated values for groups of sites observing the same PFT. Such analyses would

provide essential insights, as a recent study highlighted the differences between vegetation type concepts used in observation networks, e.g., the International Geosphere-Biosphere Programme (IGBP) classification, and PFTs used in LSMs and underlined the importance of improving these PFT concepts (Cranko Page et al., 2024).

The phenology of ecosystem processes, i.e., their seasonal cycles and evolution through the year and the growing season length, have shifted in timing due to climate change. A recent study investigated which factors drive the

changes in the mean annual dynamics of ecosystem processes in Europe (Rahmati et al., 2023), and many of these discovered feedbacks, for instance, the effect of increased atmospheric dryness on growing season length, are only implemented simplistically in LSMs. Furthermore, robust simulations of LSMs for impact assessments become even more critical as ecosystems experience more disturbances along with the changing climate. For example, projections show that droughts have recently become more frequent in Europe (Vautard et al., 2023; Rousi et al.,

2022) and that these extreme events will become even more frequent and severe in the future (Lehner et al., 2017). While the combined effect of a higher occurrence of compound drought events is currently not fully understood, it is clear from observations that individual drought years, or droughts in general, have already had a profound impact on ecosystem processes in Europe (Graf et al., 2020; Van Der Woude et al., 2023; Poppe Terán et al., 2023). Given that the frequency and severity of extreme events affect GPP and ET's statistical distributions, investigating how

the characteristics of the simulated distributions compare with the observed can contextualize findings of modeled ecosystem drought responses in Europe.

One predominantly used LSM is the Community Land Model version 5 (CLM5) (Lawrence et al., 2019, 2018). In the most recent version, CLM5 solves the biogeochemistry (BGC), i.e., the carbon and nitrogen cycles between the atmosphere, vegetation, and soil. CLM5 has been widely employed for quantifying and examining ecosystems at

various scales, including global (Xie et al., 2020; Sitch et al., 2015; Lawrence et al., 2019), regional (Cheng et al., 2021; Boas et al., 2023) and site-scale (Strebel et al., 2023; Umair et al., 2020; Song et al., 2020; Fisher et al., 2019a) applications. Several studies have highlighted the ability of CLM5 to simulate ecosystem processes close to the observations (Wozniak et al., 2020; Lawrence et al., 2019; Cheng et al., 2021; Zhang et al., 2023; Boas et al., 2023). However, they have also emphasized an underestimated magnitude and variability in the simulations
across different time scales and under various conditions.

The present study assesses CLM5's ability to capture ecosystem processes at a continental scale. To ensure comparability to point scale observations, we conducted high-resolution simulations at 0.0275° (approx. 3 km) resolution over the European Coordinated Regional Climate Downscaling Experiment (CORDEX) domain (Giorgi et al., 2009), resulting in 1544 x 1592 grid cells. Notably, the output contained variables from the subgrid-scale,
i.e., from within a 3 km grid cell, for PFTs present in the grid cell. We then compared the CLM5 grid level (CLM5$_{grid}$) and PFT level data (CLM5$_{PFT}$) to observations from a continental network of sites: The Integrated Carbon Observation System (ICOS) provides the WARM-WINTER-2020 data (Warm Winter 2020 Team and ICOS Ecosystem Thematic Centre, 2022), which includes Eddy Covariance measurements over a dense network of over 70 sites in Europe. It was named after and curated to support research on the effect of the warm winter of
2020 on terrestrial carbon fluxes. These ICOS data are regarded as the gold standard for calibrating and evaluating process-based models due to their ample spatial coverage as a network encompassing diverse land cover types. Thus, it offers an excellent opportunity to comprehensively assess simulated GPP and ET for specific PFT from our CLM5 setup over Europe.

Additionally, we include remote sensing data from the Global Land Surface Satellite (GLASS, (Liang et al., 2021))
and reanalyses from the European Center for Medium-range Weather Forecasts Reanalysis 5 - Land (ERA5L, Copernicus Climate Change Service (2019)) as well as from the Global Land Evaporation Amsterdam Model (GLEAM, Martens et al. (2017)) in our analyses to identify common patterns of ecosystem process variability between CLM5, in-situ observations, reanalysis, and remote sensing data.

In summary, this study uses ICOS observations as ground truth data. It compares them with grid and PFT level
CLM5 data and terrestrial surface fluxes from reanalyses and remote sensing derivatives to:

1. Compare performance indices (root mean square error and percent bias) between the models and ICOS measurements on a per-site and PFT-group basis to assess the systematic error and accuracy of ET and GPP simulations.

2. Investigate how the models represent the observed ET and GPP for different PFTs regarding their sub-annual averaged phenologies, standard deviation, and timing of important phenological events.

3. Evaluate the simulated PFT-level ET and GPP statistical distributions and their moments (mean, variance, skewness, and excess kurtosis) to contextualize assessments of factors, like droughts, which impact the shape of these distributions.

4. Compare the inter-site differences between ET and GPP time series within PFT groups to estimate how the observed intra-PFT variability is represented in the models.

Thus, these findings offer critical information for comparisons of GPP and ET from the evaluated models. Furthermore, this study also paves the way for a better-informed analysis of the drought response of ET and GPP from the models being assessed over Europe. We expect that:

1. There is a lower systematic bias, and the simulation is closer to the observations by the PFT scale than the grid-scale CLM5 outputs, remote sensing, and reanalysis data.

2. The remotely sensed and modeled data approximate critical events in the phenologies of ET and GPP within the standard deviation of the ICOS measurements for sites of one PFT. However, this ability varies between PFTs.

3. The remotely sensed and modeled ET and GPP data distributions show a lower range among the moments within the PFT groups than the ICOS measurements.

## 2. Methods and Data

### 2.1. Community Land Model version 5

We use the CLM5 (Lawrence et al., 2018, 2019), which is forced offline with custom input data. The land surface of a region in CLM5 is first disaggregated into grid cells, which are uniformly distributed and simulated individually. These grid cells are tiled into land units (i.e., natural vegetation, crops, lakes, urban areas, and glaciers) with a relative area coverage within the grid cell. Importantly, plants in the naturally vegetated land units compete for water in a single soil column. The vegetation is grouped into PFTs (Lawrence and Chase, 2007), which are distinguished through leaf habit (evergreen or deciduous), morphology (needle and broad leaves, grass, and shrubs), and the bioclimate of the grid cell location (boreal, temperate, and tropical). While competition for soil moisture includes interactions among different PFTs, this is closer to natural conditions than separated soil columns and encourages evaluations on the PFT scale. Here, we use CLM5-BGC, which calculates vertical carbon and nitrogen pools and fluxes between the vegetation, soil, and atmosphere. In the following subsections, we briefly describe the essential processes in CLM5 that are particularly relevant to this study, as well as the input data and leading features of the European CLM5 setup.

#### 2.1.1. Gross primary production and evapotranspiration

The stomatal conductance of plants ($g_s$) couples water exchange with carbon uptake between vegetation and the atmosphere. In CLM5, $g_s$ is calculated by the Medlyn stomatal conductance model (Medlyn et al., 2011):

$$g_s = g_0 + 1.6\left(1 + \frac{g_1}{\sqrt{D}}\right)\frac{A}{c_s} \tag{1}$$

Where $g_0$ is the Medlyn intercept and defaults to 100 mol m$^{-2}$ s$^{-1}$, and $g_1$ is the Medlyn slope, a PFT-specific parameter. D is the vapor pressure deficit indicating atmospheric water demand, and $c_s$ is the $CO_2$ partial pressure at the leaf surface relative to the total atmospheric pressure. A is the carbon assimilated through photosynthesis.

$$A = \frac{c_s - c_i}{1.6 r_s} \tag{2}$$

The calculation of A is adapted from Bonan et al. (2011). It is based on the Farquhar model (Farquhar et al., 1980) and limited by the photosynthetic capacity given by the LUNA model (Ali et al., 2016). It requires knowledge of the gradient of $CO_2$ concentration from the outside to the inside of the leaf and neglects $CO_2$ storage at the leaf

surface. $c_s$ and $c_i$ are the leaf surface and internal partial $CO_2$ pressures, and $r_s$ is the stomatal resistance, which is the inverse of $g_s$. Further, $c_s$ and $c_i$ are calculated.

$$c_s = c_a - 1.4r_b A \tag{3}$$

$$c_i = c_a - (1.4r_b + 1.6r_s)A \tag{4}$$

The factor 1.4 refers to the diffusivity ratio between $CO_2$ and $H_2O$ gases in the leaf boundary, and 1.6 is the same ratio in the stomata. The equations for A, $g_s$, $c_i$, and $c_s$ are computed iteratively until $c_i$ converges, using a hybrid algorithm with the secant method and Brent's method (Lawrence et al., 2018). The photosynthesis is scaled to the canopy GPP by considering the effect of sunlit to shaded area ratios of the total leaf area.

The water input from the atmosphere to the land surface can be snow accumulating on the ground, streamflow, lake water, intercepted by the vegetation canopy, or can infiltrate the ground. The water in the ground percolates through 20 soil layers and is stored, directly evaporated, or taken up by plant roots relative to their transpiration demand. Hydraulic stress in a plant is calculated in a hydraulic framework using Darcy's law for transient porous media flow (Bonan et al., 2014).

The transpiration flux T is calculated with the resulting $r_s$ from above.

$$T = \frac{e_s - e_i}{r_s} \tag{5}$$

$e_s$ is the $H_2O$ vapor pressure at the leaf surface, and $e_i$ is the saturation $H_2O$ vapor pressure resulting from the leaf temperature. If T cannot meet the atmospheric water demand because of a soil moisture shortage, CLM5-BGC introduces water stress and attenuates $g_s$ based on that transpiration deficit factor. Through decreased $g_s$, water stress also regulates photosynthesis, A.

Total evapotranspiration is then determined by summing the transpiration and evaporation from vegetation interception, surface water, the ground, and potentially snow.

**2.1.2. Setup of the European CLM5**

The European Coordinated Regional Climate Downscaling Experiment (CORDEX, Giorgi et al., 2009) domain delimited the extent of this study, matching with the extent of regional atmospheric models. With a resolution of 3

km (0.0275°), our grid contains 1544 × 1592 grid cells, including the ocean. We used stand-alone CLM5 with the activated BGC module and stub models for ice, sea, and waves.

The simulations were forced by the Consortium for Small-Scale Modeling (COSMO) Reanalysis 6 (Bollmeyer et al., 2015; Wahl et al., 2017), a 6 km resolution data set providing meteorological variables over the European CORDEX domain from 1995 to 2019. The main advantage of using this reanalysis is the high resolution and a better representation of seasonal precipitation intensities compared to a coarser resolved global reanalysis (Bollmeyer et al., 2015). Using this forcing in high-resolution LSM simulations should lead to a more accurate simulation of sub-surface and surface hydrological fluxes, especially in regions with a relatively heterogeneous land surface (Wahl et al., 2017; Prein et al., 2016).

The static surface information was initialized for the year 2000 and was determined using input data from a standard repository (Lawrence et al., 2018). These data include land use information from (Hurtt et al., 2020), PFT distribution maps from (Lawrence and Chase, 2007), soil texture from (IGBP, 2000), and slope and elevation taken from (Earth Resources Observation And Science (EROS) Center, 2017).

The CLM5-BGC needs initial conditions for the carbon pools. For that, a spin-up workflow is necessary to bring the carbon pools and fluxes of carbon to a steady state before starting with production simulations. The spin-up method consists of two steps. Firstly, an accelerated decomposition simulation step, where carbon pools are artificially minimized. Secondly, a conventional simulation step, growing the carbon pools to the desired equilibrium state. During both spin-up steps, the atmospheric forcing from 1995 to 2012 was cycled (i.e., a cycling period of 18 years). The progress towards a steady state is monitored by assessing the difference in total carbon fixed in the ecosystem between a selected year within the last 18-year cycling period and the same year in the previous cycling period. $C_{tot,y}$ is the total ecosystem carbon (including vegetation and soil) in the year y, and $C_{tot,y-t}$ is the complete ecosystem carbon in the year y-t. A grid cell's carbon pools are in carbon equilibrium if the following is fulfilled.

$$\frac{\Delta C_{tot}}{t} < 1 gCm^2 year^{-1} \qquad (6)$$

The following conditions define the final steady state on the continental scale.

1.  97% of the grid cells (and the total area) are in equilibrium.

2. The change in continental ecosystem carbon across the continent is lower than 2 Tg C year$^{-1}$ for the three preceding cycle periods.

The soil organic matter carbon pools in high northern latitudes were the slowest to reach equilibrium, which was reached after just about 1500 simulation years.

After the spin-up, we conducted a 24-year (1995 until 2018) transient simulation starting with the initial conditions established by the spin-up. We output the simulated variables from two model levels for the analyses.

1. **CLM5$_{PFT}$**: This is the model's native resolution of vegetation-related states and fluxes calculation. Using output at this level (not the default configuration) allows for multiple time series per grid cell, each corresponding to a single PFT. This enables a selection of modeled data as needed. For instance, when

comparing model data to ecosystem-level measurements, CLM5$_{PFT}$ relates to the simulated time series of the corresponding PFT, resulting in an adequate assessment of model functions. When comparing to in-situ observations, we will refer to CLM5$_{PFT}$ when we subset the ICOS site location and the agreeing PFT from the CLM5 data.

2. **CLM5$_{grid}$**: The grid cell level output aggregates the PFT and the other tiles (i.e., croplands, urban areas,

and lakes) that compose the grid cell area. Consequently, this data does not relate to a single functional type. Instead, it informs about the average state and fluxes in the grid cell area. In this study, CLM5$_{grid}$ designates CLM5 data extracted from the grid cell closest to the station's location.

## 2.2. Evaluation data

### 2.2.1. Station data

As ground truth data in the comparisons, we used the ICOS research infrastructure, which has a station observation network spanning 14 European countries (ICOS RI, 2021). Each station has at least one eddy covariance measurement tower and incorporates a processing workflow following a standardized protocol. We use the curated data, the WARM-WINTER-2020 data set (Warm Winter 2020 Team and ICOS Ecosystem Thematic Centre, 2022), which consists of homogenized variable time series following the ONEFLUX data pipeline (Pastorello et al., 2020).

The ICOS WARM-WINTER-2020 data has measurements of 73 stations totaling over 800 station-years (available years are station-dependent). corresponding to multiple land cover types (see Figure 1 for a map with the station locations and Table S1 for more information on the available years per station). Note that the land cover type

indicated by the ICOS site metadata and represented in the measurements refers to the predominant PFT in the footprint of the eddy covariance station. We omitted the stations over wetland and mixed forest land cover types to

ensure a coherent analysis because no PFT counterpart is implemented in CLM5$_{PFT}$. Also, shrub PFTs were not included in our analyses because there were insufficient shrubland sites in the ICOS data to support a robust evaluation. The analyses also excluded stations whose land cover type was not included in metadata sites (e.g., DEIMS-SDR https://deims.org), leaving a total of 42 stations for our analyses. Because the land cover types from the selected sites correspond well with PFTs in CLM5, we will also refer to them as PFTs.

The processing workflow of the WARM-WINTER-2020 data extracts daily time series for GPP, partitioned from the net ecosystem exchange (NEE) using the night-time method and a dependence on a variable friction velocity threshold (in g C day$^{-1}$, *GPP_NT_VUT_REF*). We retained negative GPP values in these data, which stem from the uncertainty of the NEE measurements and partitioning method, to avoid introducing bias into the GPP distributions (Reichstein et al., 2012; Pastorello et al., 2020). For the ET evaluation, we also extracted the gap-filled latent heat

flux (W m$^{-2}$, *LE_F_MDS*). Importantly, we verified our results by checking for inconsistencies in the analysis of ICOS NEE (*NEE_VUT_REF*), ecosystem respiration (*RECO_NT_VUT_REF*), and energy balance corrected latent heat flux (*LE_CORR*).

The conversion of latent heat (W m$^{-2}$) into ET (mm day$^{-1}$) is achieved by multiplying with the factor 0.035, assuming a constant enthalpy of vaporization decoupled from temperature because variable enthalpy has a

negligible effect on the overall outcome of the conversion.

Lastly, we use the leaf area index (LAI) from the ICOS Archive final quality data set (ETC L2 Archive). LAI is measured by only sparsely available starting from 2017 and, thus, only has two years intersecting with our study period (2017 and 2018). Furthermore, the data within this intersection period is only available for a smaller number (in relation to the EC data above) at ENF and CRO sites. Therefore, we do not include the analysis in the main text

but include these results only in the Supplementary Material for the context of the main analyses of ET and ET.

### 2.2.2. Remote sensing and reanalysis data

To assess CLM5 performance in the context of additional complementary data products, we include remotely sensed GPP data from the Global Land Surface Satellite (GLASS, Liang et al. (2021)). The GLASS GPP product uses the Moderate Resolution Imaging Spectroradiometer (MODIS) and Advanced Very High-Resolution

Radiometer (AVHRR) sensors and the revised Light Use Efficiency (LUE) model (Zheng et al., 2020) in 8-daily resolution in time and 0.05° resolution in space.We also compare the CLM5 outputs with GLASS ET data, which applies a multi-model ensemble (e.g., MODIS-ET, remote sensing Penman-Monteith ET) to remote sensing information to estimate 8-daily latent heat on a 0.05° grid. We convert latent heat to ET, as described in Section 2.2.1. Similarly, MODIS-derived GLASS LAI data (Ma and Liang 2022) is used in this study to provide context to the ET and GPP analyses (same 0.05° grid and 8-daily resolution).

Lastly, we use LAI and ET reanalysis data for evaluation, which fuse observations and models. Namely, they are the European Center for Medium-range Weather Forecasts Reanalysis 5 - Land product (ERA5L, Copernicus Climate Change Service (2019)), which has a spatial resolution of 0.1° and hourly temporal resolution, and the Global Land Evaporation Amsterdam Model (only ET, GLEAM version 3.5a, Martens et al. (2017)), which has a spatial resolution of 0.25° and daily temporal resolution.

## 2.3. Data processing

First, the remote sensing and reanalysis data are bilinearly remapped to the 3 km European CORDEX grid and interpolated to 8-daily means for 1995 - 2018. The ICOS observation time series are interpolated to 8-daily means for each station whose data availability overlaps with our study period. Then, we extracted the $CLM5_{grid}$, GLASS, ERA5L, and GLEAM data from the grid cell closest to the location of each selected ICOS station. Further, we select the time series in $CLM5_{PFT}$ that coincides with that grid cell and the station's predominant PFT. Importantly, we focus only on the four predominant PFTs represented in the entire ICOS station network: Evergreen Needleleaf Forest (ENF), Deciduous Broadleaf Forest (DBF), Grasslands (GRA), and Croplands (CRO), as outlined in Table 1. Finally, the periods where station data is absent or of bad quality (determined by the corresponding measurement or gap-filling quality flag in the ICOS data) are discarded from the simulations to ensure we are comparing the same set of conditions.

**Table 1: The predominant plant functional types (PFTs) in the Integrated Carbon Observation System (ICOS) WARM-WINTER-2020 observation dataset that correspond with the International Geosphere–Biosphere Programme (IGBP) land cover classifications, the number of corresponding sites, and the accordant PFTs in the European Community Land Model v5 (CLM5) setup.**

| ICOS IGBP PFT | Number of Stations | Corresponding CLM5 PFTs |
|---|---|---|
| Evergreen needleleaf forest (ENF) | 18 | Needleleaf evergreen tree – temperate<br>Needleleaf evergreen tree - boreal |
| Deciduous Broadleaf forest (DBF) | 8 | Broadleaf deciduous tree – tropical<br>Broadleaf deciduous tree – temperate<br>Broadleaf deciduous tree – boreal |
| Grasslands (GRA) | 8 | $C_3$ arctic grass<br>$C_3$ grass<br>$C_4$ grass |
| Croplands (CRO) | 8 | $C_3$ Unmanaged Rainfed Crop<br>$C_3$ Unmanaged Irrigated Crop |

The ICOS observations were also interpolated to 8-daily means, encompassing a time scale with significant variability of ecosystem processes (De Pue et al., 2023), to match the coarsest time resolution of other data sets (i.e., GLASS remote sensing) and thus to facilitate comparison of processes at the same scale. For a consistent comparison, the analyses only account for time steps where valid values are present for all data sources. We evaluate the data for each variable over each station and groups of stations with the same PFT.

## 2.4. Analyses

### 2.4.1. Yearly evolution and statistical distributions

We calculate ET and GPP PFT-specific phenology (mean sub-annual dynamics), resulting in day-of-year (DOY) plots. This is done by averaging the same 8-daily time step across years for each site and calculating the mean and standard deviation of site-specific DOY belonging to one PFT.

Further, we determined the statistical distributions as probability density functions resulting from the Gaussian kernel density estimate (Scott, 1992). Subsequently, the distribution moments (mean, variance, skewness, and

excess kurtosis) are calculated. The distributions and their moments are based on all 8-daily values corresponding to one PFT for each data source. The uncertainties of the distribution moments are calculated based on Harding et al. (2014).

### 2.4.2. Shift of phenological events

The three analyzed phenological events of ET and GPP – the start of the growing season, the peak, and the end of

the growing season – are determined for each PFT group and data source as the average DOY of the event among the stations and available years within that PFT group for each variable. The 8-daily time series of each variable was first smoothed with a 1-dimensional Gaussian filter to rule out potential errors due to small-scale variability and dampen the effect of potential outliers. More specifically, the peak timing is the mean DOY of the overall maxima of the smoothed averaged yearly evolution across stations for each PFT and data source. The start and the

end of the growing season were determined by the mean DOY of the two infliction points (Li et al. 2023; Lian et al. 2020; Whitcraft et al, 2015) of the smoothed yearly averaged evolution across stations for each PFT and data source. The shift of these events is simply the difference of the determined mean PFT-specific DOY between the models and the observations. As a measure of uncertainty of the mean PFT-specific DOY, we also calculate the standard deviation of the DOY of the events across stations in each PFT group.

### 2.4.3. Performance metrics

The percent bias (PBIAS) measures systematic model error and is calculated as follows.

$$PBIAS = \frac{\sum_{i=1}^{n} X_{S,i} - X_{O,i}}{\sum_{i=1}^{n} X_{O,i}} \times 100 \qquad (7)$$

Where n is the number of time steps, $X_{S,i}$ is the simulated value of the variable X at the time i, and $X_{O,i}$ is the observed value of the variable X at the time i. If the PBIAS for variable X is positive, the model overestimates; if negative, it underestimates the observed variable X. In our analysis, $X_i$ is the interpolated 8-daily mean.

Furthermore, we estimated the root mean square error (RMSE) to indicate model accuracy and the root mean square difference (RMSD) to indicate similarity. RMSE and RMSD are calculated the same. However, the term 'error' assumes the truthfulness of the reference data. Hence, we use the RMSD when comparing data only between models.

$$RMSE = RMSD = \sqrt{\frac{\sum_{i=1}^{n}(X_{S,i} - X_{O,i})^2}{n}} \tag{8}$$

A RMSE close to zero indicates that the model approximates the observations nicely. Similarly, a low RMSD
reveals a high similarity between the two analyzed series. We calculate these metrics on a per-station basis and a set of stations belonging to the same PFT.

### 2.4.4 Modified Taylor diagrams

The Taylor diagram (Taylor, 2001) depicts multiple model performance indices in a single diagram by making use of the relationship of the calculation terms of the standard deviation, correlation, and RMSE. Their relationship can
be summarized in the following equation of error propagation:

$$RMSE^2 = \sigma_O{}^2 + \sigma_S{}^2 - 2\sigma_o\sigma_S r \tag{9}$$

Where $\sigma_O$ is the standard deviation of the observations, $\sigma_S$ is the standard deviation of the simulation, and r is the Pearson correlation coefficient. The multi-variate diagram can be constructed due to the geometric relationship between these statistical indices through the law of cosines. Thereby, plotting the calculated Pearson correlation against the standard deviation of the models and the observation on a trigonometric polar plane, the RMSE
manifests as the polar Euclidean distance from the reference observations. We calculate the standard deviation and the Pearson correlation on the PFT-grouped stacked time series and plot these for e for each data source on one Taylor diagram per PFT. We modify the default Taylor diagram by scaling each marker's size by the absolute PBIAS for the corresponding source and PFT.

# 3. Results

## 3.1. Land surface representation

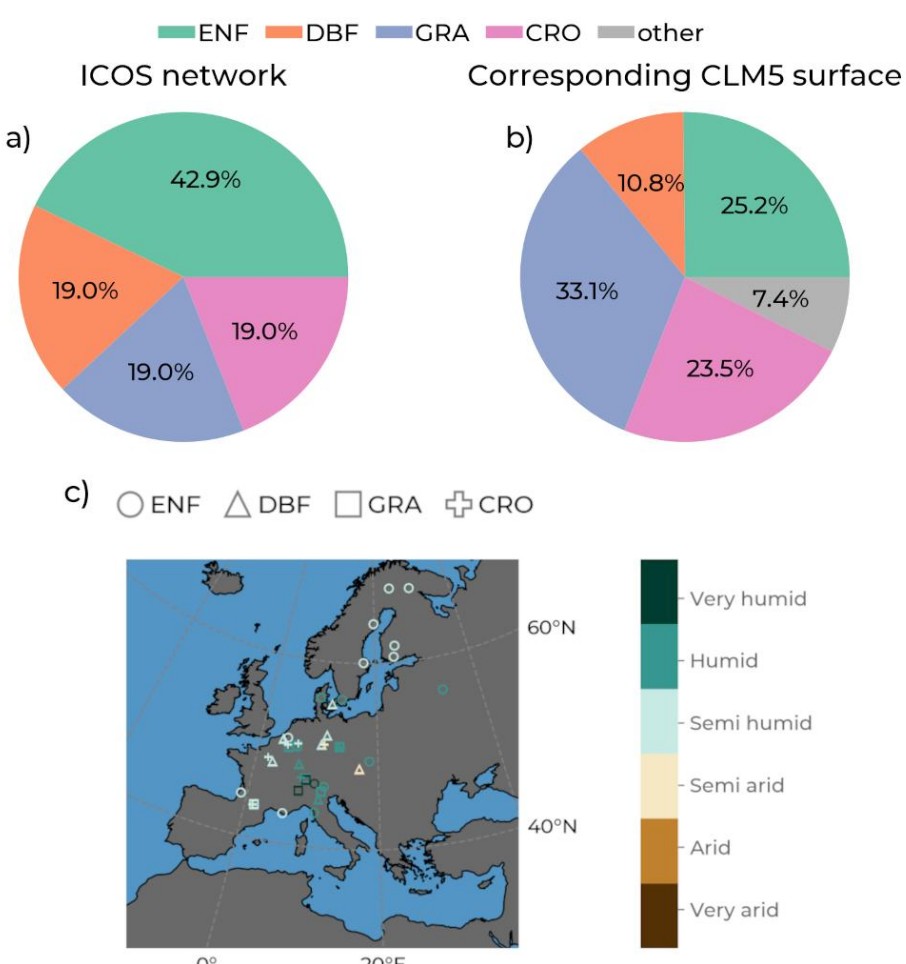

*Figure 1: The share of represented plant functional types (by color: Evergreen Needleleaf Forest (ENF, green), Deciduous Broadleaf Forest (DBF, orange), Grasslands (GRA, purple), and Croplands (CRO, pink)) in a) in the ICOS station network used in subsequent analyses and b) in the corresponding grid cells in our European CLM5 setup. In c) is a map showing the locations of the ICOS stations, with the marker type indicating their PFT and the color of the marker indicating their hydro-climate (adapted from Jafari et al. (2018)) based on the mean annual precipitation from the Consortium for Small-Scale Modeling (COSMO) -Reanalysis 6. Our 3 km European CLM5 simulation domain corresponds to the entire map box in c).*

Before evaluating the GPP and ET variables from CLM5 and how they are compared with observations, we first assess if the PFT composition of the entire ICOS station network is comparable to the PFT composition in the respective cells selected in CLM5$_{grid}$. This is important, as GPP and ET magnitudes, variability, seasonality, drought responses, and trends strongly depend on the present vegetation type. In Figure 1 we observe that ENF, the PFT of almost half of the present ICOS stations, represents only around a quarter of the corresponding CLM5$_{grid}$ area. DBF also covers a smaller share of the area in those grid cells than in the ICOS station network. On the other hand, GRA and CRO are overrepresented in CLM5$_{grid}$ compared to the share of respective ICOS stations. Consequently, when comparing with the ICOS observations, the selected data from CLM5$_{grid}$ data are, on average, overrepresenting the functionality of GRA and CRO and underrepresenting ENF and DBF, which hampers the evaluation of CLM5$_{grid}$ with in-situ ET and GPP. Hence, we also included the respective CLM5$_{PFT}$ GPP and ET in the subsequent analysis, enabling an accurate assessment of the functionality and relationships between PFT in the model. Additionally, we assess the similarities and differences between the two model scales, CLM$_{grid}$ and CLM$_{PFT}$, and their approximation to the observations.

## 3.2. General model performance

This section presents model performance indices correlation, RMSE, and PBIAS, comparing each model's ET and GPP estimates with measurements from the ICOS sites. We compared the RMSE and PBIAS on a per-site basis (Table S2 and Table S3), which yielded good results for most sites. The focus of this study, though, is the performance of PFT aggregations, combining data from sites that belong to the same PFT.

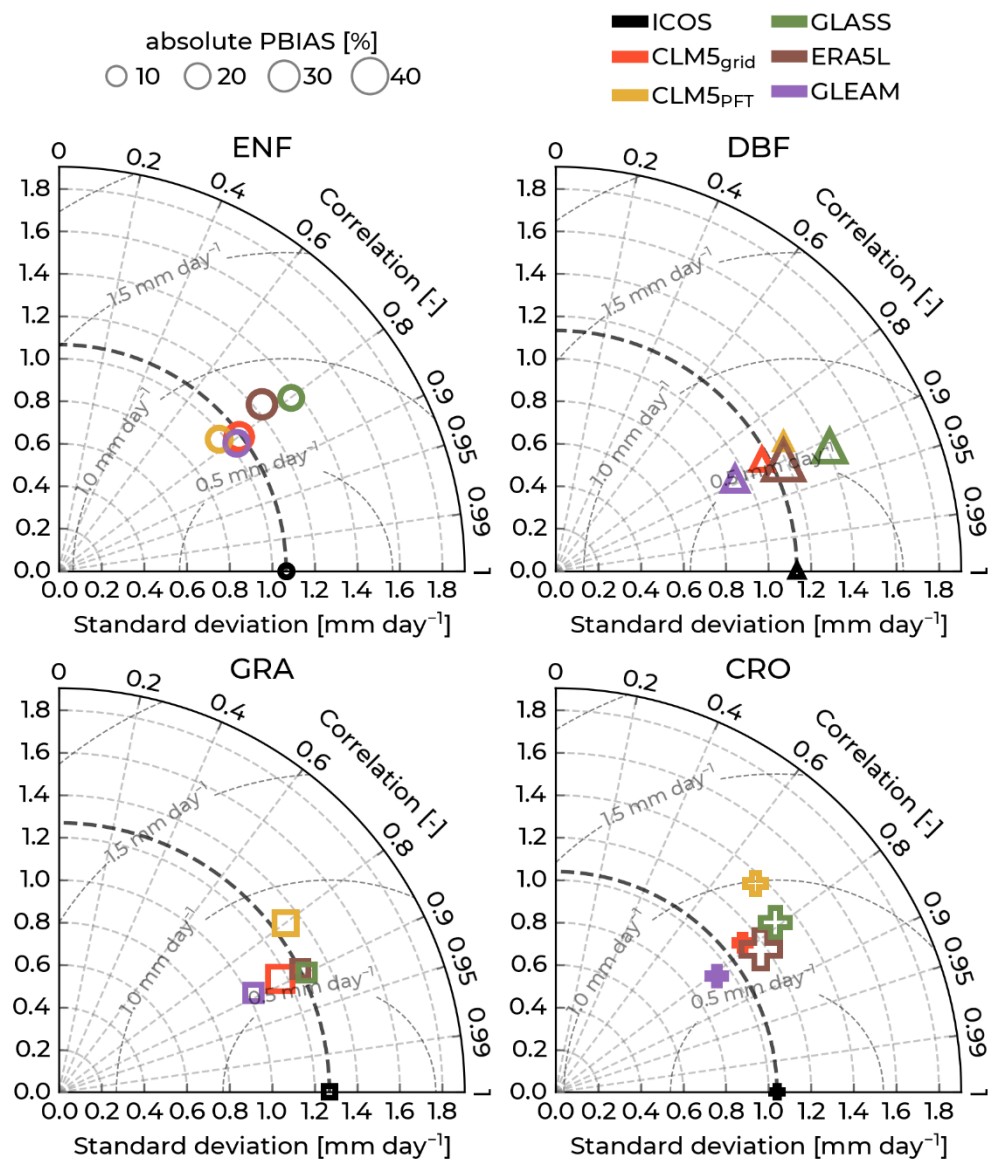

*Figure 2: Modified Taylor diagrams with observations from the Integrated Carbon Observation System (ICOS) of evapotranspiration as reference (black markers) and showing model performances between the years 1996 – 2018 (years varying by station; see Supplementary Table S1. Data sources by color: Community Land Model v5 (CLM5), CLM5grid: red, CLM5PFT: yellow, Global Land Surface Satellite (GLASS): green, European Center for Medium-Range Weather Forecasts Reanalysis 5 – Land (ERA5L): brown, Global Land Evaporation Amsterdam Model (GLEAM): purple). Each diagram shows these plots for one plant functional type. Upper left: Evergreen Needleleaf Forest (ENF, circles), upper right: Deciduous Broadleaf Forest (DBF, triangles), lower left: Grasslands (GRA, squares), and lower right: Croplands (CRO, crosses). The azimuth angle indicates the Pearson correlation with*

*the ICOS data, the radial distance is the standard deviation, and the semicircles centered at the reference standard deviation show the root mean square error (RMSE). The size of each marker indicates the percent bias (PBIAS).*

360 Figure 2 shows modified Taylor plots visualizing the performance indices of model ET against observations for each PFT. For more specific information, Supplementary Table S4 lists the number of ET 8-daily time steps that went into calculating these indices and their values. For ENF, all the models indicate a correlation of around 0.8 with the ICOS observations, and CLM5$_{grid}$, CLM5$_{PFT}$, and GLEAM have a similar variability to ICOS. CLM5$_{PFT}$ has a higher absolute RMSE and a smaller absolute PBIAS than CLM5$_{grid}$ for ET across PFTs, except in CRO.

365 Notably, the systematic bias in CLM5 is generally negative, with the same exception. On the other hand, ERA5L, GLASS, and GLEAM exhibit a general positive systematic bias for ET. ERA5L and GLASS show more significant deviations from the ICOS ET observations at ENF and DBF than CLM5$_{PFT}$ and CLM$_{grid}$ but have smaller RMSE values at GRA and CRO. GLEAM has generally low RMSEs and performs best among the models simulating ET at ENF and CRO. The most considerable systematic ET biases are found for ERA5L at CRO and DBF sites,

370 followed by GLASS for the same PFTs. The low absolute PBIAS of CLM5$_{PFT}$ across all PFT and the lower correlation than the other model data at GRA and CRO points to potentially missing or simplistic representations of eco-hydrological processes or management. Besides, all models approximate the ICOS ET observations fairly well, with correlations mostly over 0.8 but with partly high systematic biases by ERA5L at DBF and CRO sites.

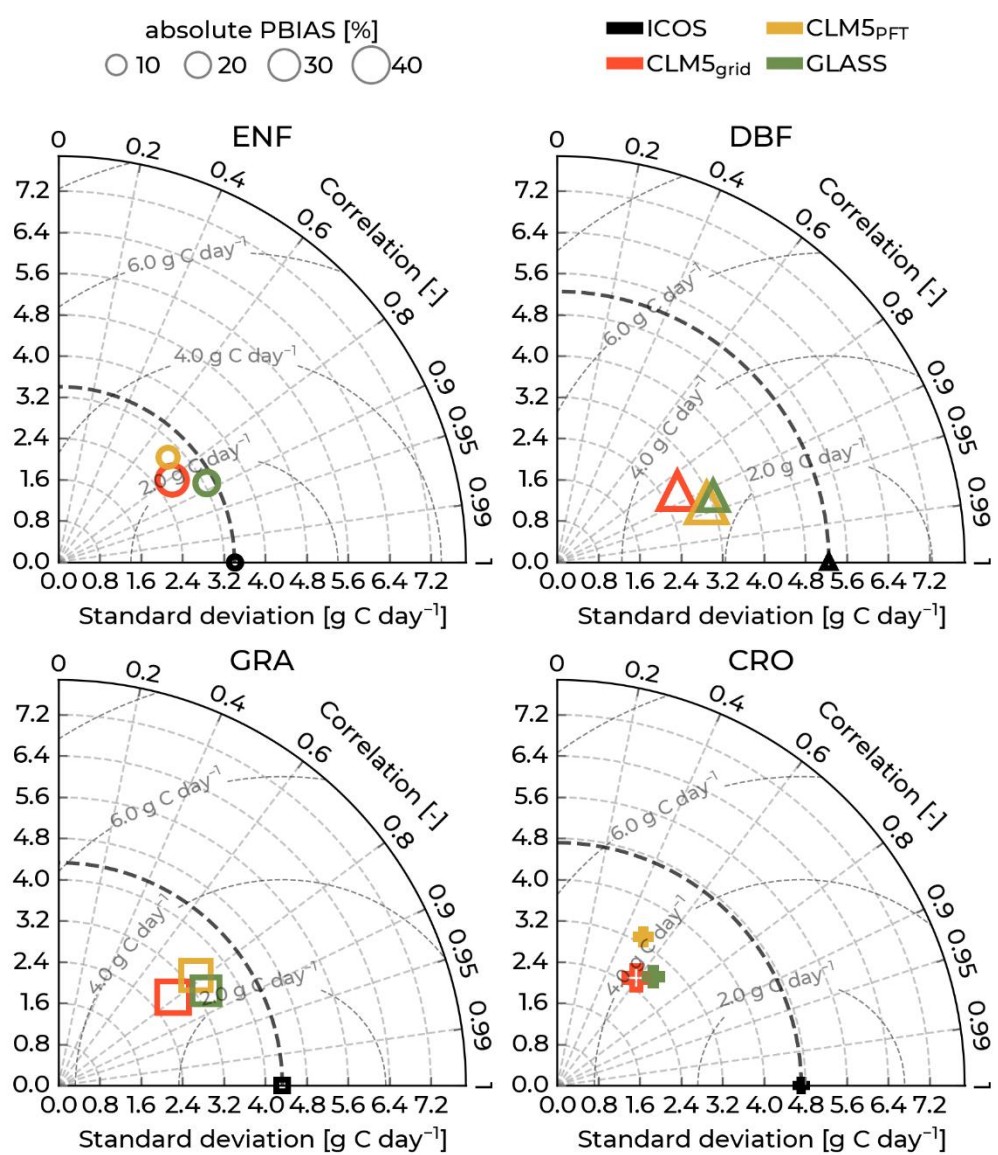

*Figure 3: Modified Taylor diagrams with observations from the Integrated Carbon Observation System (ICOS) of gross primary production as reference (black markers) and showing model performances between the years 1996 – 2018 (years varying by station; see Supplementary Table S1. For colors, labels, and acronyms, refer to Figure 2.In Figure 3, we show modified Taylor diagrams with the GPP performance indices of the models against the ICOS observations for each of the selected PFT. For more specific information, Supplementary Table S5 lists the number of GPP 8-daily time steps that went into calculating these indices and their values. CLM5PFT performed better than CLM5grid in approximating the ICOS GPP observations at DBF sites, showing a higher correlation and lower RMSE and GRA sites. Conversely, CLM5grid is closer to the observations for ENF and CRO PFTs. The GLASS data show the lowest GPP RMSEs and highest correlation values concerning ICOS measurements across*

*all PFTs. All models approximated the ICOS GPP best (lowest RMSE) at ENF, and the worst performance was at CRO sites. Furthermore, all models exhibit a negative, systematic bias in simulating the observed GPP across all PFTs. Especially at DBF and GRA PFTs, CLM5$_{grid}$, CLM5$_{PFT}$, and GLASS show large systematic underestimations of the measurements. CLM5$_{PFT}$ has a notably small PBIAS related to the ICOS data for ENF and CRO sites. Especially at CRO sites, all models showcase comparatively low correlation values ($<0.7$). While the correlation is high ($>0.75$) for all models at DBF and GRA sites, especially for CLM5$_{PFT}$ and GLASS at DBF sites (0.93 and 0.92), the high PBIAS hints that modeled data do not incorporate important processes or management practices that cause to the high carbon uptake at DBF sites over the long term. Because of the slowly evolving carbon states in the terrestrial ecosystems, the initial conditions of the carbon pools (e.g., soil organic matter, carbon in plant organs in the vegetation) could be a cause for the difference in the magnitude of the GPP.*

### 3.3. PFT phenology and its variability

#### 3.3.1. ET

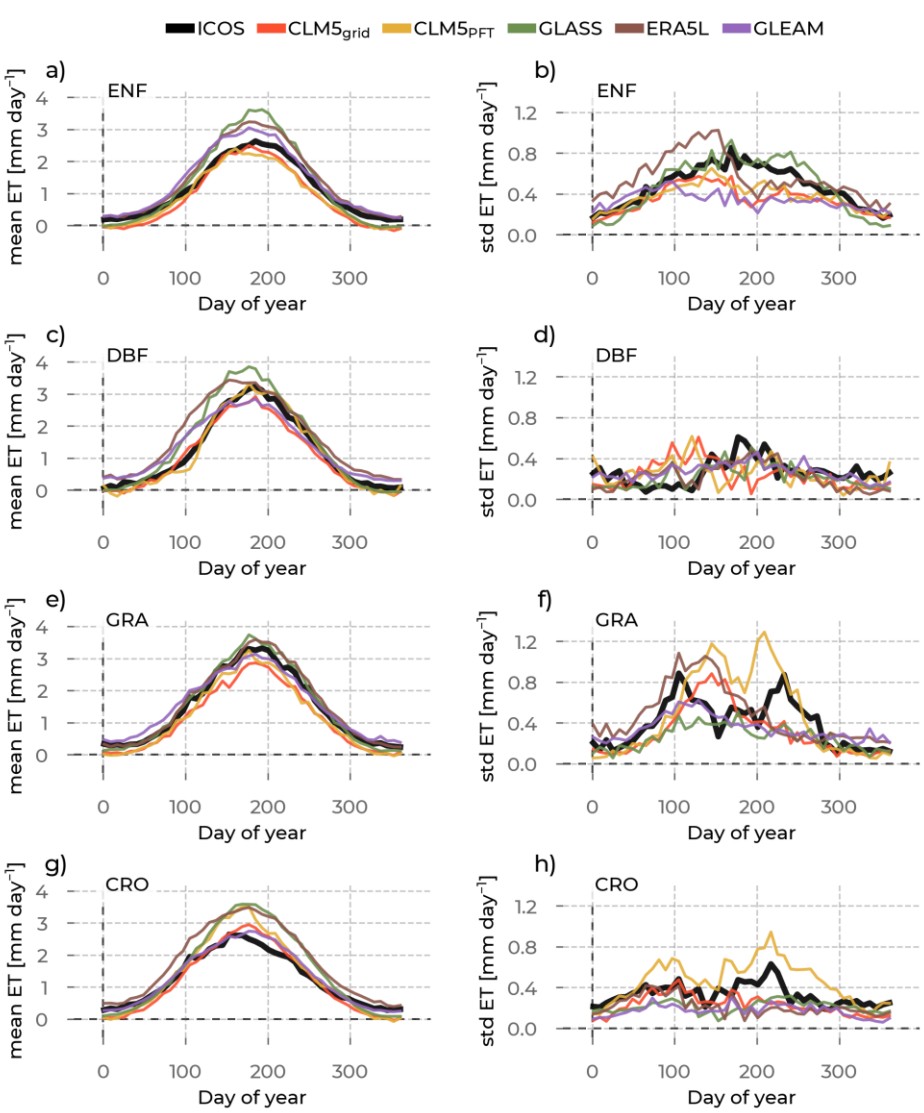

*Figure 4: In the left column are the yearly evapotranspiration (ET) evolutions averaged across stations belonging to one plant functional type (rows: Evergreen Needleleaf Forest (ENF), Deciduous Broadleaf Forest (DBF), Grasslands (GRA), and Croplands (CRO)) and across the years (available years vary per station, see Supplementary Table S1). We differentiate the data source by color (Integrated Carbon Observation System (ICOS) observations: black, Community Land Model v5 (CLM5), CLM5grid: red, CLM5PFT: yellow, Global Land Surface Satellite (GLASS): green, European Center for Medium-Range Weather Forecasting Reanalysis 5 – Land*

(ERA5L): brown, Global Land Evaporation Amsterdam Model (GLEAM): purple). The corresponding standard deviations across the sites and across the years are plotted in the right column to measure the spread around this mean.

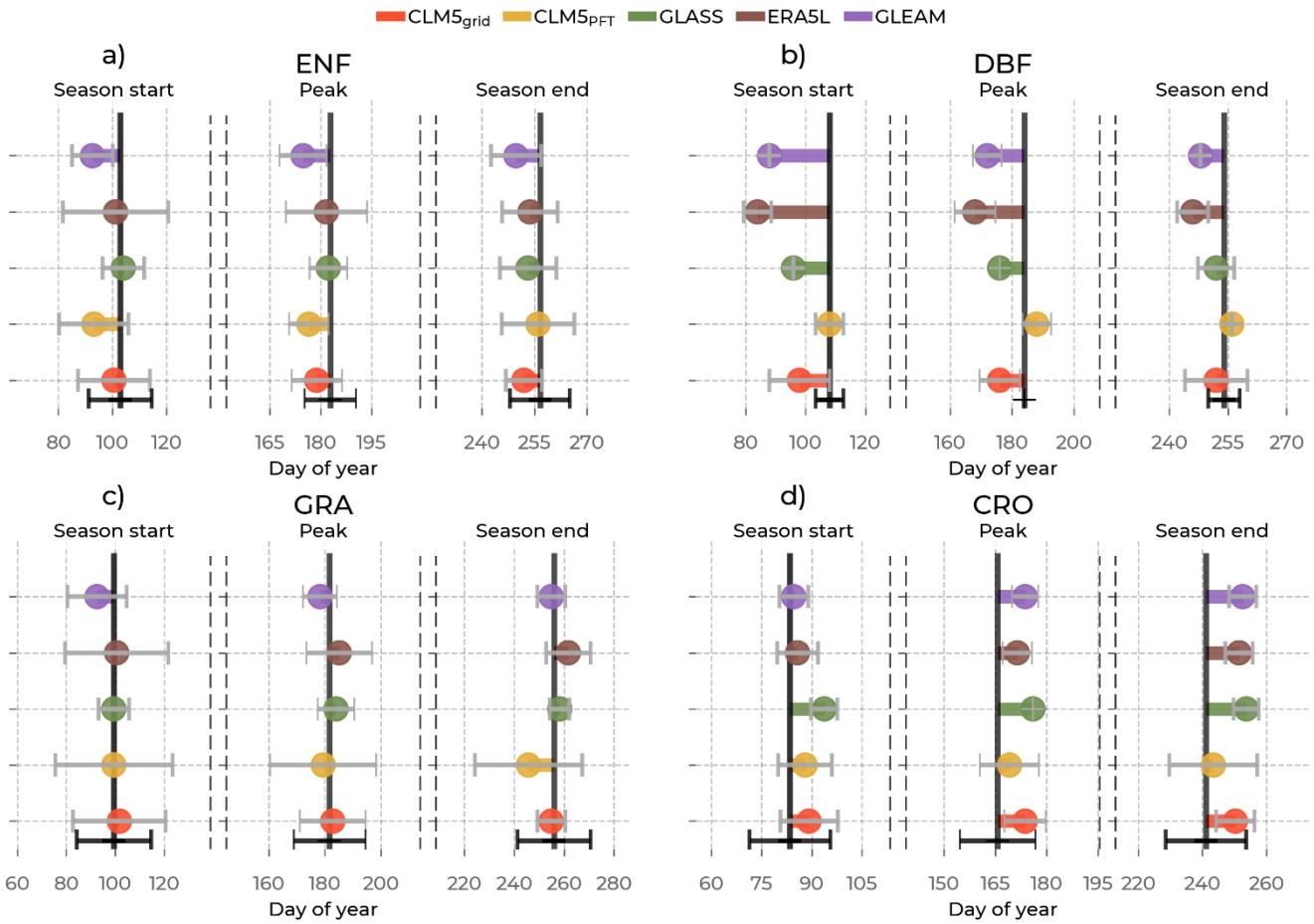

*Figure 5: Mean shifts in ET phenological events (the start of the growing season, peak, and the end of the growing season) between the Integrated Carbon Observation System (ICOS) observations (solid black line) and the models (by color: Community Land Model v5 (CLM5), CLM5$_{grid}$: red, CLM5$_{PFT}$: yellow, Global Land Surface Satellite (GLASS): green, European Center for Medium-Range Weather Forecasts Reanalysis 5 Land (ERA5L): brown, Global Land Evaporation Amsterdam Model (GLEAM): purple), among sites belonging to one plant functional type: Evergreen Needleleaf Forest (ENF), Deciduous Broadleaf Forest (DBF), Grasslands (GRA), and Croplands (CRO). On the x-axis is the day of the year of the event. Error bars in grey correspond to the standard deviation of the day of the event in the models across sites of one plant functional type, and the error bars in black correspond to the standard deviation across the respective observations.*

This section describes the results of the investigation on the mean and the standard deviation of the yearly evolution of ET across PFTs and data sources (Figure 4 a, c, e, g). We will analyze the ET mean and standard deviation for each PFT sequentially. On average, the annual evolution of ET for CLM5$_{grid}$ and CLM5$_{PFT}$ compares well to the ICOS measurements, as already hinted by the good correlation values in the previous section. They also capture the observed seasonal transitions between low winter ET and high summer ET well. Except for CRO sites, CLM5$_{grid}$ and CLM5$_{PFT}$ ET are slightly lower than the ICOS observations throughout the year, but especially in summer (mean PBIAS of -13.08 and -18.70%, respectively, see Supplementary Table S4). ERA5L and GLASS overestimate ET at sites of all PFTs, most predominantly in the ENF and DBF sites and during summer (mean PBIAS of +28.64 and +18.25%, respectively). The magnitude of variation across sites within each PFT (Figure 4 b, d, f, h) is captured well, generally showing smaller variation at DBF and CRO sites and larger variation at ENF and GRA. Some specific aspects of this variation across sites are captured best by CLM5$_{PFT}$: The bimodality of the intra-station variation at GRA sites across the year (Figure 4 f) and the peak variability across stations at CRO sites in the second half of the year (Figure 4 h). This exhibits the ability of CLM5$_{PFT}$ to differentiate ET between stations and the PFTs better than CLM5$_{grid}$ and the other models. The GLASS ET variability across stations compares remarkably well to the observed across ENF at DBF sites (Figure 4b and d).

Figure 5 reveals the shift in the timings of key phenological events based on ET (growing season start, summer peak, and growing season end) between each model and the ICOS observations. Generally, for ENF and DBF sites (Figure 5 a, b), all models show the earlier occurrence, and at CRO (Figure 5 d) they show a later occurrence of these phenological events than the measurements. CLM5$_{PFT}$ has the mean timing of the events within the standard deviation of the ICOS timing across all PFT. However, it shows a substantial variability, larger than observed in the event timings across GRA sites. Similarly, GLASS and CLM5$_{grid}$ show close approximations to the observed timings but simulate all these events significantly earlier at DBF sites and significantly later at CRO sites, with little variation in the timings across sites. The ERA5L and GLEAM data exhibit a much earlier growing season start (24 and 20 days earlier) and summer peak (16 and 12 days earlier, respectively) than observed by ICOS at GRA sites.

### 3.3.2. GPP

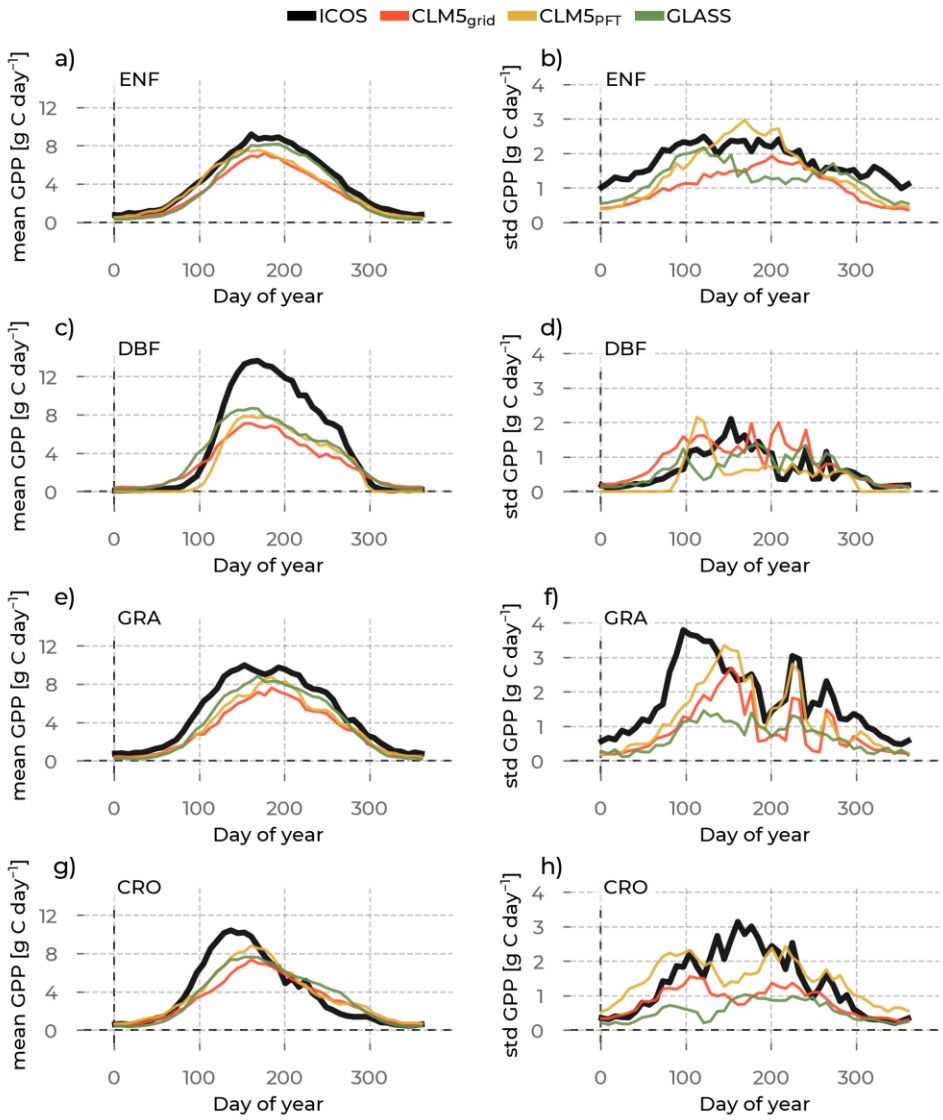

*Figure 6: In the left column are the yearly Gross Primary Production (GPP) evolutions averaged across stations belonging to one plant functional type (rows: Evergreen Needleleaf Forest (ENF), Deciduous Broadleaf Forest (DBF), Grasslands (GRA), and Croplands (CRO)) and across the years (available years vary per station; see Supplementary Table S1). We differentiate the data source by color (Integrated Carbon Observation System (ICOS) observations: black, Community Land Model v5 (CLM5), CLM5$_{grid}$: red, CLM5$_{PFT}$: yellow, Global Land Surface Satellite (GLASS): green). The corresponding standard deviations across the sites and across the years are plotted in the right column to measure the spread around this mean.*

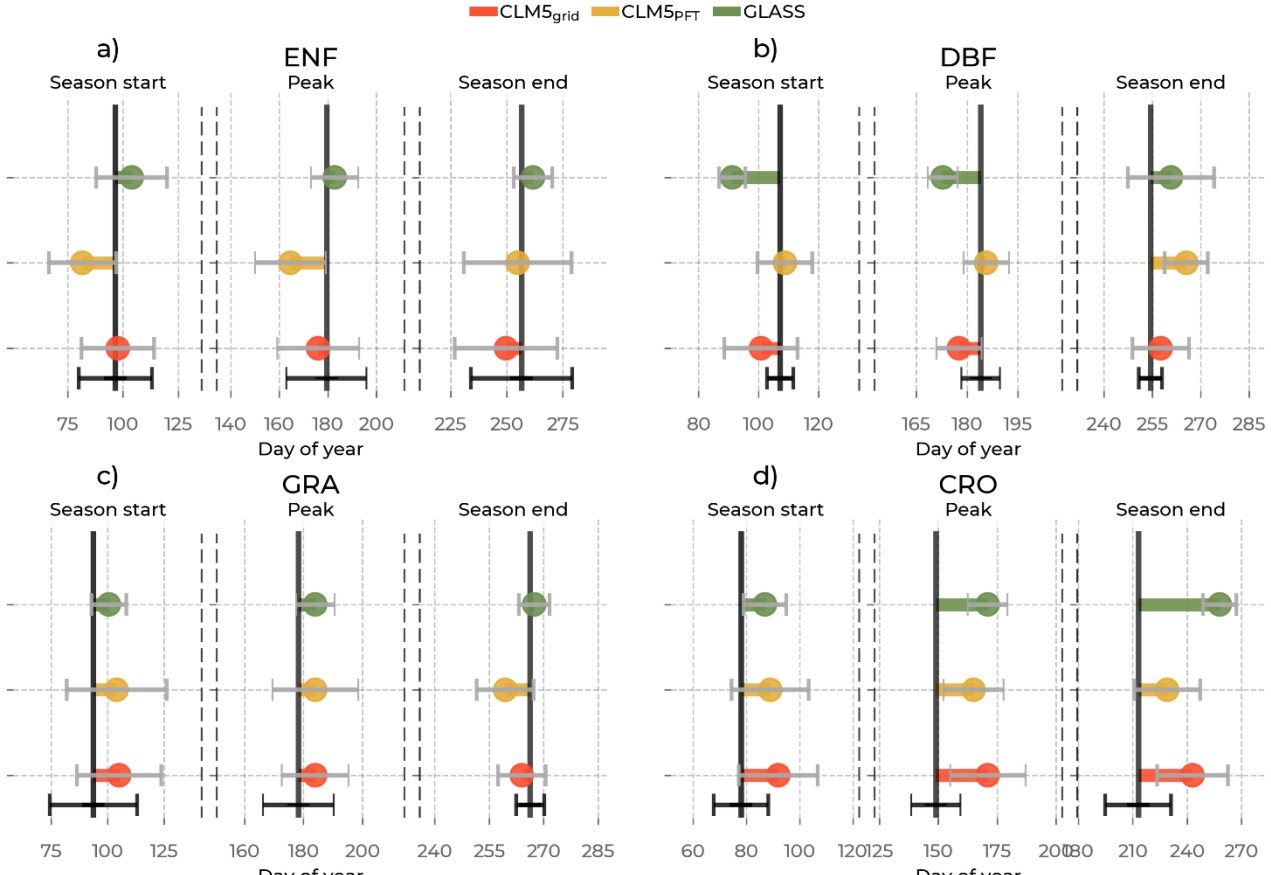

*Figure 7: Mean shifts in GPP phenological events (the start of the growing season, peak, and the end of the growing season) between the Integrated Carbon Observation System (ICOS) observations (solid black line) and the models (by color: Community Land Model v5 (CLM5), CLM5$_{grid}$: red, CLM5$_{PFT}$: yellow, Global Land Surface Satellite (GLASS): green), among sites belonging to one plant functional type: Evergreen Needleleaf Forest (ENF), Deciduous Broadleaf Forest (DBF), Grasslands (GRA), and Croplands (CRO). On the x-axis is the day of the year of the event. Error bars in grey correspond to the standard deviation of the day of the event in the models across sites of one plant functional type, and the error bars in black correspond to the standard deviation of the respective observations.*

405 The GPP values of all PFTs show a summer peak and a low period in winter (Figure 6). The negative values present in the ICOS measurements are caused by the processing of the measurements by ICOS and are, therefore, not represented by CLM5 or GLASS. Again, a general underestimation of observed GPP is shown across all PFTs (Figure 6 a, c, e, g), particularly during the summer months from all models. CLM5$_{PFT}$ shows larger GPP than CLM5$_{grid}$ and, therefore, has a lower systematic bias (mean PBIAS across PFTs of -19.61 and -27.65%. see

410 Supplementary Table S5). GLASS GPP is closer to the ICOS GPP at ENF, DBF, and GRA, and has the lowest

mean PBIAS across PFTs of -16.67). The most substantial underestimations are at DBF during summer (Figure 6 c), where CLM5$_{grid}$ and CLM5$_{PFT}$ have a PBIAS of -38.88 and -43.76%, and GLASS -24.52%. The GPP variability across sites is, similarly to ET, lowest at DBF sites. Notably, GLASS remote sensing GPP underestimates the variability among sites of one PFT substantially throughout the year at GRA and CRO sites (Figure f, h). The observed variability dynamics across the year, e.g. the bimodality at GRA sites (Figure 6 f) that was also visible for ET, is captured best by CLM5$_{PFT}$. However, not all models capture the behavior of CRO GPP inter-site variability (Figure 6 h). This supports the suspicion of the influence of management and missing processes in CRO in the models, possibly concerning the timings of planting, fertilizing, and harvesting the crops as the cause of these mismatches. The overall negative systematic bias in the models points at potentially missing sensitivities to or lower levels of, e.g., atmospheric CO2 and VPD that have been recently found to increase the water-use efficiency and carbon assimilation (Poppe Terán et al. 2023; Friedlingstein et al. 2023).

Shifts in phenological events between the observations and the models are already noticeable in Figure 6, but are quantified and visualized in detail in Figure 7. CLM5$_{PFT}$ and CLM5$_{grid}$ predominantly simulate the timing of these events within the standard deviation across ICOS stations for each PFT. In the GLASS GPP data, the events are more shifted from the measurements, most notably at DBF sites (16 days earlier growing season start and 11 days earlier summer peak) and at CRO sites (22 days belated peak and 45 days belated end of the growing season). Generally, in both CLM5 scales, the shifts to the ICOS observations were the largest in CRO. Similar to the ET event timings, CLM5$_{PFT}$ shows the largest variation of these timings among the models, but especially at GRA sites, and also considerable differences in the timing of the growing season end of ENF sites. These findings confirm the ability of CLM5$_{PFT}$ to approximate PFT-specific variation of ecosystem processes, but the contrasting results of the model performance indices will be further reviewed in the Discussion section.

## 3.4. Statistical distributions

### 435 3.4.1. ET

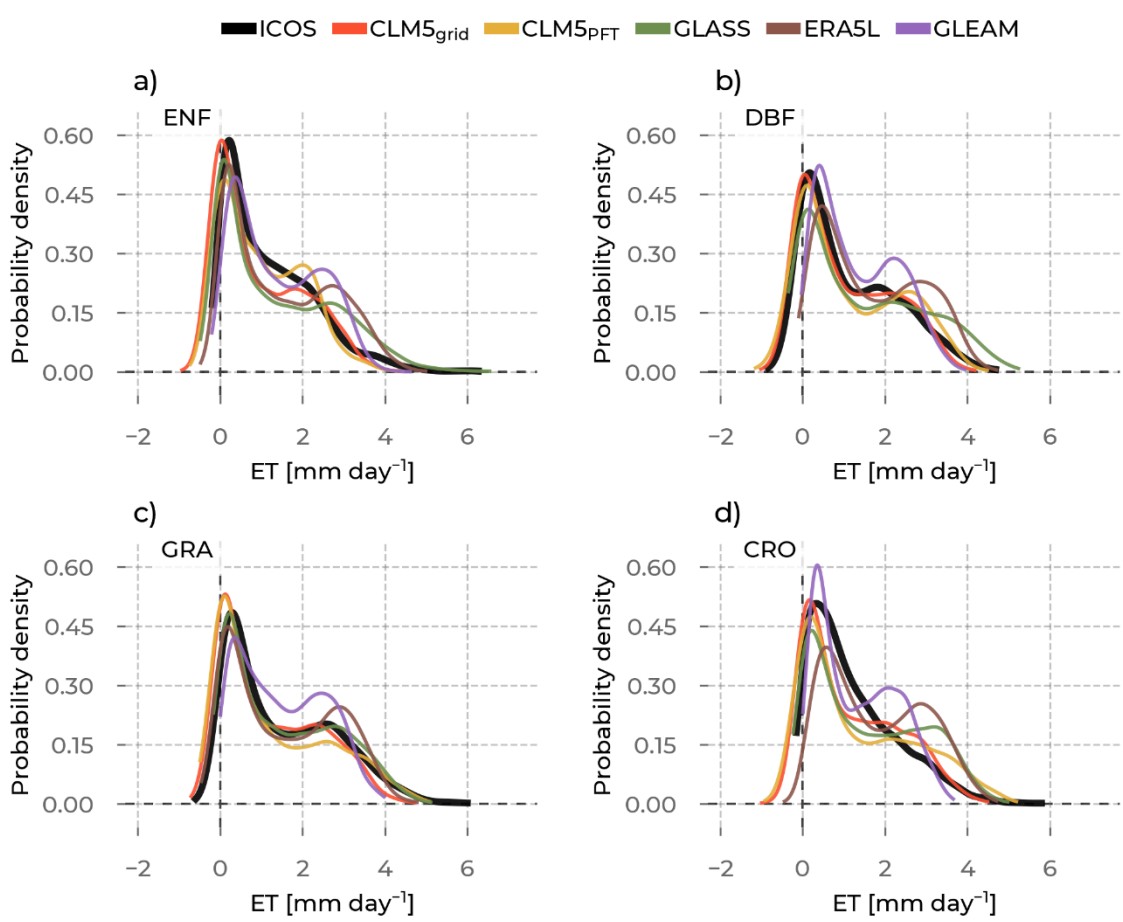

*Figure 8: The probability density curves for all evapotranspiration (ET) values from stations belonging to the selected plant functional types: Evergreen Needleleaf Forest (ENF), Deciduous Broadleaf Forest (DBF), Grasslands (GRA), and Croplands (CRO). The data source differs by color (Integrated Carbon Observation System (ICOS) observations: black, Community Land Model v5 (CLM5), CLM5$_{grid}$: red, CLM5$_{PFT}$: yellow, Global Land Surface Satellite (GLASS): green, European Center for Medium-Range Weather Forecasts Reanalysis 5 Land (ERA5L): brown, Global Land Evaporation Amsterdam Model (GLEAM): purple).*

In this section, we describe the results of the statistical distributions of ET in the model and the observations for each PFT. Then, we give more details on the moments of these distributions and how the models compare to the observations. Generally, the models approximate well the shape of the distributions (Figure 8), with a pronounced peak in the occurrence of positive ET values close to 0 that represent low winter values across all PFTs and, moreover, the slowly decreasing frequency of values towards the high ET summer values, which is more variable among the models. The variability of the summer peak magnitude (see previous section) among stations of the same PFT causes the ICOS and CLM5 ET distributions to have only a slightly pronounced second mode at the high summer ET values. On the other hand, the ERA5L and GLEAM ET distributions show a very pronounced second mode at the higher ET values for each PFT. This hints at the lower variability of the summer peak magnitude among these stations, which misrepresents the observed high variation in ICOS.

The moments of these distributions give more insights into their specific characteristics. Further, differences in moments between the observations and the models can yield important information on potential misrepresentations (Figure 9). For example, a differing mean between ICOS and a model points to a general shift in the distribution, specifically its center of mass. Therefore, we confirm a shift of ET distributions of ERA5L, GLASS, and GLEAM towards higher values for all PFTs in reference to ICOS. $CLM5_{grid}$ and $CLM5_{PFT}$ have lower means, except for $CLM5_{PFT}$ at CRO. The second moment, the variance, informs about the variability of values. Notably, GLEAM data underestimate, and GLASS data overestimate the observed variability of ET at all PFTs. $CLM5_{PFT}$ has a broad range of variability across PFTs, which corresponds well with ICOS observations, while $CLM5_{grid}$ and the other models show a very similar level of variability independent of the PFT. All models agree with the observed positive sign of the skewness (indicating a longer right tail of the distribution) for all PFTs. And while all the models simulate a platykurtic (negative excess kurtosis, pronounced relative tailedness) characteristic of the distributions across PFTs, ICOS shows leptokurtic (positive excess kurtosis, less pronounced tails, and more pronounced peak) behavior at ENF and CRO sites. Furthermore, the variation of the reach model's skewnesses and kurtoses (y-axis ranges for each color in Figure 9 c and d) across the PFTs are considerably lower than the observed ranges (corresponding x-axis ranges). Altogether, these findings showcase the ability of $CLM5_{PFT}$ to model intra- and inter-PFT ET variance better than the other considered models on the one hand, but shortcomings of all the considered models to represent the variation of the extreme ends of the ET distributions across all PFTs.

465

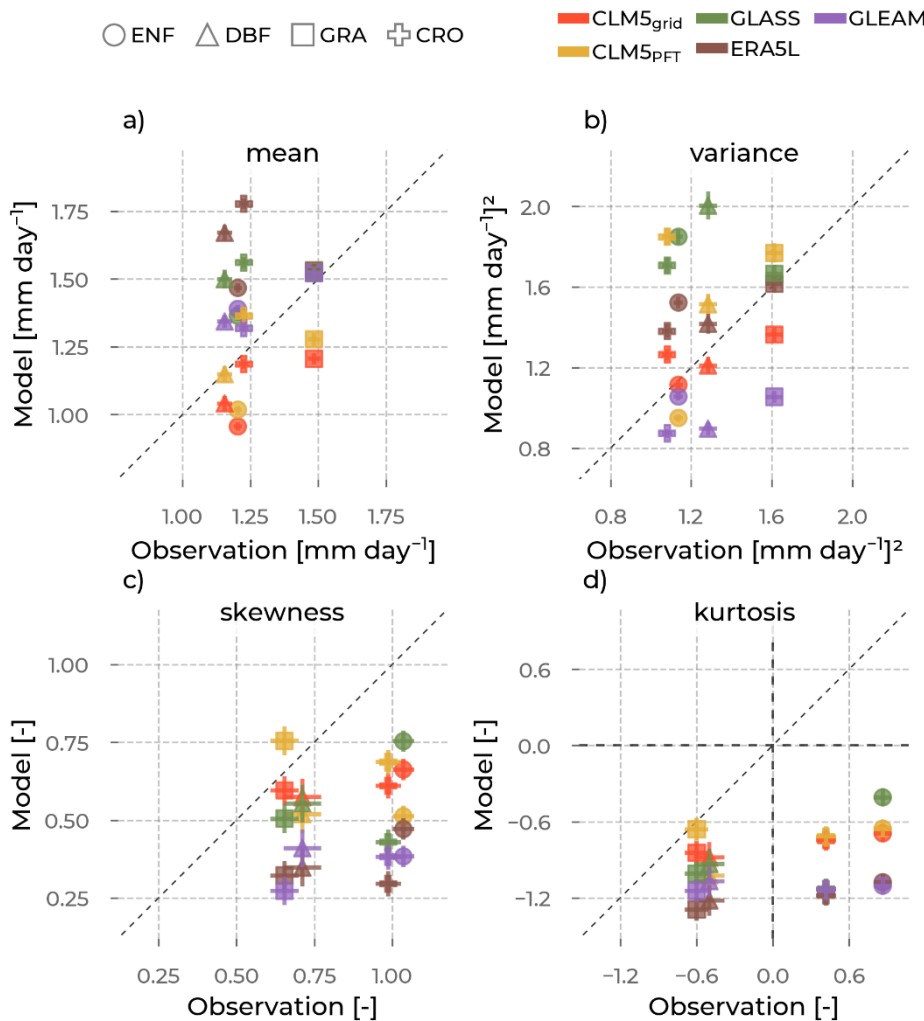

*Figure 9: The mean (a), variance (b), skewness (c), and excess kurtosis (d) of the evapotranspiration (ET) distributions (visualized in Figure 8) from the models (y-axis, colors: Community Land Model v5 (CLM5), CLM5 grid: red, CLM5PFT: yellow, Global Land Surface Satellite (GLASS): green, European Center for Medium-Range Weather Forecasts Reanalysis 5 Land (ERA5L): brown, Global Land Evaporation Amsterdam Model (GLEAM): purple), as opposed to the corresponding values from observations (x-axis) aggregated for each plant functional type (marker type): Evergreen Needleleaf Forest (ENF), Deciduous Broadleaf Forest (DBF), Grasslands (GRA), Croplands (CRO). The error bars are the standard errors of the respective moment, depending on the sample size.*

### 3.4.2. GPP

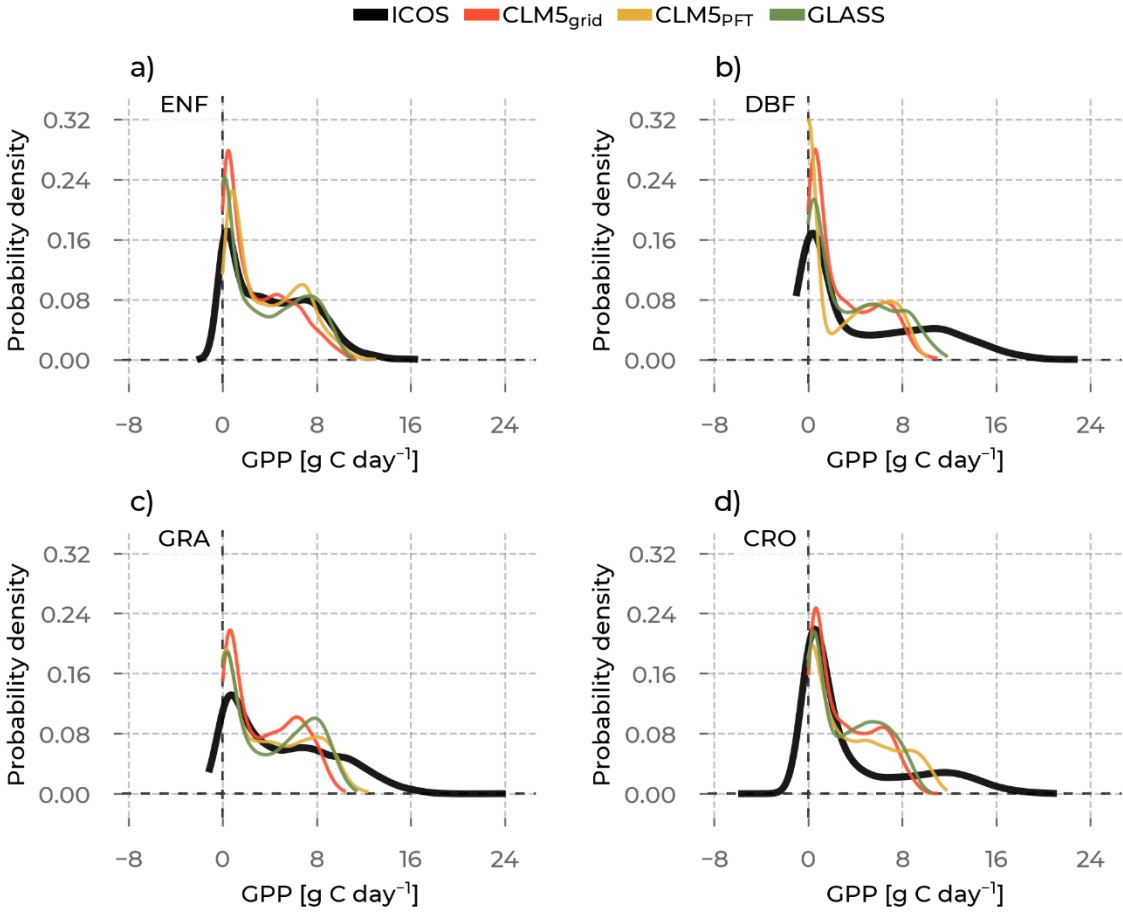

*Figure 10: The probability density curves for all Gross Primary Production (GPP) values from stations belonging to the selected plant functional types are shown: Evergreen Needleleaf Forest (ENF), Deciduous Broadleaf Forest (DBF), Grasslands (GRA), Croplands (CRO). The data source differs by color (Integrated Carbon Observation System (ICOS) observations: black, Community Land Model v5 (CLM5), CLM5$_{grid}$: red, CLM5$_{PFT}$: yellow, Global Land Surface Satellite (GLASS): green).*

We continue to delineate the results of the same analyses for the GPP distributions and their moments (Figure 10). The frequency peaks at the low GPP values, which correspond to the base winter GPP, are overestimated by all models at ENF, DBF, and GRA. This could partly be explained by negative GPP values in the ICOS data, which the models do not represent. By definition, there is no negative GPP. However, these negative values are given through the uncertainty range of the NEE partitioning method and are retained in the analysis to preserve the partitioning distribution (Reichstein et al. 2012, Pastorello et al. 2020). This is probably related to underestimating the observed winter GPP in ENF and GRA sites seen in Figure 6 a and e. Another striking finding is the missing occurrence of the highest observed GPP values in the models at all PFTs, but most noticeable at DBF sites, where the upper half range of GPP values ($>12$ g C day$^{-1}$) is not represented in any model. The overrepresented mid-range GPP values and the partly pronounced second modes in the mid-range GPP values across PFTs are possibly caused by the low summer peaks and low variability across sites (see Figure 6).

The models show lower GPP means than the ICOS measurements for all PFTs in Figure 11 a. Similarly, for all models across all PFTs, the underestimated GPP variance indicates a lower spread of the PFT distributions than in ICOS. While models agree on the positive skewness of the GPP distribution (skewed to the left), the largest skewness at CRO sites is not well represented by all the models. Finally, similar to the findings with ET kurtosis, the models fail to distinguish the distinct leptokurtic characteristics (less heavy tails) of the GPP distribution of CRO sites compared to the other PFTs, as seen in the observations. Across PFTs and for all models, the ranges spanned by the intra-PFT distribution moments are smaller than the observed. Most strikingly, the GPP variance range across PFTs, which is, among the models, the largest for CLM5$_{PFT}$ (between 8 g C day$^{-1}$ and 12 g C day$^{-1}$), is much smaller than for ICOS (11 g C day$^{-1}$ to 27 g C day$^{-1}$). This suggests the models do not simulate GPP differently enough between the PFT groupings. Thus, model development and parameter optimization studies that aim to improve these representations should focus on enhancing the variability at DBF.


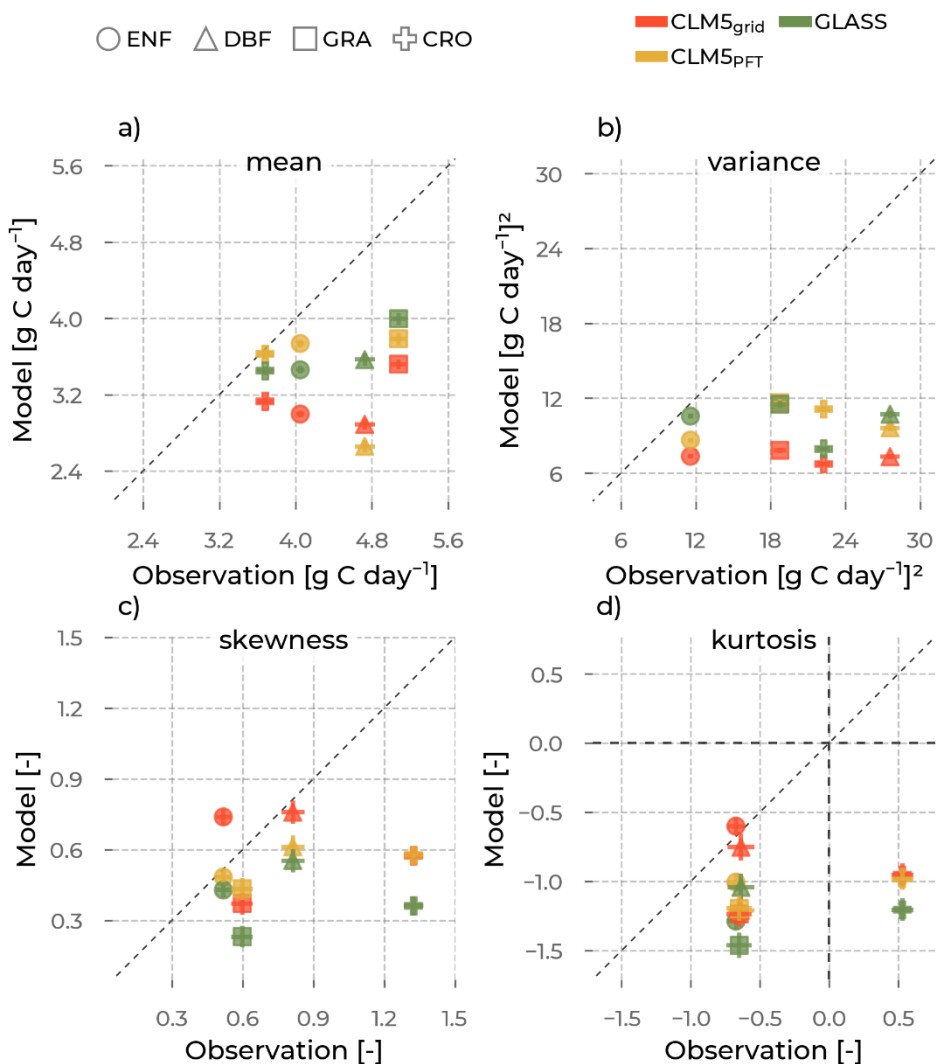

*Figure 11: The mean (a), variance (b), skewness (c), and excess kurtosis (d) of the gross primary production (GPP) distributions (visualized in Figure 10) from the models (y-axis, colors: Community Land Model v5 (CLM5), CLM5 grid: red, CLM5PFT: yellow, Global Land Surface Satellite (GLASS): green), as opposed to the corresponding values from observations (x-axis) aggregated for each plant functional type (marker type): Evergreen Needleleaf Forest (ENF), Deciduous Broadleaf Forest (DBF), Grasslands (GRA), Croplands (CRO). The error bars are the standard errors of the respective moment, depending on the sample size.*

### 3.5. The inter-site similarity of PFT groups

To support the interpretations of our findings, we quantify the similarity of ET and GPP across sites of the same PFT and compare the differences between the models and the observations. In this section, we analyze the mean RMSD of each PFT per ET and GPP data sources. A low RMSD indicates that the stations corresponding to one PFT are similar, while a high RMSD hints at a greater diversity within the PFT. By comparing the mean RMSD per PFT for ET and GPP across data sources, we can evaluate how much diversity is captured in the data of a particular PFT in the observations and models. The standard deviation of the RMSD for each PFT gives information on the spread of the inter-site RMSDs within the PFT group around that mean.

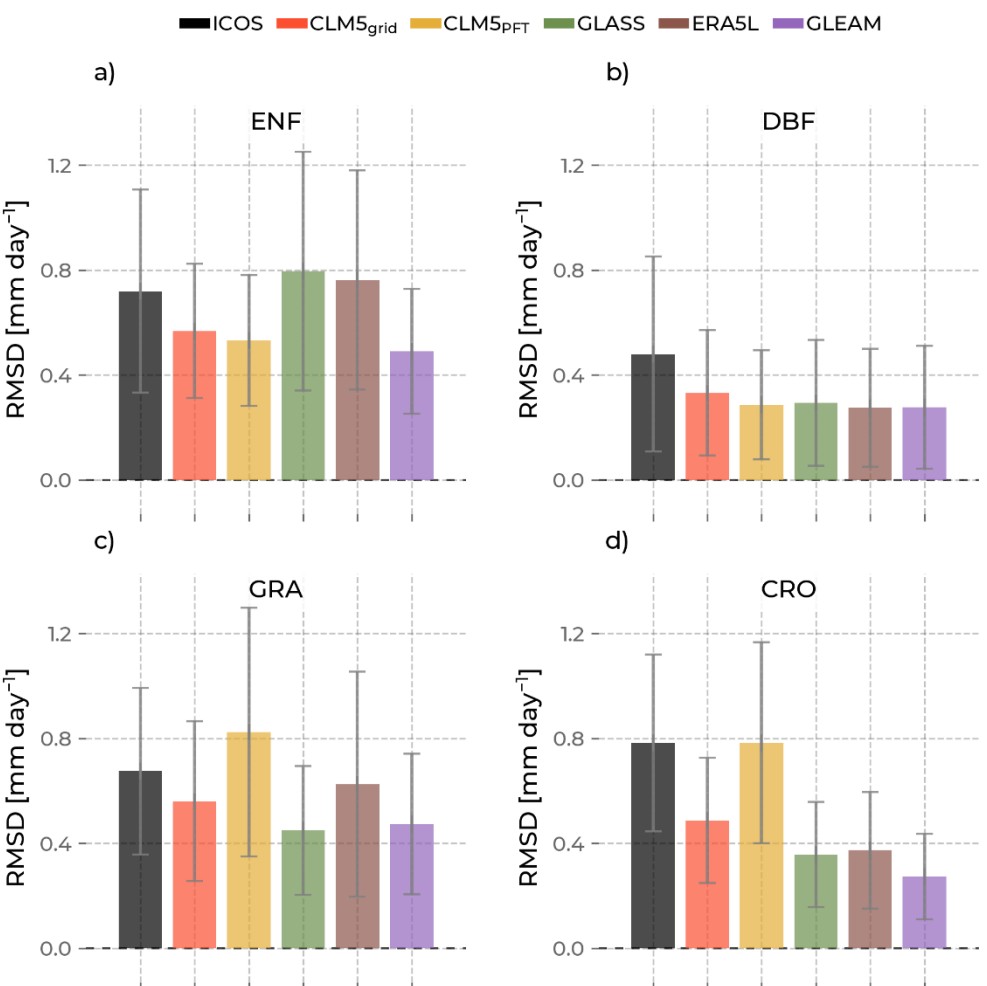

*Figure 12: The bars indicate the mean of the root mean square difference (RSMD) of evapotranspiration calculated for sites with the same plant functional type. The error bars are their standard deviation. Low values indicate high*

*similarity between the sites, and high values show high dissimilarity. The color of the bars differentiates the data source (Integrated Carbon Observation System (ICOS): black, Community Land Model v5 (CLM5), CLM5$_{grid}$: red, CLM5$_{PFT}$: yellow, Global Land Surface Satellite (GLASS): green, European Center for Medium-Range Weather Forecasts Reanalysis 5 Land (ERA5L): brown, Global Land Evaporation Amsterdam Model (GLEAM): purple).*

Figure 12 shows that CLM5$_{grid}$ and GLEAM have lower ET time series differences between the corresponding sites for all PFT than ICOS. CLM$_{PFT}$ has a lower mean RMSD than CLM5$_{grid}$ among ENF and DBF sites. Both CLM5$_{PFT}$ and CLM5$_{grid}$ underestimate the observed diversity of ET at ENF and DBF sites. Interestingly, the variation of ERA5L and GLASS ET time series for ENF is higher than observed, and they also show the most significant variation of RMSD. Meanwhile, DBF's mean RMSD of all models is lower than that of ICOS. CLM5$_{PFT}$ shows a

higher diversity of ET between GRA sites and CRO sites than CLM5$_{grid}$. The CLM5$_{PFT}$ surpasses the observed mean RMSD for the GRA PFT, highlighting the potential to simulate GRA sites variably. All other models underestimate it slightly (CLM5$_{grid}$, ERA5L) or more pronouncedly (GLASS, GLEAM). Particularly at CRO sites, the ET RMSD of CLM5$_{PFT}$ is substantially higher than the other models and at a similar level as ICOS observations. In contrast, all other models show significantly lower mean RSMDs there. Generally, a higher ET RMSD mean in

a PFT group comes with a higher spread (higher standard deviation) for all data sources. The RSMD in ET between stations is lower for CLM5$_{grid}$ and GLEAM than for ICOS for all PFTs.

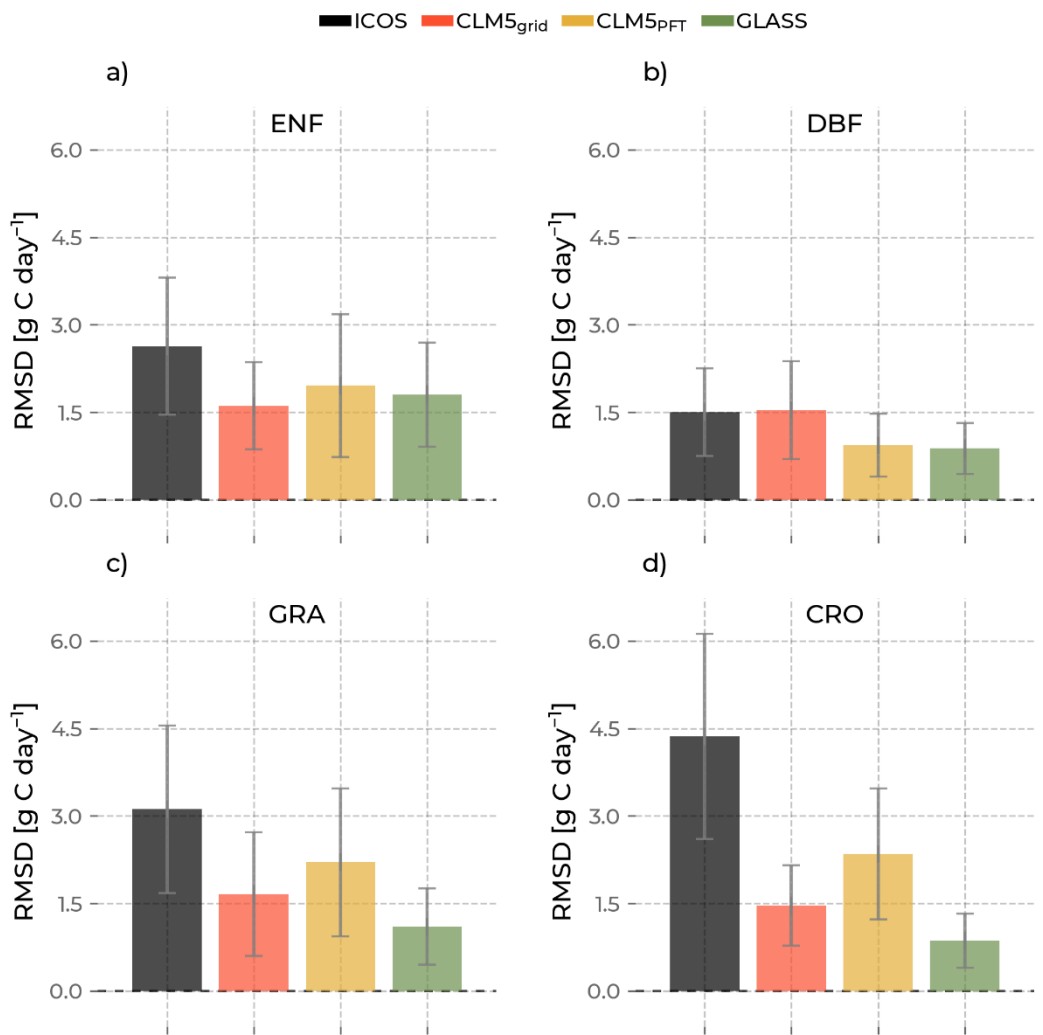

*Figure 13: The bars indicate the mean of the root mean square difference (RSMD) of gross primary production calculated for sites with the same plant functional type. The error bars are their standard deviation. Low values indicate high similarity between the sites, and high values show high dissimilarity. The color of the bars differentiates the data source (Integrated Carbon Observation System: black, Community Land Model v5 (CLM5), CLM5grid: red, CLM5PFT: yellow, Global Land Surface Satellite (GLASS): green).*

Figure 13 shows that for GPP, the models generally have a lower mean RSMD than ICOS across stations for all
PFT, except for CLM5$_{grid}$ at DBF. CLM5$_{PFT}$ has a more diversely simulated ET across ENF, GRA, and CRO sites than CLM5$_{grid}$. Interestingly, the observed magnitude of the RMSD is lowest for DBF and highest for CRO and has a more extensive range across PFTs than the models. For example, the RMSDs of ICOS data differ by approximately 1.3 g C day$^{-1}$ between GRA and CRO, while CLM$_{grid}$, CLM5$_{PFT}$, and GLASS indicate similar

RSMDs for those PFTs. Especially CLM5$_{grid}$ shows a constant within-PFT variability of around 1.5 g C day$^{-1}$
independent of the PFT. Higher mean GPP RMSD values also come with a higher standard deviation. These results
hint at a complex relationship of variability representation within the PFTs. The higher RMSE values of CLM5$_{PFT}$
in the general model performance analysis (Section 3.2) suggest that the variation across sites of one PFT seen here
does not directly translate to a better model performance. Apart from the magnitude of the variability, its accurate
and proportionate timing is pivotal for enhanced model performance.


## 4. Discussion

Our results show that $CLM5_{grid}$ and $CLM5_{PFT}$ approximate the ET observations from ICOS better than GLASS remote sensing and ERA5L reanalysis but worse than GLEAM reanalysis. Moreover, especially for $CLM5_{PFT}$ the systematic error in simulating ET is lower than all other evaluated data sets. For GPP, we found that $CLM5_{grid}$ and 530 $CLM5_{PFT}$ performed worse than GLASS data, indicated by a larger PBIAS and larger RMSE. Surprisingly, $CLM5_{PFT}$ generally had a higher RMSE than $CLM_{grid}$ but, at the same time, a lower PBIAS. Averaged ET and GPP phenologies were relatively well simulated but exhibited underestimations across all PFT, especially in DBF, compared to ICOS measurements. $CLM5_{PFT}$ better captured the PFT-specific mean and standard deviation of the ET and GPP annual dynamics than $CLM5_{grid}$, the reanalyses, and remote sensing data. The GPP and ET 535 distributions analysis showed underestimations of their observed variability for all models, $CLM5_{grid}$, $CLM5_{PFT}$, GLASS, ERA5L, and GLEAM. Lastly, we found that for most PFTs, the modeled and remotely sensed data was too similar between stations of the same PFT group compared to the ICOS observations.

### 4.1. Uncertainty

### 4.1.1. Observations

Notably, the EC measurements carry uncertainties that might affect the results of this study, especially related to the systematic errors in the simulations. For instance, EC measurements neglect the energy from large eddies. To check for possible inconsistencies, we evaluated the energy balance corrected ET ($ET_{corr}$) from the ICOS sites (Pastorello et al., 2020). This methodology assumes a constant Bowen ratio to close the energy imbalance. Simulated ET underestimates $ET_{corr}$ to a greater degree than the non-corrected ICOS ET (Figure S1, Figure S2), 545 suggesting a higher systematic error than in the analysis of non-corrected ET. Besides that, we discovered the same patterns with the corrected ET, concluding that the energy balance error did not introduce significant bias to our results and the interpretations. Furthermore, GPP is not directly measured but partitioned from NEE. The NEE partitioning method has an underlying uncertainty stemming from potentially unfulfilled assumptions that propagate to the GPP and ER variables in the ICOS data. So, we also ensured that our results remained consistent 550 by evaluating the non-partitioned NEE and the ER variables (Figure S3, Figure S4, Figure S5, Figure S6). We discovered a substantial underestimation and missing variability in NEE and ER across PFTs in CLM5, confirming the systematic underestimation in our analysis of GPP. While we believe that our analyses have followed meticulous approaches to ensure robust results by applying the ICOS quality flags and comparing these additional

variables, many studies still emphasized the biases arising from a shifting footprint with varying wind direction and wind speed and the energy balance correction method assuming a constant Bowen ratio (Jung et al., 2020; Eshonkulov et al., 2019; Chu et al., 2021). Therefore, we encourage developing and using novel and more accurate energy balance closure methods (Zhang et al., 2024). Furthermore, dropping bad-quality gap-fill data from the ET and GPP time series might introduce a bias that underrepresents periods of low friction velocity and atmospheric inversion conditions. Lastly, based on the geographical distribution of the ICOS station network, the results might misrepresent Southern and Eastern Europe and semi-arid and arid hydro-climates (Figure 1, also read Ohnemus et al. 2024). Those factors might have influenced the diversity of ET and GPP values and the ranges of their distributions.

### 4.1.2. Forcing

Importantly, discrepancies between the COSMO Reanalysis used to force the European CLM5, and the station observations might introduce deviation into our analyses that could hamper interpretations of our results regarding the model functionality. While the high-resolution forcing data already includes information from observations through data assimilation, particular locations, and conditions might be less well represented than others, and a resulting bias in the meteorological variables would propagate to the simulation of ET and GPP. However, data assimilation approaches minimize the systematic error of the atmospheric model to the observations. Furthermore, the probability and potential influence of including a bias from the forcing of a single location is lowered by considering multiple sites in the performance and statistics of the PFTs. Nevertheless, we assessed the meteorological variables from the COSMO Reanalysis 6 (temperature, shortwave incoming radiation, precipitation, relative humidity) with the ICOS station data to scrutinize potential errors arising from the forcing. We used the same approach for the GPP and ET evaluation (Figure S7 – Figure S14). We discovered that the forcing variables' average yearly dynamics and distributions represent the ICOS observations well. More minor yet notable misrepresentations include underestimations of shortwave downward radiation and precipitation in summer and relative humidity over GRA and CRO sites throughout the year compared to the measurements. This could explain some of our analyses' ET and GPP underestimations by CLM5. Notably, the mean and variance across the PFTs and their ranking are represented reasonably well for all forcing variables, as opposed to our results with GPP and ET. Furthermore, the skewness and excess kurtosis of the forcing temperature and shortwave downward radiation compare well to the ones from ICOS, indicating well-matching distributions between the COSMO Reanalysis 6 and the observations. However, in particular, the higher-degree moments of the distribution are not well simulated

for precipitation and relative humidity. These characteristics of the distributions affect the CLM5 simulations of GPP and ET and might have influenced our results. Further considerations, including ensemble simulations with perturbed forcings, are required to capture the uncertainty introduced into CLM5 fully, but this is beyond the scope of this study.

### 4.1.3. Static information and initial conditions

The static surface information, including the soil texture, elevation, aspect, land unit, and PFT distributions, affect the simulation of ET and GPP in CLM5. The soil texture composition will define how water is stored and conducted in the soil, contributing to the evaporation from the soil, an essential ET component. Further, the soil texture will influence root water uptake if vegetation is present in the soil column, indirectly impacting plants' transpiration, another critical ET component. Further, ET is regulated by the available energy, which is determined by how the canopy, the elevation, and the aspect of that location influence the incoming radiation. Especially the diversity between these input variables across the locations of the ICOS stations might have played an essential role in the simulation of the PFT-specific ET and GPP distributions.

Lastly, particularly for $CLM5_{grid}$, GLASS, GLEAM, and ERA5L, the distribution of PFTs across the domain and in the grid cells corresponding to the ICOS stations define the equations and parameters that will be used for the calculation of ET and GPP. Consequently, if the grid cells corresponding to ICOS stations are dominated by PFTs that do not comply with the stations' footprints, the simulations of specific PFTs in the model are negatively affected. Importantly, this does not apply to the $CLM5_{PFT}$ because we could select the data that belongs to the adequate PFT. Therefore, interpretations of our results relating directly to vegetation functions implemented in CLM5 are here primarily focusing on the $CLM5_{PFT}$ data.

The initial conditions of the carbon cycle, most notably the size of the soil and vegetation carbon pools, are another source of uncertainty. Essentially, our spin-up and production simulations were restricted to the years where the high-resolution forcing was available (1995 – 2018). The spin-up simulations, therefore, recycle atmospheric forcings for a substantial period, which we also use in the production simulations. Hence, the production simulations adopted the equilibrium state (incoming carbon equals outgoing carbon) required to conclude the spin-up. However, in natural conditions, there was no carbon equilibrium in the simulated years. Instead, the carbon cycle experiences dynamic changes, such as long-term trends resulting from changing environmental conditions. Many European ecosystems exhibited a net carbon uptake, thus acting as a carbon sink (Pilli et al., 2017; Winkler

et al., 2023), measured in ICOS accordingly. The negative long-term mean NEE indicates carbon sources, evident across all PFTs in the EC observations (Figure S4 a). On the other hand, the simulations show a NEE close to zero for all PFTs, directly showing the effect of the equilibrium state of the land surface in the model. The results of DBF, which is the most significant carbon sink in the ICOS data and simultaneously shows the largest GPP underestimations by CLM5, underline a potentially important role of the carbon equilibrium in our results. Future work will conduct a more comprehensive spin-up under conditions closer to a real-world carbon equilibrium (the 1950s or earlier) and a transition run before the production simulations to capture the dynamic trends of the land surface processes. Possibly, the bias in the EC measurements towards conditions with low friction velocity and atmospheric inversion might also cause overestimations of GPP and the resulting carbon sink in ICOS.

## 4.2. PFT-specific evaluation

While CLM5$_{PFT}$ showed a smaller systematic error than CLM$_{grid}$ for most PFT compared to the observations (lower absolute PBIAS), the ability to approximate the observation time series is worse (higher RMSE). A shifting sign in the bias of the CLM5$_{PFT}$ data explains these counterintuitive results. The presence of both positive and negative bias (in time and across stations) cancels out and yields an overall low PBIAS. In summary, we find in the evaluation that the ET time series of CLM$_{PFT}$ are not closer to observations than CLM5$_{grid}$ for any PFT, but CLM5$_{PFT}$ generally approximates the ET sum over time better than CLM5$_{grid}$ for ENF, DBF, and CRO. However, it is also clear that, on average, the phenology of CLM5$_{PFT}$ is closer to the observed than CLM5$_{grid}$, for instance, for both ET and GPP at DBF and GRA sites. Furthermore, the timings of the phenological events in CLM5$_{PFT}$ are most often closer to the observed than in CLM5$_{grid}$. Importantly, critical PFT-specific characteristics, like the timing of DBF's steep spring GPP increase, are only captured by CLM5$_{PFT}$ and the inter-site variability of ET and GPP throughout a standard year. This discrepancy between the evaluation metrics and the vegetation phenology suggests that CLM5$_{PFT}$ could better capture the PFT-specific variability that ICOS observes. However, this variability is modeled in a way that did not contribute to a low RMSE, for instance, shifted in time or space, so the averaged PFT-specific comparisons (the phenology and the distribution moments) compare better with ICOS than CLM5$_{grid}$. Further evidence for this explanation is that CLM5$_{PFT}$ generally captures more variability (higher ET and GPP standard deviation across sites throughout the year for ENF, GRA, and CRO, and higher variance for each PFT). This ability to capture more variability than the other models, closer to the observed variability, can improve the represented variability in CLM5$_{PFT}$ if the suitable variation can be modeled at the right time and location. This spatiotemporal discrepancy of simulated and observed GPP and ET variability could potentially be solved with optimized PFT

parameters (Baker et al., 2022; Birch et al., 2021; Cheng et al., 2021; Dagon et al., 2020; Deng et al., 2021; Fisher et al., 2019b).

Several past studies also indicated the underestimation of ET and GPP in CLM5 compared to observations (Boas et al., 2023; Strebel et al., 2023; Cheng et al., 2021; Birch et al., 2021), which we confirm in this study. Parameter improvements could also alleviate these general underestimations of GPP and ET across PFTs, especially during

summer (Dagon et al., 2020). However, optimal parameters might vary from site to site (Lin et al., 2015) even if they have the same PFT. Thus, CLM5, and more generally, LSMs that implement plant traits as parameters on the PFT level, cannot capture this intrinsic PFT variability resulting from these traits. Albeit optimized parameters might still reduce the bias on the continental level, a more comprehensive approach to the spatiotemporal variability of plant traits might improve regional simulations drastically (Anderegg et al., 2022; Van Bodegom et al., 2014;

Kattge et al., 2011).

Given the hydraulic role of vegetation leaves in controlling transpiration, there is a tight relationship between ET, GPP, and LAI. In CLM5-BGC, the assimilated carbon by GPP gets further partitioned to respiration and the carbon storage in the plant organs, i.e., leaves, roots, and stems. Furthermore, the leaf carbon then controls the development and state of the vegetation leaves and, thus, the LAI. On the other hand, LAI controls GPP by determining the

upscaling factors from leaf photosynthesis to the canopy, thereby driving canopy conductance. Unfortunately, no large-scale LAI in-situ measurements and no CLM5$_{PFT}$ simulated LAI are available, and comparisons between CLM5$_{grid}$ LAI and reanalysis or remote sensing LAI suffer from known biases in the latter and yield no further context on our evaluation based on ground truth information. We adhered to an LAI evaluation of CLM5 with sparse but systematic ICOS measurements, ERA5L reanalysis, and GLASS based on MODIS (Supplementary

Figure S15). Notably, the ICOS LAI measurements are only available for two years of the study period (2017 to 2018) and are limited to ENF and CRO sites. Additionally, LAI measurements' expensive and time-intensive nature restricts the time resolution to a few yearly measurement points. As a result, the data points for comparison are few, and the uncertainties are larger (noticeable larger error bars in Supplementary Figure S15). Another caveat is the potential mismatch of the land surface representation between the EC tower footprint (ET and GPP

measurements) and the area covered by the LAI measurement campaigns. However, some key findings from this analysis are still robust. For example, all models overestimate LAI at ENF and CRO sites (Supplementary Figure 14 a), contrasting the results of GPP and ET. The variance in ENF sites is much more significant in GLASS and ERA5L than in CLM5$_{grid}$, which is closest to the observations. The higher-order moments are more uncertain

because of the small number of data points. The contrasting results, especially between the LAI and GPP PFT-level

averages, suggest that processes and parameters connecting the assimilated carbon to the leaf area, depending on the environmental conditions, must be revisited. However, we make an even stronger case for systematic, long-term, and high-resolution LAI in-situ measurements (for example, using drones (Bates et al. 2021)), which would support a more robust and diverse evaluation of the simulations of this essential variable.

### 4.3. Inter-site similarity of PFT groups

For all models (CLM5$_{grid}$, CLM5$_{PFT}$, ERA5L, GLASS, GLEAM), the distributions of ET and GPP across PFTs are very similar, which is not the case for the observations. This is especially true for their variances (i.e., their spread around the mean) but also notable for the means, skewnesses, and kurtoses. We expected CLM5$_{PFT}$ to show more significant variability than CLM5$_{grid}$ and the other grid-scale models because the aggregated, mixed PFT data of the grid cell would homogenize the variables and cancel out some of the variability. While CLM5$_{PFT}$ shows a more

extensive range of variation of ET and GPP across PFTs than CLM5$_{grid}$, ERA5L, GLASS, and GLEAM, it still vastly underestimates the observed range of variance by ICOS, especially for GPP (Figure 9, Figure 11).

The mean RMSD across sites of the same PFT indicates that ET across sites can be as different in CLM$_{PFT}$ for GRA and CRO as in the observations (Figure 12). However, the ET differences across sites with the same PFT were underestimated at ENF and DBF. GPP differences across sites with the same PFTs were underestimated for all

PFTs (Figure 13). This suggests the missed variance could mainly stem from missed PFT internal inter-site differences or unresolved differences in site-specific abiotic conditions (e.g., soil depth and texture). Possibly, this could not be improved through optimization of PFT-specific parameters, as these sites would still share the same set of parameters. An enhanced concept of functional types in vegetation, focusing on the spatiotemporal variability of observed plant traits, could better facilitate improvements that raise the simulated ET and GPP variance in space

and time.

### 4.4. Data requirements

As outlined above, beyond parameter optimizations, a comprehensive implementation of functional ecosystem diversity could significantly improve the LSM simulation outputs regarding multiple aspects of their distributions. This could introduce a state-of-the-art understanding of vegetation function into LSMs, which is essential to

evaluate different theories of plan trait evolution and their effect on current and future energy, water, and carbon cycles.

In that light, we encourage sites to co-locate research infrastructures (Futter et al., 2023), like ICOS and the Integrated European Long-Term Ecosystem, critical zone, and socio-ecological Research Infrastructure (eLTER-RI). Thereby, sites cover additional observation spheres like biodiversity (e.g., functional diversity of plants) and socio-ecology (through forest and crop management and driving land use change) and establish a strong base for studies to increase the understanding of the whole system (Ohnemus et al., 2024; Mirtl et al., 2018; Mirtl et al., 2021; Baatz et al., 2018). Further, this would promote large-scale observations needed to introduce more trait variability into LSMs. Lastly, combining LSMs and these holistic observations by data assimilation, going beyond decoupled modeling efforts (Bloom et al., 2020) and resulting in an ecosystem reanalysis (Baatz et al., 2021), would provide essential, explicit and accurate data on the carbon cycle, which are currently unavailable.

## 4.5 Distribution moments and droughts

Investigating the influence of drought on the analyses, or generally the ability of the models to simulate drought and the vegetation response, is complex due to the differences in drought response functionality. For instance, plant water stress might occur due to different magnitudes of water deficit in the soil, on different aggregation time scales, and with a variable lag to the water deficit. A future study will investigate the PFT-scale drought responses from the model and how the drought propagates through the eco-hydrological sphere and compare it to observations. However, drought frequency, duration, and severity affect the shapes of the distribution of the precipitation and, eventually, the ecosystem processes. Thus, we briefly discuss possible insights into their drought responses.

Importantly, the skewness and excess kurtosis moments, which inform about the characteristics of the distribution tails (relativity between the tails and the general tailedness, respectively) of precipitation (Guo, 2022), and vegetation states and function (Kanavi et al., 2020; Liu et al., 2022; Cooley et al., 2022), are influenced by dry conditions, depending on their frequency, duration and severity. We found a low variability in the skewness and excess kurtosis of the precipitation used to force our CLM5 simulations (Figure S10 c and d), specifically a significantly lower skewness and excess kurtosis at ENF and DBF sites. A lower positive skewness than the observations means that the distribution is less skewed towards lower values, and a lower positive excess kurtosis than the observations indicates generally larger tails. A possible interpretation of these differences in the

distribution moments is that the atmospheric forcings show more frequent, longer, and more severe extreme precipitation events, while the ICOS measurements are more concentrated around their mean. While the propagation of these extreme events could be complex and non-linear, we generally found the same results (lower skewness and smaller absolute excess kurtosis) for the simulated distributions of ET and GPP for almost all models and PFTs (Figures 9 c and d and 11 c and d), suggesting a more direct relationship. However, because of the possible non-linearity and the influence of other factors, the detailed relationship between these findings and the ability of CLM5 to simulate ecosystem drought responses must be examined in future studies. In any case, the missing accuracy in representing higher distribution moments in the atmospheric forcings and in land surface models must be considered in studies using these to investigate drought.

## 5. Conclusions

We evaluated the simulated evapotranspiration (ET) and gross primary production (GPP) from a 3 km resolved Community Land Model v5 (CLM5) set up over the European CORDEX domain. We differentiated the model outputs between the grid scale (CLM5$_{grid}$) and the plant functional type scale (CLM5$_{PFT}$) and compared them with ICOS station data as ground truth data. Furthermore, we compared with ET and GPP from remote sensing derived data from the Global Land Surface Satellite (GLASS) and reanalysis products such as the European Centre for Medium-Range Weather Forecast Reanalysis 5 - Land (ERA5L) and the Global Land Evaporation Amsterdam Model (GLEAM). CLM5$_{grid}$ and CLM5$_{PFT}$ exhibited promising skills in approximating the observations and often performed better than ERA5L, GLASS, and GLEAM. CLM5$_{PFT}$ showed a lower systematic bias (lower percent bias) but approximated the ICOS observations generally worse (larger root mean square error) than CLM5$_{grid}$ (Figures 2 and 3 and Supplementary Tables S4 and S5). ET and GPP were systematically underestimated across all PFTs throughout the year for both model scales. Especially during summer at DBF sites, GPP was substantially lower for CLM5$_{PFT}$ and CLM5$_{grid}$ than for ICOS observations (Figure 4, Figure 6).

Essentially, CLM5$_{PFT}$ and, to a greater degree, CLM5$_{grid}$, ERA5L, GLEAM, and GLASS showed a lower spatiotemporal variability of ET and GPP than the measurements exhibited by a lower range of all the modeled ET and GPP distribution moments across PFTs than in ICOS. This smaller range and a lower root mean square difference between sites of one PFT group suggests that CLM5$_{grid}$, and more surprisingly, CLM5$_{PFT}$, simulate GPP and ET more similarly across PFTs than the ICOS measurements.

Future studies should investigate whether optimizing parameters in CLM5$_{PFT}$ with observation data increases the diversity of ET and GPP values or whether this is a structurally induced bias. This work provides essential insights for studies that aim to find optimized parameters and meaningful context for analyses of more specific ET and GPP dynamics using the evaluated data.

**Code availability**

A frozen version of the CLM5 version used here is stored here: https://doi.org/10.5281/zenodo.11091890. The case setup for the European 3 km simulation and a post-processing script are available under https://doi.org/10.5281/zenodo.11091845. Analysis, processing, and plotting scripts are available at https://doi.org/10.5281/zenodo.13885473, which requires the helper scripts in this additional repository: https://doi.org/10.5281/zenodo.13885466.

**Data availability**

We used publicly available data, namely the Warm-Winter-2020 data set from the Integrated Carbon Observation System (ICOS, https://www.icos-cp.eu/data-products/2G60-ZHAK, and https://meta.icos-cp.eu/collections/nBLHm8lrY2FHpiybxuS3po2B), the ERA5-Land reanalysis (https://cds.climate.copernicus.eu/cdsapp#!/dataset/reanalysis-era5-land), Global Land Surface Satellite (GLASS)

data derived from remote sensing (http://www.glass.umd.edu/index.html) and reanalysis data from the Global Evaporation Amsterdam Model (GLEAM, https://www.gleam.eu/). Intermediary tabular data in parquet format corresponding to the location of the ICOS stations are stored in https://doi.org/10.5281/zenodo.11091898 for each data source used here, including CLM5grid and CLM5PFT. The raw CLM5 outputs over the whole European domain, which were not used in this study, can be made available upon request (approx. eight terabytes).

**Author contribution**

C.P.T., B.S.N., and H.J.H.F. conceived and designed the study. C.P.T. processed the data and performed the analyses. B.S.N., H.J.H.F., R.B., and H.V. suggested the analyses and helped interpret the results. C.P.T. wrote the manuscript and edited the suggestions from all co-authors.

**Competing interests**

The authors declare that no competing interests are present.

**Acknowledgments**

The authors gratefully acknowledge computing time on the supercomputer JURECA (Jülich Supercomputing Centre, 2021) at Forschungszentrum Jülich under grant number jibg31. The authors also thank the editor and anonymous reviewers for their constructive feedback during the review process.

This research has been supported by the European Union's Horizon 2020 research and innovation program under grant agreement number 871128 (eLTER PLUS), as well as from Deutsche Forschungsgemeinschaft (DFG, German Research Foundation) – SFB 1502/1–2022 – project number 450058266 (DETECT).

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

**Table S1: A list of ICOS stations, their land cover, coordinates, years of data availability for our study period (1995 – 2018), the coordinates of the corresponding grid cell of the 3 km European Coordinated Regional Climate Downscaling Experiment (CORDEX) grid used in our**
**simulations, and the number of 8-daily data points available for the analyses for evapotranspiration (ET) and gross primary production (GPP). Note that stations that do not belong to the plant functional types (PFT) of evergreen needleleaf forest (ENF), deciduous broadleaf forest (DBF), grasslands (GRA), and croplands (CRO) were omitted, and some included sites did not have data corresponding with the study period, thus having a count of 0**
**data points. See Section 2.2.1. The indicated PFT is the predominant PFT in the footprint of the ICOS eddy covariance towers. Stations, where the land cover was not directly indicated in the metadata sites were also left out in our analyses.**

| ID | country | PFT | lat | lon | years | lat (cell) | lon (cell) | N (ET) | N (GPP) |
|---|---|---|---|---|---|---|---|---|---|
| **BE-Bra** | Belgium | ENF | 51.31 | 4.52 | 1996 – 2018 | 51.29 | 4.51 | 608 | 670 |
| **BE-Dor** | Belgium | GRA | 50.31 | 4.97 | 2011 – 2018 | 50.31 | 4.96 | 0 | 270 |
| **BE-Lcr** | Belgium | DBF | 51.11 | 3.85 | | 51.10 | 3.85 | 0 | 0 |
| **BE-Lon** | Belgium | CRO | 50.55 | 4.75 | 2004 – 2018 | 50.57 | 4.76 | 519 | 476 |
| **CH-Cha** | Switzerland | GRA | 47.21 | 8.41 | 2005 – 2018 | 47.21 | 8.43 | 423 | 459 |
| **CH-Dav** | Switzerland | ENF | 46.82 | 9.86 | 1997 – 2018 | 46.80 | 9.84 | 578 | 866 |
| **CH-Fru** | Switzerland | GRA | 47.12 | 8.54 | 2005 – 2018 | 47.11 | 8.53 | 284 | 447 |
| **CH-Oe2** | Switzerland | CRO | 47.29 | 7.73 | 2004 – 2018 | 47.28 | 7.72 | 0 | 592 |

| CZ-BK1 | Czech Republic | ENF | 49.50 | 18.54 | 2004 – 2018 | 49.50 | 18.54 | 146 | 389 |
|--------|---------|-----|-------|-------|-------------|-------|-------|-----|-----|
| CZ-Lnz | Czech Republic | DBF | 48.68 | 16.95 | 2015 – 2018 | 48.67 | 16.95 | 0 | 145 |
| DE-Geb | Germany | CRO | 51.10 | 10.91 | 2001 – 2020 | 51.10 | 10.93 | 824 | 638 |
| DE-Gri | Germany | GRA | 50.95 | 13.51 | 2001 – 2018 | 50.95 | 13.49 | 673 | 492 |
| DE-Hai | Germany | DBF | 51.08 | 10.45 | 2000 – 2018 | 51.07 | 10.45 | 813 | 548 |
| DE-HoH | Germany | DBF | 52.09 | 11.22 | 2015 – 2018 | 52.09 | 11.23 | 184 | 113 |
| DE-Kli | Germany | CRO | 50.89 | 13.52 | 2004 – 2018 | 50.90 | 13.54 | 481 | 450 |
| DE-RuR | Germany | GRA | 50.62 | 6.30 | 2011 – 2018 | 50.62 | 6.28 | 336 | 309 |
| DE-RuS | Germany | CRO | 50.87 | 6.45 | 2011 – 2018 | 50.86 | 6.44 | 285 | 224 |
| DE-RuW | Germany | ENF | 50.50 | 6.33 | 2012 – 2018 | 50.51 | 6.31 | 0 | 125 |
| DE-Tha | Germany | ENF | 50.96 | 13.57 | 1996 – 2018 | 50.96 | 13.58 | 1012 | 888 |
| DK-Gds | Denmark | ENF | 56.07 | 9.33 |  | 56.07 | 9.34 | 0 | 0 |
| DK-Sor | Denmark | DBF | 55.49 | 11.64 | 1996 – 2018 | 55.48 | 11.65 | 437 | 882 |

| | | | | | | | | |
|---|---|---|---|---|---|---|---|---|
| **FI-Hyy** | Finland | ENF | 61.85 | 24.29 | 1996 – 2018 | 61.86 | 24.29 | 435 | 812 |
| **FI-Ken** | Finland | ENF | 67.99 | 24.24 | 2018 | 67.99 | 24.23 | 0 | 18 |
| **FI-Let** | Finland | ENF | 60.64 | 23.96 | 2009 – 2018 | 60.63 | 23.96 | 412 | 254 |
| **FI-Var** | Finland | ENF | 67.75 | 29.61 | 2016 – 2018 | 67.76 | 29.63 | 135 | 133 |
| **FR-Aur** | France | CRO | 43.55 | 1.11 | 2005 – 2018 | 43.54 | 1.12 | 470 | 483 |
| **FR-Bil** | France | ENF | 44.49 | -0.96 | 2014 – 2020 | 44.50 | -0.98 | 203 | 144 |
| **FR-FBn** | France | ENF | 43.24 | 5.68 | 2008 – 2018 | 43.25 | 5.69 | 0 | 358 |
| **FR-Fon** | France | DBF | 48.48 | 2.78 | 2005 – 2018 | 48.47 | 2.80 | 0 | 566 |
| **FR-Gri** | France | CRO | 48.84 | 1.95 | 2004 – 2018 | 48.86 | 1.95 | 563 | 313 |
| **FR-Hes** | France | DBF | 48.67 | 7.06 | 2014 – 2018 | 48.67 | 7.05 | 229 | 219 |
| **FR-Lam** | France | CRO | 43.50 | 1.24 | 2005 – 2018 | 43.51 | 1.25 | 548 | 431 |
| **FR-Tou** | France | GRA | 43.57 | 1.37 | 2018 | 43.58 | 1.38 | 46 | 28 |
| **IT-BFt** | Italy | DBF | 45.20 | 10.74 | | 45.21 | 10.75 | 0 | 0 |

| IT-MBo | Italy | GRA | 46.01 | 11.05 | 2003 – 2018 | 46.00 | 11.04 | 616 | 582 |
|--------|-------|-----|-------|-------|-------------|-------|-------|-----|-----|
| IT-Ren | Italy | ENF | 46.59 | 11.43 | 1999 – 2018 | 46.58 | 11.44 | 531 | 525 |
| IT-SR2 | Italy | ENF | 43.73 | 10.29 | 2013 – 2018 | 43.74 | 10.31 | 255 | 214 |
| IT-Tor | Italy | GRA | 45.84 | 7.58 | 2008 – 2018 | 45.85 | 7.57 | 481 | 251 |
| RU-Fy2 | Russia | ENF | 56.45 | 32.90 | 2015 – 2018 | 56.46 | 32.89 | 156 | 138 |
| SE-Htm | Sweden | ENF | 56.10 | 13.42 | 2015 – 2018 | 56.10 | 13.42 | 177 | 152 |
| SE-Nor | Sweden | ENF | 60.09 | 17.48 | 2014 – 2018 | 60.09 | 17.50 | 229 | 181 |
| SE-Svb | Sweden | ENF | 64.26 | 19.77 | 2014 – 2018 | 64.26 | 19.77 | 161 | 109 |

**Table S2: The root mean square error (RMSE) and percent bias (PBIAS) for model evapotranspiration (ET) in relation to the Integrated Carbon Observation System (ICOS) observations. Stations from ICOS that did not belong to plant functional types (PFTs) of evergreen needleleaf forest (ENF), broadleaf deciduous forest (DBF), croplands (CRO), or grasslands (GRA) or did not have overlapping periods were omitted. See Section 2.4.2. For the**
**amount of data points per station used for the calculations, see Table S1.**

| | ET RMSE [mm day$^{-1}$] | | | | | ET PBIAS [%] | | | | |
|--------|------------------------|----------|-------|-------|-------|--------------|----------|-------|-------|-------|
| | CLM5$_{grid}$ | CLM5$_{PFT}$ | ERA5L | GLASS | GLEAM | CLM5$_{grid}$ | CLM5$_{PFT}$ | ERA5L | GLASS | GLEAM |
| BE-Bra | 0.54 | 0.51 | 1.12 | 1.1 | 0.65 | 20.53 | 22.4 | 103.3 | 86.1 | 53.95 |
| BE-Lon | 0.67 | 0.99 | 0.82 | 0.91 | 0.49 | 12.76 | 24.31 | 66.69 | 43.88 | 19.71 |
| CH-Cha | 0.8 | 0.85 | 0.59 | 0.54 | 0.56 | -33.03 | -21.19 | -13.73 | -10.68 | -8.47 |

| | | | | | | | | | |
|---|---|---|---|---|---|---|---|---|---|
| **CH-Dav** | 1.2 | 0.95 | 0.91 | 1.35 | 0.85 | -51.08 | -33.29 | -54.41 | -32.38 | -27.66 |
| **CH-Fru** | 0.62 | 0.85 | 0.52 | 0.62 | 0.62 | -23.73 | -8.69 | -6.68 | -5.21 | 7.17 |
| **CZ-BK1** | 0.48 | 0.54 | 0.76 | 0.57 | 0.52 | -23.06 | -26.04 | 29.54 | 19.72 | 25.78 |
| **DE-Geb** | 0.51 | 0.82 | 0.7 | 0.85 | 0.48 | -7.61 | -5.35 | 64.26 | 40.08 | 14.93 |
| **DE-Gri** | 0.48 | 0.77 | 0.57 | 0.55 | 0.36 | 2.45 | 11.15 | 33.24 | 20.49 | 9.14 |
| **DE-Hai** | 0.49 | 0.6 | 0.73 | 0.76 | 0.52 | 2.64 | 8.99 | 58.52 | 46.6 | 31.18 |
| **DE-HoH** | 0.69 | 0.65 | 0.6 | 0.58 | 0.66 | -28.06 | -16.86 | -1.62 | -10.44 | -24.37 |
| **DE-Kli** | 0.69 | 1 | 0.79 | 0.74 | 0.63 | 6.77 | 19.04 | 38.9 | 27.7 | 21.78 |
| **DE-RuR** | 0.39 | 0.76 | 0.6 | 0.54 | 0.45 | -17.86 | 5.37 | 28.01 | 9.89 | 17.22 |
| **DE-RuS** | 0.78 | 0.97 | 0.68 | 0.55 | 0.68 | -32.8 | -31.45 | 7.9 | -12.81 | -24.98 |
| **DE-Tha** | 0.62 | 0.5 | 0.72 | 0.71 | 0.48 | 0.59 | -0.52 | 39.68 | 20.84 | 13.92 |
| **DK-Sor** | 0.6 | 0.6 | 0.57 | 0.66 | 0.5 | -26.29 | -14.98 | 42.64 | 20.57 | 2.18 |
| **FI-Hyy** | 0.5 | 0.51 | 0.49 | 0.41 | 0.62 | -35.58 | -27.65 | 20.64 | 11.27 | 41.7 |
| **FI-Let** | 0.68 | 0.65 | 0.63 | 0.8 | 0.73 | -31.77 | -21.53 | 51.02 | 11.16 | 40.21 |
| **FI-Var** | 0.37 | 0.49 | 0.73 | 0.48 | 0.6 | -30.13 | -9.59 | 67.09 | 58.22 | 84.39 |
| **FR-Aur** | 0.85 | 1.19 | 1.1 | 1.05 | 0.78 | 5.44 | 45.08 | 52.04 | 37.1 | 16.89 |
| **FR-Bil** | 0.67 | 0.92 | 1.46 | 0.72 | 0.67 | -25.5 | -28.35 | 24.98 | 48.24 | 24.47 |
| **FR-Gri** | 0.77 | 1.01 | 0.9 | 0.85 | 0.58 | -1.63 | 0.98 | 44.94 | 30.06 | 3.86 |
| **FR-Hes** | 0.58 | 0.67 | 0.83 | 0.86 | 0.72 | 0.19 | 13.09 | 51.71 | 35.65 | 36.79 |
| **FR-Lam** | 0.86 | 1.09 | 0.97 | 1.01 | 0.79 | -6.76 | 20.9 | 31.79 | 17.15 | -1.53 |
| **FR-Tou** | 0.69 | 0.89 | 0.86 | 1.04 | 0.49 | -36.01 | -45.95 | 60.87 | 30.99 | 17.48 |
| **IT-MBo** | 0.55 | 0.84 | 0.5 | 0.49 | 0.72 | -2.29 | -17.01 | 8.99 | 6.24 | 16.68 |
| **IT-Ren** | 0.85 | 0.81 | 0.74 | 0.72 | 0.76 | -23.81 | -3.55 | -9.57 | -15.41 | 2.18 |
| **IT-SR2** | 0.89 | 1.53 | 0.73 | 0.76 | 0.8 | -34.1 | -60.81 | 28.98 | 3.25 | -23.83 |

| | | | | | | | | | |
|---|---|---|---|---|---|---|---|---|---|
| **IT-Tor** | 0.91 | 1.01 | 0.6 | 0.78 | 0.75 | -45.19 | -48.2 | -38.59 | -10.22 | -28.59 |
| **RU-Fy2** | 0.4 | 0.51 | 0.65 | 0.69 | 0.7 | -4.43 | -16.31 | 52.09 | 26.21 | 54.79 |
| **SE-Htm** | 0.45 | 0.45 | 1.19 | 0.88 | 0.9 | -7.31 | -3.36 | 72.78 | 61.52 | 79.05 |
| **SE-Nor** | 0.36 | 0.37 | 0.66 | 0.58 | 0.59 | -14.29 | -4.12 | 47.2 | 22.25 | 46.44 |
| **SE-Svb** | 0.45 | 0.64 | 0.55 | 0.35 | 0.56 | -18.82 | -0.66 | 16.38 | 16.8 | 35.55 |

*Table S3: The root mean square error (RMSE) and percent bias (PBIAS) for model gross primary production (GPP) in relation to the Integrated Carbon Observation System (ICOS) observations. Stations from ICOS that did not belong to the plant functional types (PFTs) of evergreen needleleaf forest (ENF), deciduous broadleaf forest (DBF), croplands (CRO), or grasslands (GRA) or did not have overlapping periods were omitted. See Section 2.4.2. For the amount of data points per station used for the calculations, see Table S1.*

| | GPP RMSE [g C day$^{-1}$] | | | GPP PBIAS [%] | | |
|---|---|---|---|---|---|---|
| | CLM5$_{grid}$ | CLM5$_{PFT}$ | GLASS | CLM5$_{grid}$ | CLM5$_{PFT}$ | GLASS |
| BE-Bra | 2.29 | 1.69 | 1.3 | -35.36 | 0.58 | 4.7 |
| BE-Dor | 3.19 | 3.39 | 2.74 | -41.69 | -40.3 | -35.11 |
| BE-Lon | 4.31 | 4.31 | 3.98 | -18.21 | -8.23 | -11.32 |
| CH-Cha | 4.61 | 3.94 | 4.29 | -50.9 | -38.52 | -47.17 |
| CH-Dav | 2.4 | 2.13 | 2.13 | -16.93 | 31.37 | -25.57 |
| CH-Fru | 3.6 | 2.84 | 2.62 | -40.1 | -23.16 | -23.97 |
| CH-Oe2 | 3.75 | 3.95 | 3.53 | -10.8 | -12.63 | 2.72 |
| CZ-BK1 | 2.79 | 2.31 | 1.95 | -37.05 | -22.83 | -20.65 |
| CZ-Lnz | 4.64 | 3.44 | 2.9 | -62.06 | -49.31 | -28.91 |
| DE-Geb | 3.63 | 4.32 | 2.98 | -35.96 | -40.43 | -1.84 |
| DE-Gri | 2.61 | 2.68 | 2.02 | -21.19 | -11.94 | -9.65 |
| DE-Hai | 2.83 | 2.59 | 1.7 | -34.83 | -42.5 | -1.51 |
| DE-HoH | 2.94 | 2.51 | 3.04 | -30.53 | -40.55 | -27.82 |
| DE-Kli | 3.5 | 3.66 | 3.15 | 1.74 | 2.04 | -2.73 |
| DE-RuR | 2.4 | 2.39 | 2 | -26.99 | -10.45 | -19.5 |
| DE-RuS | 4.74 | 5.05 | 4.34 | -43.49 | -45.67 | -34.68 |
| DE-RuW | 2.63 | 2.61 | 2.14 | -32.13 | -27.64 | -23.88 |
| DE-Tha | 1.87 | 1.48 | 1.29 | -28.99 | -3.95 | -19.27 |

| | | | | | |
|---|---|---|---|---|---|
| **DK-Sor** | 4.39 | 4.07 | 3.21 | -47.99 | -49.66 | -35.24 |
| **FI-Hyy** | 1.3 | 1.29 | 0.81 | -14.92 | -0.32 | -8.91 |
| **FI-Ken** | 1.16 | 2.34 | 0.72 | -2.8 | 54.7 | -14.37 |
| **FI-Let** | 2.05 | 2.02 | 1.53 | -19.16 | -4.72 | -19.98 |
| **FI-Var** | 1.4 | 3.22 | 0.89 | 60.3 | 159.3 | 21.48 |
| **FR-Aur** | 3.28 | 4.05 | 3.25 | 9.39 | 68.57 | 9.03 |
| **FR-Bil** | 1.75 | 2.23 | 1.67 | -24.81 | -24.43 | -0.67 |
| **FR-FBn** | 2.38 | 3.73 | 1.82 | -48.88 | -77.01 | 15.32 |
| **FR-Fon** | 3.1 | 2.87 | 2.74 | -27 | -36.28 | -21.96 |
| **FR-Gri** | 4.16 | 4.24 | 3.73 | -18.71 | -13.87 | -15.53 |
| **FR-Hes** | 3.7 | 3.24 | 3.32 | -24.49 | -36.28 | -17.36 |
| **FR-Lam** | 3.91 | 4.5 | 3.95 | -4.09 | 44.8 | -8.88 |
| **FR-Tou** | 3.44 | 2.53 | 1.77 | -73.37 | -47.07 | -10.42 |
| **IT-MBo** | 2.42 | 2.89 | 1.84 | -7.9 | -31.89 | 3.26 |
| **IT-Ren** | 1.53 | 2.32 | 1.77 | 11.62 | 33.32 | -2.04 |
| **IT-SR2** | 5.12 | 6.78 | 4.07 | -67.17 | -88.85 | -53.94 |
| **IT-Tor** | 1.82 | 2.49 | 1.66 | -0.74 | 1.02 | 1.17 |
| **RU-Fy2** | 2.63 | 2.84 | 1.93 | -26.11 | -22.3 | -23.45 |
| **SE-Htm** | 2.74 | 2.24 | 1.95 | -38.04 | -25.42 | -26.84 |
| **SE-Nor** | 1.59 | 1.37 | 1.35 | -25.59 | -3.8 | -21.62 |
| **SE-Svb** | 1.13 | 2.02 | 1.22 | 5.64 | 25.07 | -24.24 |

*Table S4: The evapotranspiration (ET) root mean square error (RMSE) indicates the general model approximations and the percent bias (PBIAS), demonstrating systematic bias of the models (Community Land Model v5 (CLM5) on grid-scale (CLM5$_{grid}$), CLM5 on PFT scale (CLM5$_{PFT}$), from the European Center of Medium-Range Weather Forecasts Renalysis 5 Land (ERA5-Land), the Global Land Surface Satellite (GLASS), and the Global Land Evaporation Amsterdam Model (GLEAM)) to the observations. Each value corresponds to a group of stations representing the same plant functional type (PFT; Evergreen Needleleaf Forest (ENF), Deciduous Broadleaf Forest (DBF), Grasslands (GRA), and Croplands (CRO)). The amount of data points (N) for each PFT is also indicated.*

| | PFT | N | CLM5$_{grid}$ | CLM5$_{PFT}$ | ERA5L | GLASS | GLEAM |
|---|---|---|---|---|---|---|---|
| **RMSE [mm day$^{-1}$]** | **ENF** | 5038 | 0.71 | 0.72 | 0.84 | 0.83 | 0.67 |
| | **DBF** | 1663 | 0.56 | 0.62 | 0.73 | 0.70 | 0.56 |
| | **GRA** | 2859 | 0.65 | 0.85 | 0.60 | 0.57 | 0.59 |
| | **CRO** | 3690 | 0.72 | 1.00 | 0.88 | 0.86 | 0.63 |
| | **mean** | 3285 | 0.66 | 0.80 | 0.76 | 0.74 | 0.61 |
| **PBIAS [%]** | **ENF** | 5038 | -20.57 | -15.42 | 21.86 | 13.32 | 15.43 |
| | **DBF** | 1663 | -9.90 | -0.54 | 44.55 | 29.74 | 16.24 |
| | **GRA** | 2859 | -18.62 | -13.94 | 3.14 | 2.63 | 2.41 |
| | **CRO** | 3690 | -3.24 | 11.20 | 44.99 | 27.30 | 7.58 |
| | **mean** | 3285 | -13.08 | -4.68 | 28.64 | 18.25 | 10.42 |

*Table S5: The gross primary production (GPP) root mean square error (RMSE) indicates the general model approximation and the percent bias (PBIAS), demonstrating systematic bias of the models (Community Land Model v5 (CLM5) on grid-scale (CLM5$_{grid}$), CLM5 on PFT scale (CLM5$_{PFT}$), from the European Center of Medium-Range Weather Forecasts Renalysis 5 Land (ERA5-Land), the Global Land Surface Satellite (GLASS), and the Global Land Evaporation Amsterdam Model (GLEAM)) to the observations. Each value corresponds to a group of stations representing the same plant functional type (PFT: Evergreen Needleleaf Forest (ENF), Deciduous Broadleaf Forest (DBF), Grasslands (GRA), and Croplands (CRO)). The amount of data points (N) for each PFT is also indicated.*

|  | PFT | N | CLM5$_{grid}$ | CLM5$_{PFT}$ | GLASS |
|---|---|---|---|---|---|
| RMSE [g C day$^{-1}$] | ENF | 5976 | 2.25 | 2.44 | 1.75 |
|  | DBF | 2473 | 3.71 | 3.35 | 2.81 |
|  | GRA | 2838 | 3.14 | 3.01 | 2.63 |
|  | CRO | 3607 | 3.85 | 4.21 | 3.55 |
|  | mean | 3723.5 | 3.24 | 3.25 | 2.69 |
| PBIAS [%] | ENF | 5976 | -26.00 | -7.7 | -14.53 |
|  | DBF | 2473 | -38.88 | -43.76 | -24.51 |
|  | GRA | 2838 | -30.73 | -25.5 | -21.34 |
|  | CRO | 3607 | -14.99 | -1.48 | -6.29 |
|  | mean | 3723.5 | -27.65 | -19.61 | -16.67 |

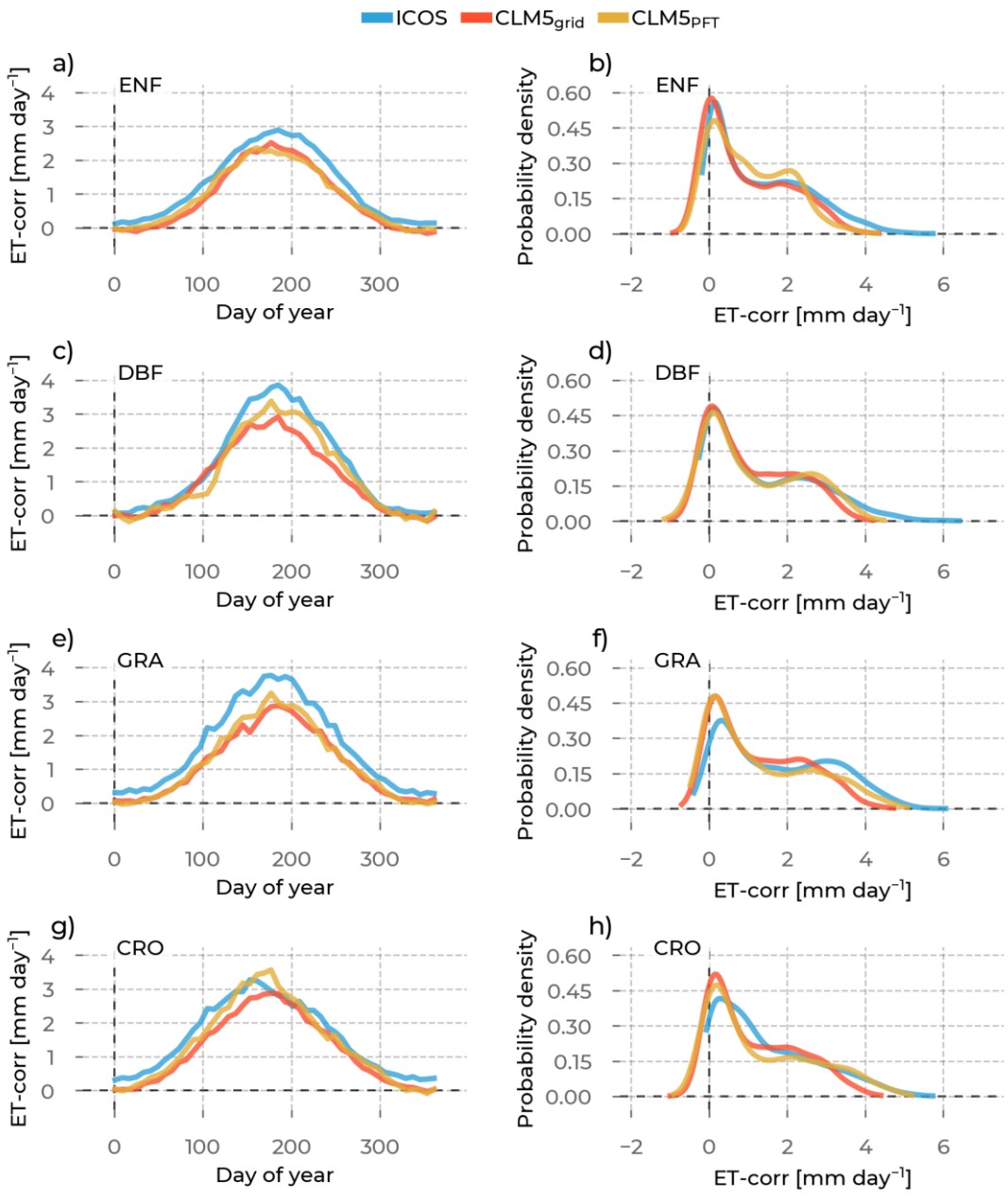

*Figure S1: In the left column are the yearly energy balance corrected evapotranspiration (ET-corr) evolutions averaged across stations belonging to one PFT (rows). We differentiate the data source by color (ICOS observations: blue, CLM5$_{grid}$: red, CLM5$_{PFT}$: yellow, GLASS: green, ERA5L: brown, GLEAM: purple). The*

*probability density curves for all ET-corr values from stations belonging to the selected PFT are in the right column. Each row shows these plots for one PFT: Evergreen Needleleaf Forest (ENF), Deciduous Broadleaf Forest (DBF), Grasslands (GRA), and Croplands (CRO).*

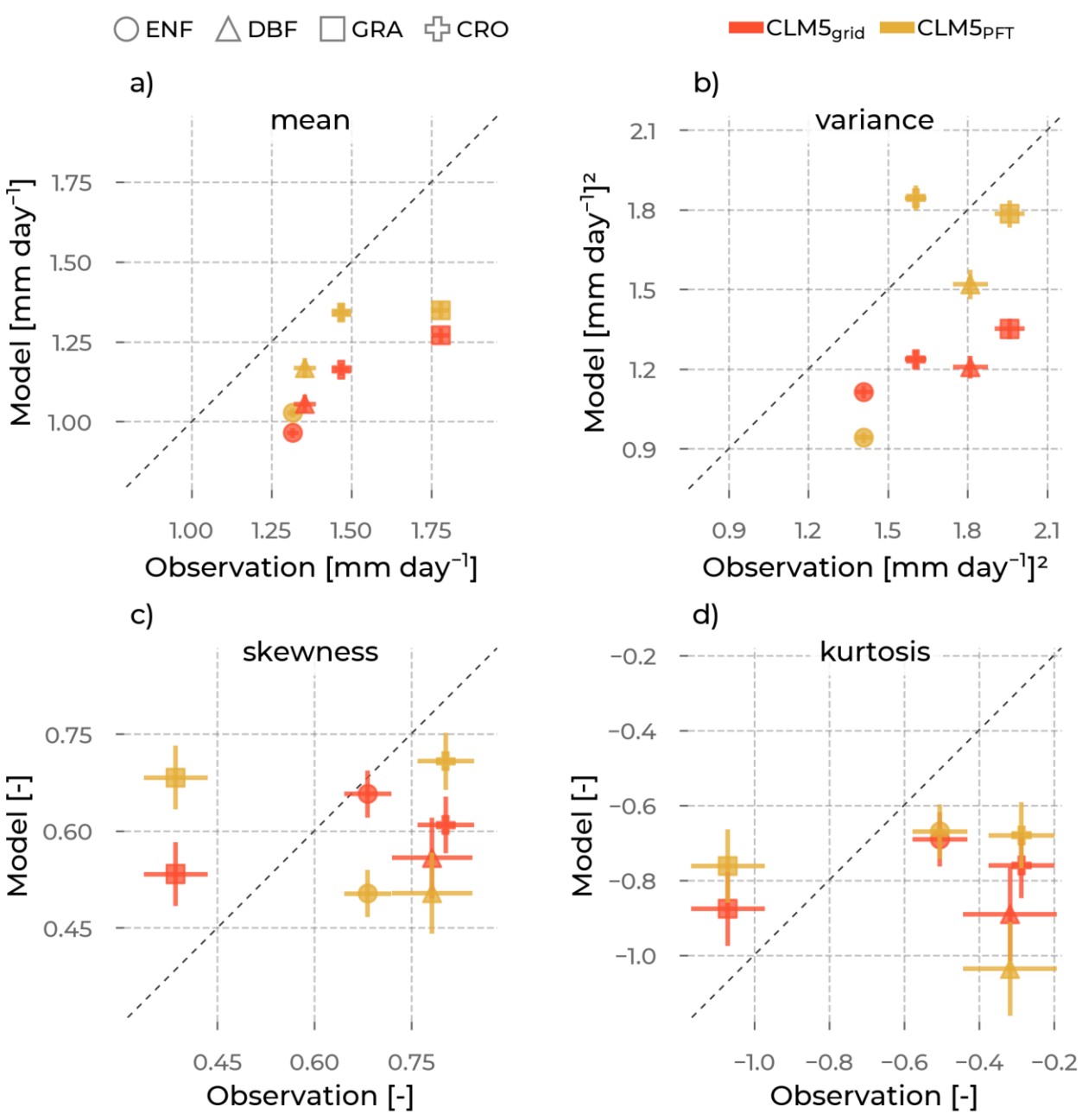

*Figure S2: The mean (a), variance (b), skewness (c), and excess kurtosis (d) of the ET-corr distributions (visualized in Figure S1) from the models (color, y-axis), as opposed to the corresponding values from observations (x-axis) aggregated for each PFT (marker type): Evergreen Needleleaf Forest (ENF), Deciduous Broadleaf Forest (DBF),*

*Grasslands (GRA), Croplands (CRO). The error bars are the standard errors of the respective moment, depending on the sample size.*

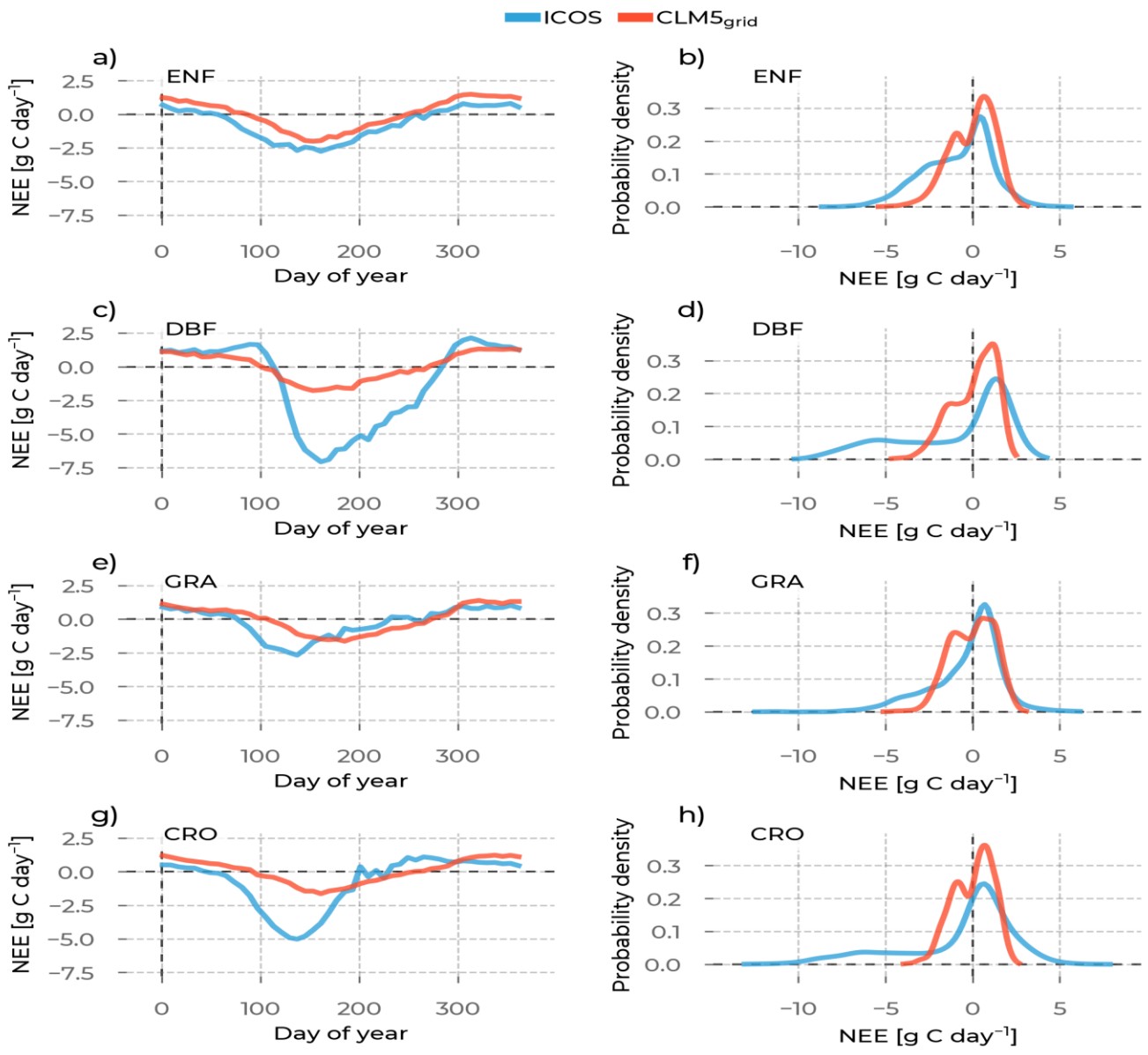

*Figure S3: In the left column are the yearly net ecosystem exchange (NEE) evolutions averaged across stations belonging to one PFT (rows). We differentiate the data source by color (ICOS observations: blue, CLM5$_{grid}$: red, CLM5$_{PFT}$: yellow, GLASS: green, ERA5L: brown, GLEAM: purple). The probability density curves for all NEE values from stations belonging to the selected PFT are in the right column. Each row shows these plots for one*

*PFT: Evergreen Needleleaf Forest (ENF), Deciduous Broadleaf Forest (DBF), Grasslands (GRA), and Croplands (CRO).*

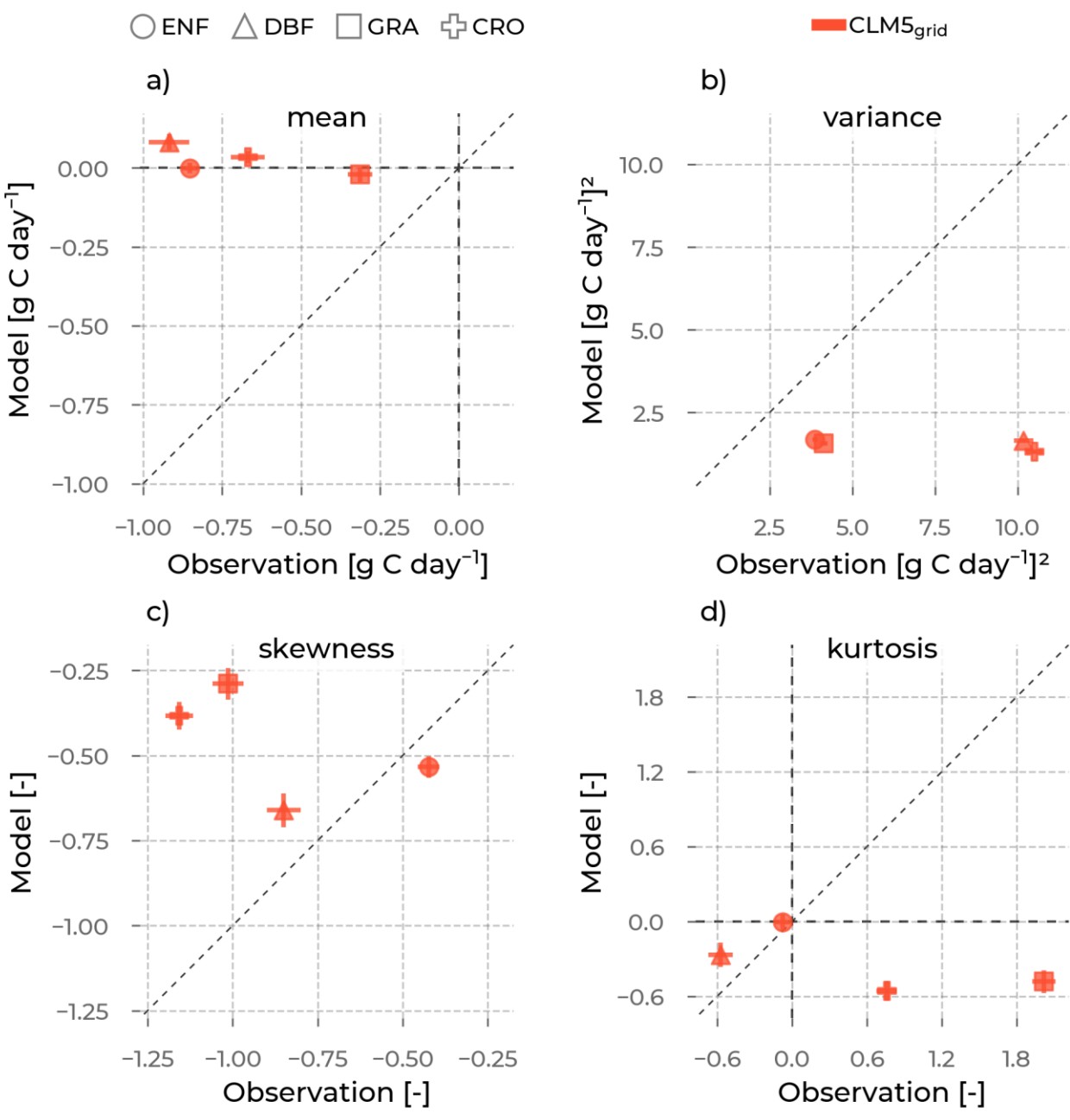

*Figure S4: The mean (a), variance (b), skewness (c), and excess kurtosis (d) of the NEE distributions (visualized in Figure S3) from the models (color, y-axis), as opposed to the corresponding values from observations (x-axis) aggregated for each PFT (marker type): Evergreen Needleleaf Forest (ENF), Deciduous Broadleaf Forest (DBF),*

*Grasslands (GRA), Croplands (CRO). The error bars are the standard errors of the respective moment, depending on the sample size.*

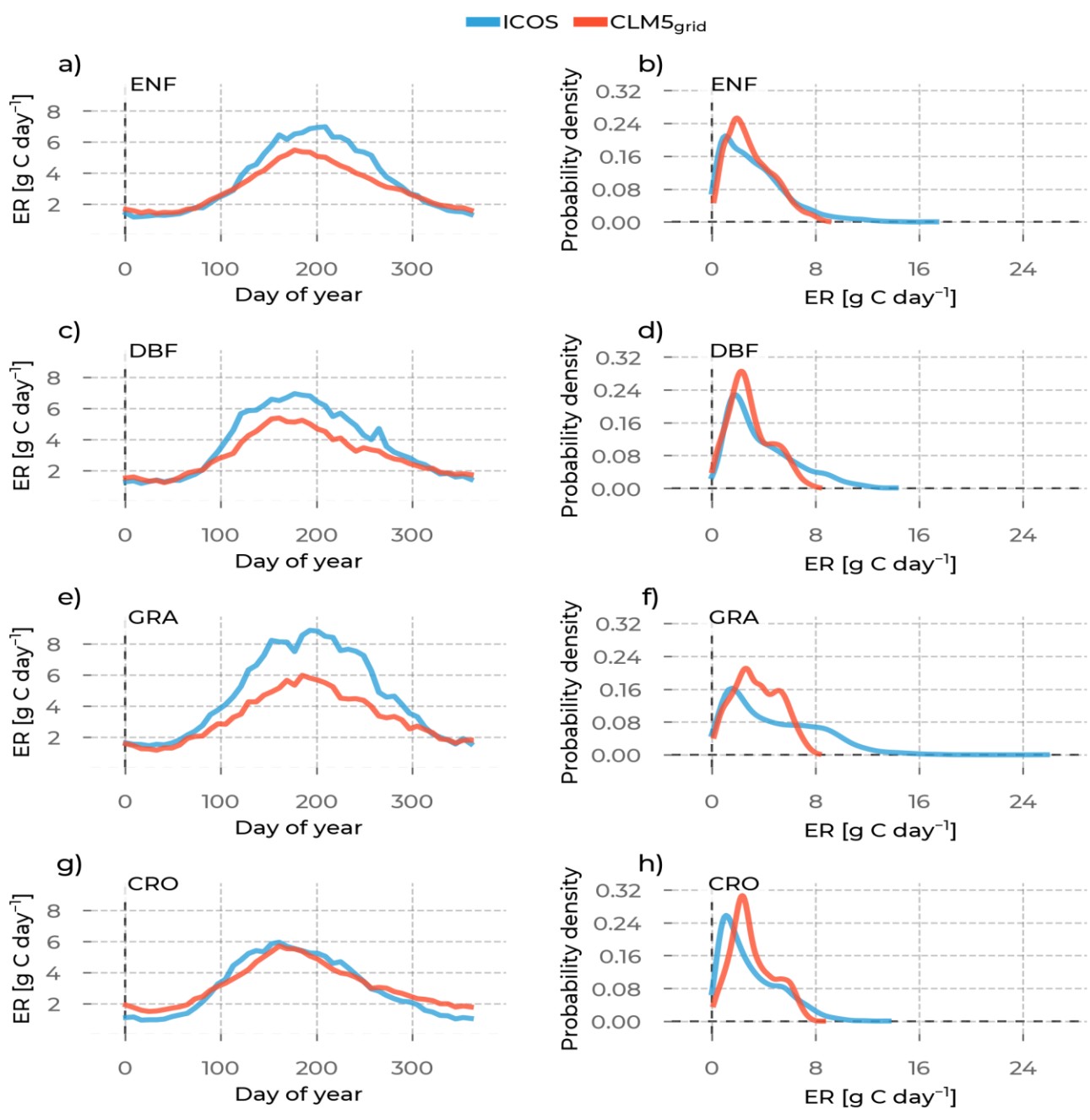

*Figure S5: In the left column are the yearly ecosystem respiration (ER) evolutions averaged across stations belonging to one PFT (rows). We differentiate the data source by color (ICOS observations: blue, CLM5grid: red, CLM5PFT: yellow, GLASS: green, ERA5L: brown, GLEAM: purple). The probability density curves for all ER values from stations belonging to the selected PFT are in the right column. Each row shows these plots for one*

*PFT: Evergreen Needleleaf Forest (ENF), Deciduous Broadleaf Forest (DBF), Grasslands (GRA), and Croplands (CRO).*

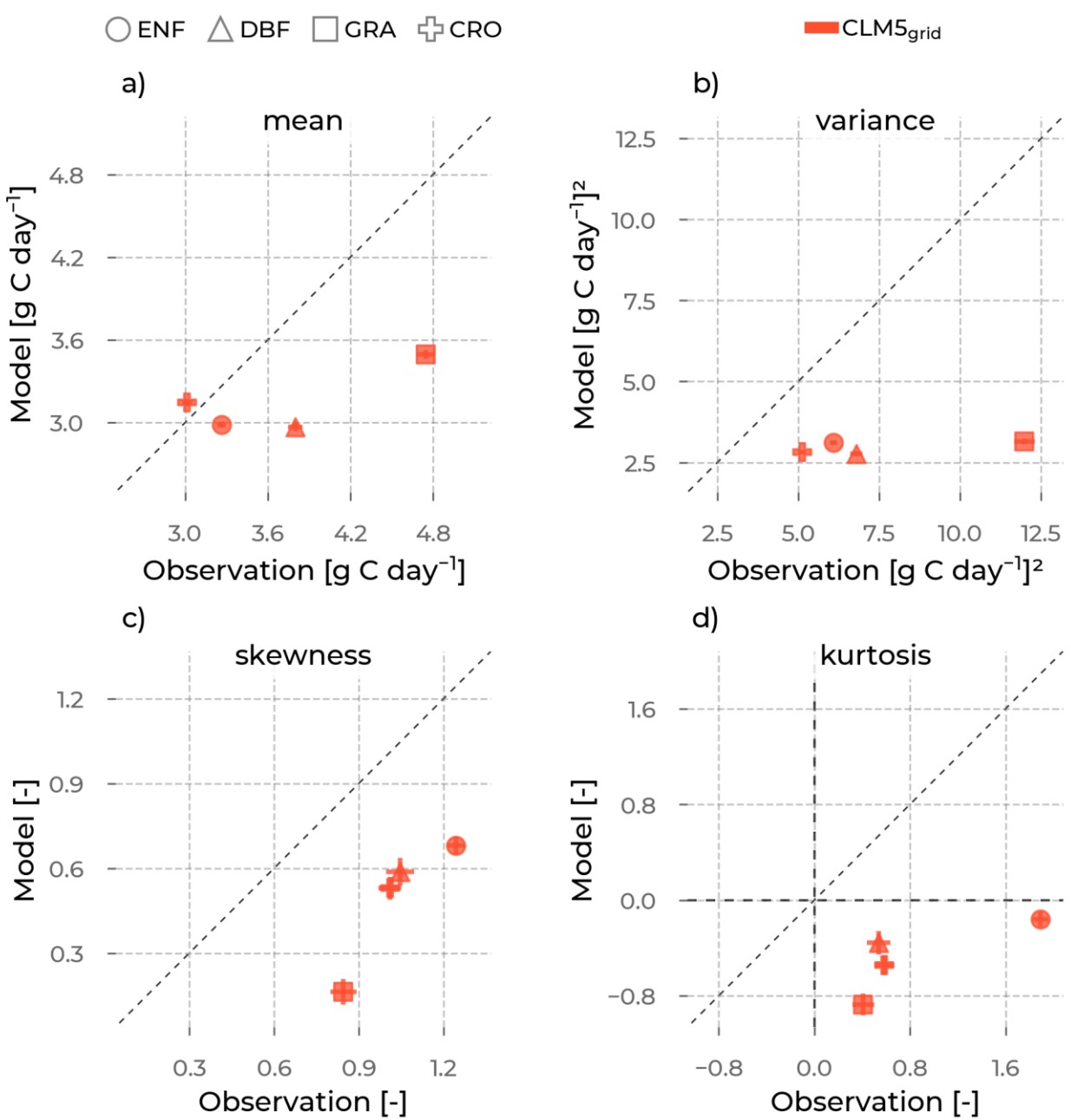

*Figure S6: The mean (a), variance (b), skewness (c), and excess kurtosis (d) of the ER distributions (visualized in Figure S5) from the models (color, y-axis), as opposed to the corresponding values from observations (x-axis) aggregated for each PFT (marker type): Evergreen Needleleaf Forest (ENF), Deciduous Broadleaf Forest (DBF),*

*Grasslands (GRA), Croplands (CRO). The error bars are the standard errors of the respective moment, depending on the sample size.*

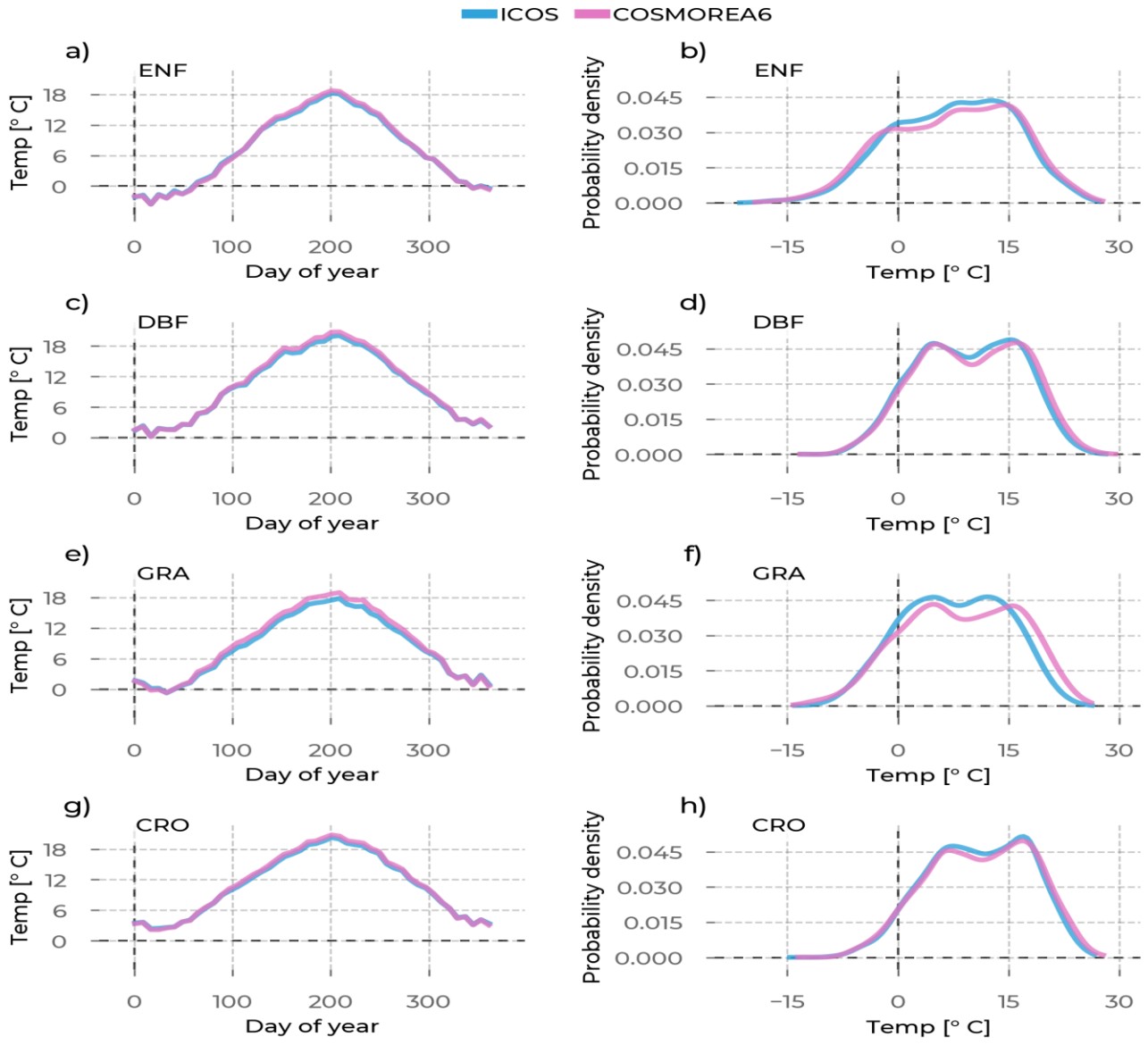

*Figure S7: In the left column are the yearly Temperature (Temp) evolutions averaged across stations belonging to one PFT (rows). We differentiate the data source by color (ICOS observations: blue, CLM5$_{grid}$: red, CLM5$_{PFT}$: yellow, GLASS: green, ERA5L: brown, GLEAM: purple). The probability density curves for all Temp values from stations belonging to the selected PFT are in the right column. Each row shows these plots for one PFT: Evergreen Needleleaf Forest (ENF), Deciduous Broadleaf Forest (DBF), Grasslands (GRA), and Croplands (CRO).*

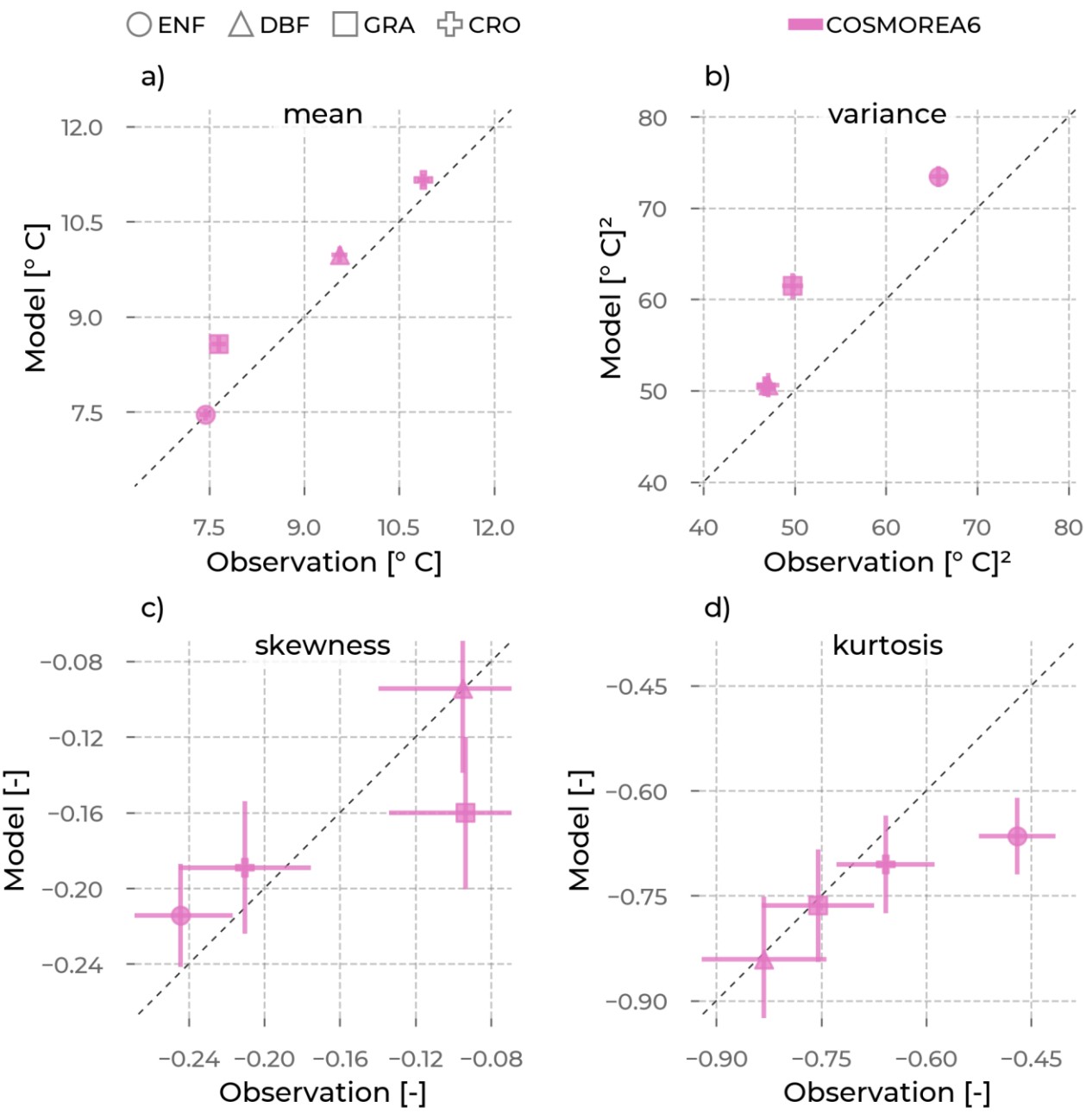

*Figure S8: The mean (a), variance (b), skewness (c), and excess kurtosis (d) of the Temp distributions (visualized in Figure S7) from the models (color, y-axis), as opposed to the corresponding values from observations (x-axis) aggregated for each PFT (marker type): Evergreen Needleleaf Forest (ENF), Deciduous Broadleaf Forest (DBF),*

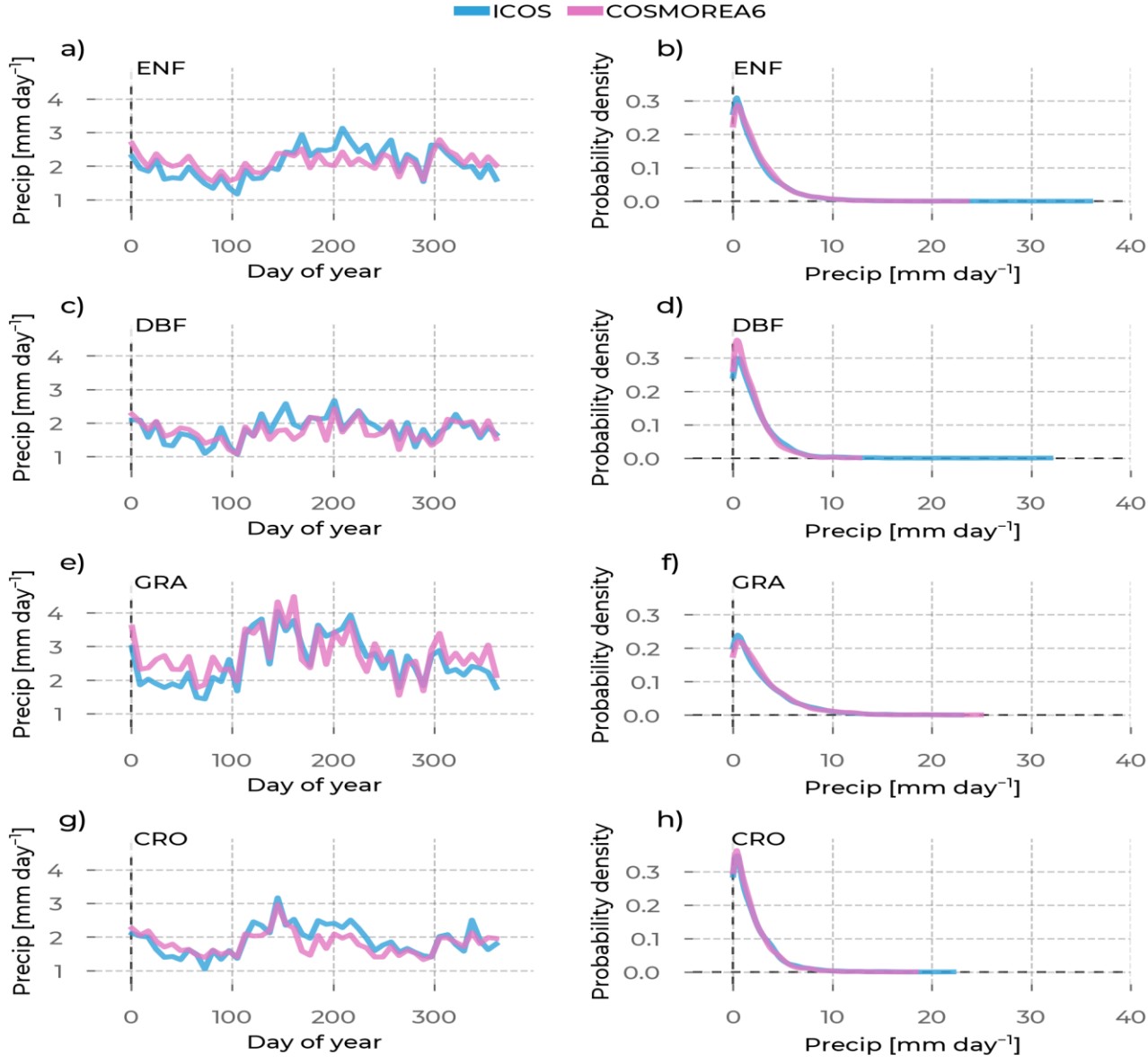

*Figure S9: In the left column are the yearly Precipitation (Precip) evolutions averaged across stations belonging to one PFT (rows). We differentiate the data source by color (ICOS observations: blue, CLM5$_{grid}$: red, CLM5$_{PFT}$: yellow, GLASS: green, ERA5L: brown, GLEAM: purple). The probability density curves for all Precip values from*

*stations belonging to the selected PFT are in the right column. Each row shows these plots for one PFT: Evergreen Needleleaf Forest (ENF), Deciduous Broadleaf Forest (DBF), Grasslands (GRA), and Croplands (CRO).*

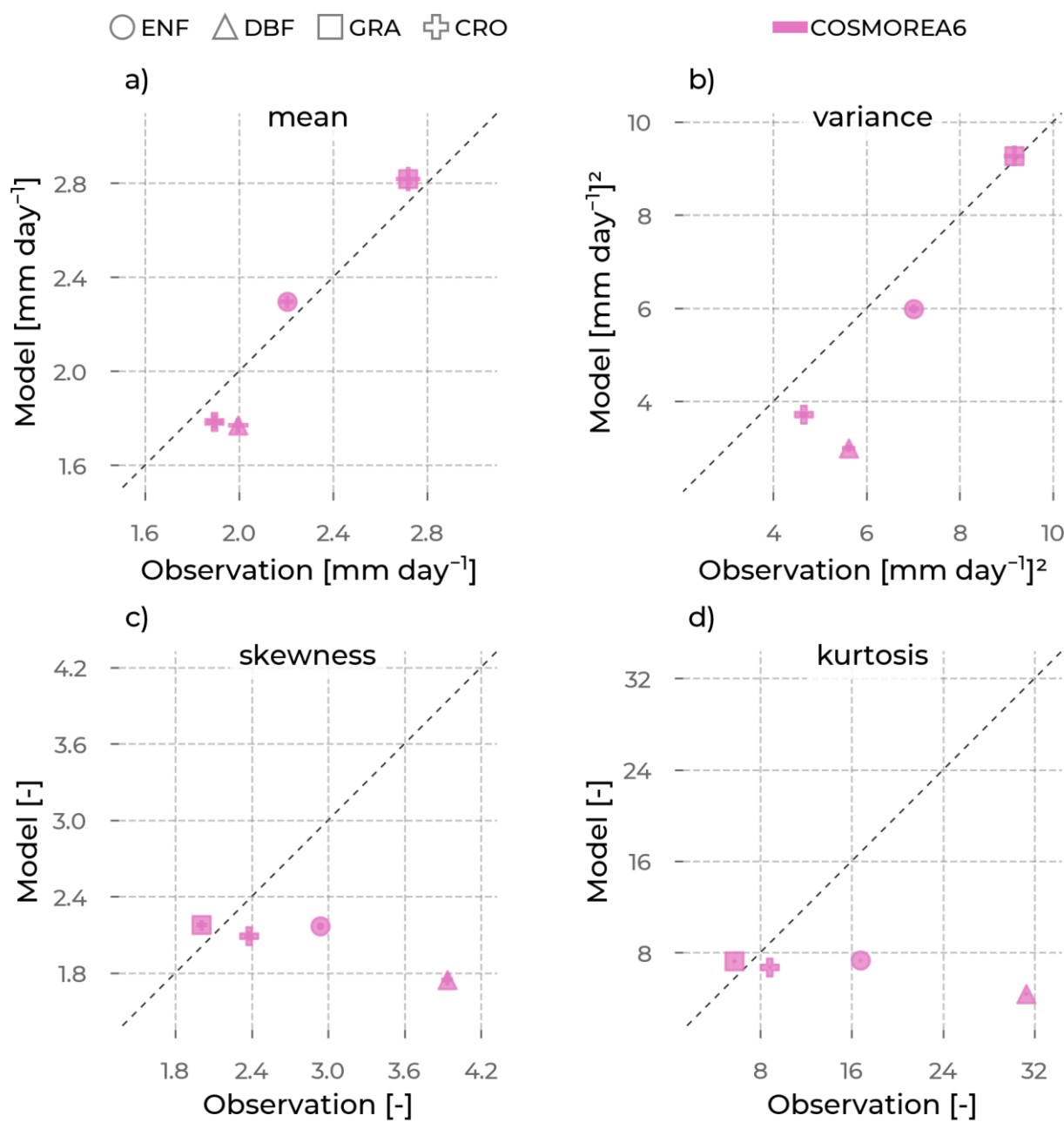


*Figure S10: The mean (a), variance (b), skewness (c), and excess kurtosis (d) of the Precip distributions (visualized in Figure S9) from the models (color, y-axis), as opposed to the corresponding values from observations (x-axis) aggregated for each PFT (marker type): Evergreen Needleleaf Forest (ENF), Deciduous Broadleaf Forest (DBF),*

*Grasslands (GRA), Croplands (CRO). The error bars are the standard errors of the respective moment, depending on the sample size.*

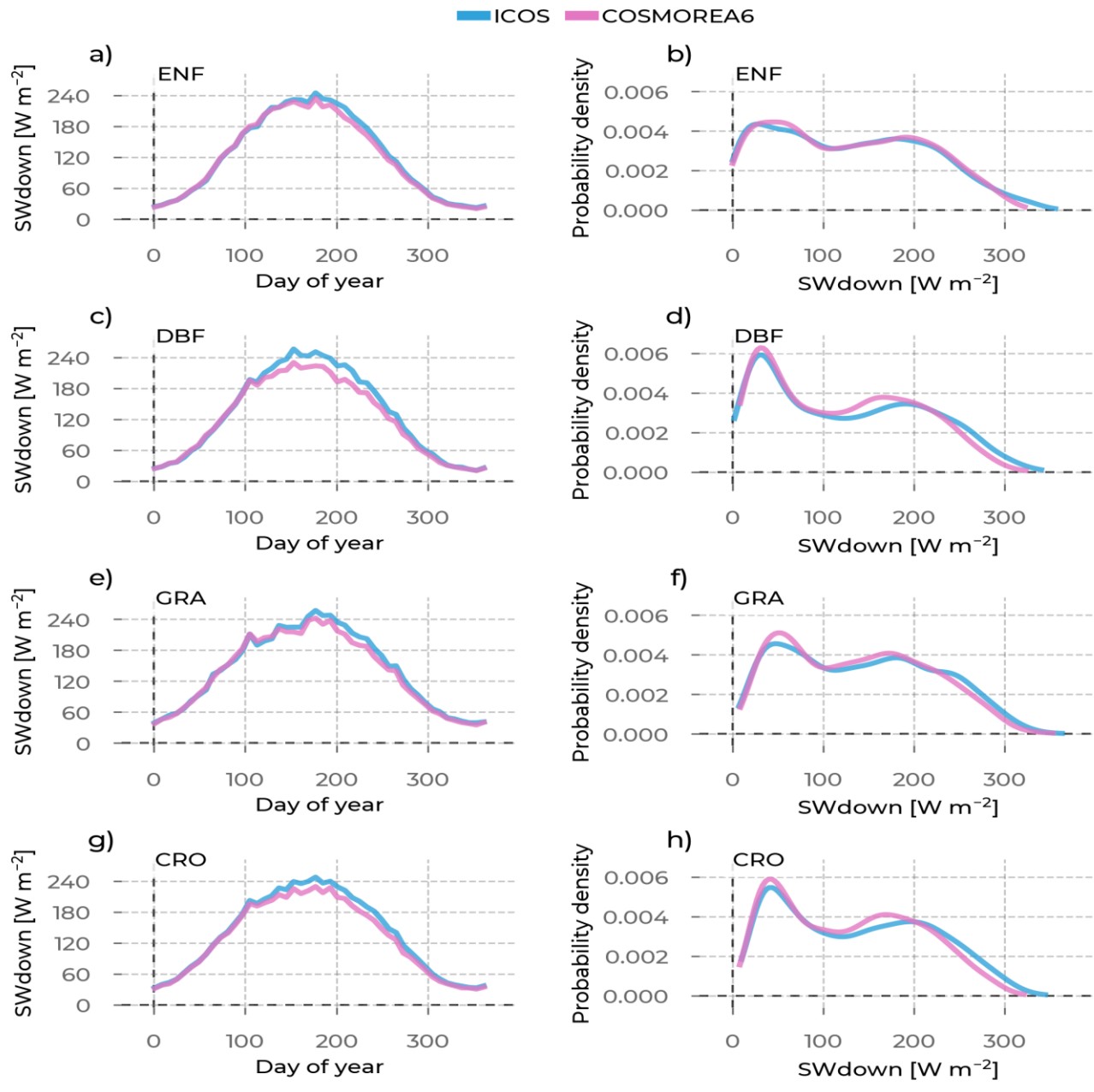

*Figure S11: In the left column are the yearly shortwave downward radiation (SWdown) evolutions averaged across stations belonging to one PFT (rows). We differentiate the data source by color (ICOS observations: blue,*

*CLM5$_{grid}$: red, CLM5$_{PFT}$: yellow, GLASS: green, ERA5L: brown, GLEAM: purple). The probability density curves for all SWdown values from stations belonging to the selected PFT are in the right column. Each row shows these plots for one PFT: Evergreen Needleleaf Forest (ENF), Deciduous Broadleaf Forest (DBF), Grasslands (GRA), and Croplands (CRO).*

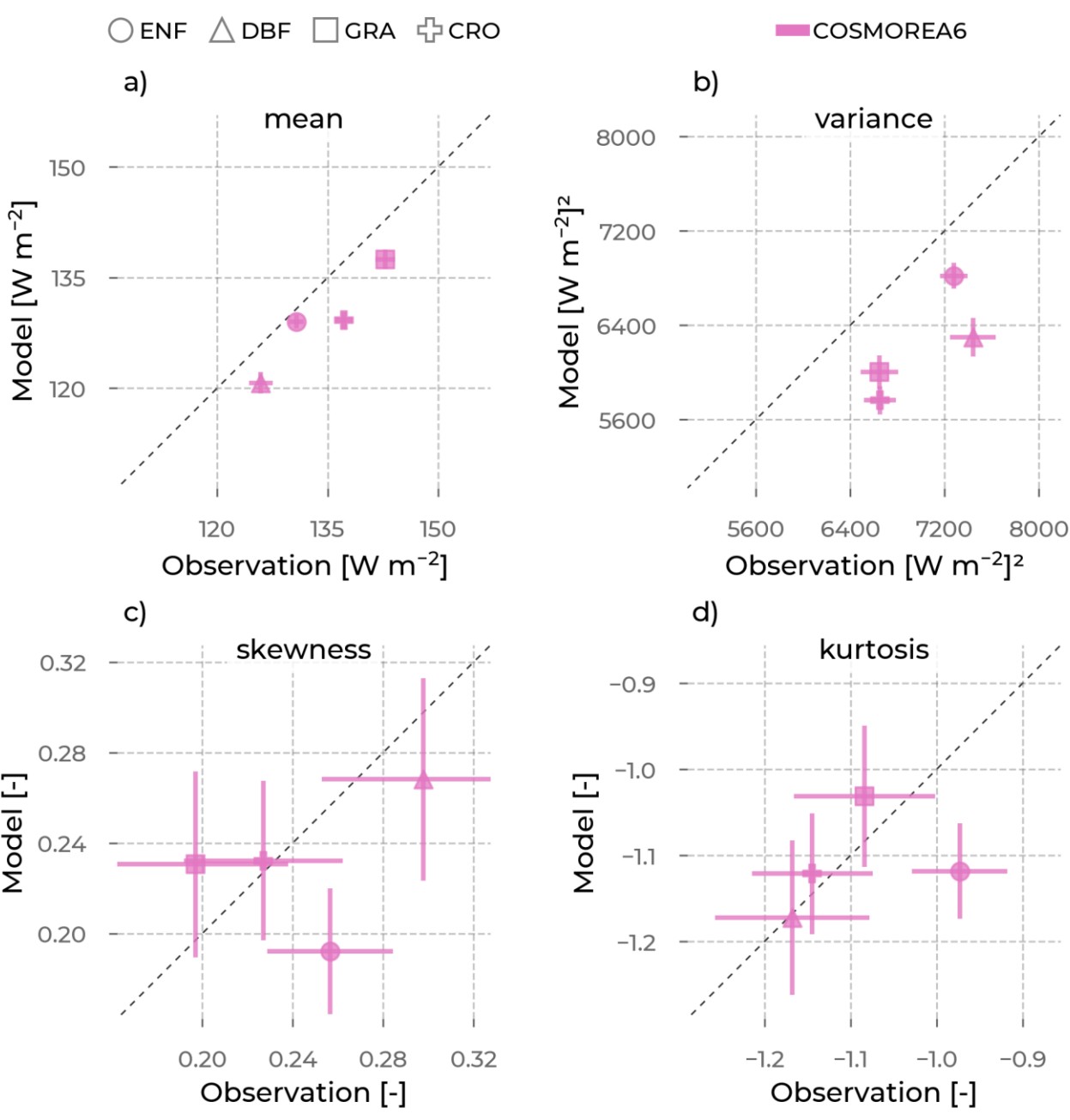

*Figure S12: The mean (a), variance (b), skewness (c), and excess kurtosis (d) of the SWdown distributions (visualized in Figure S11) from the models (color, y-axis), as opposed to the corresponding values from observations (x-axis) aggregated for each PFT (marker type): Evergreen Needleleaf Forest (ENF), Deciduous*

*Broadleaf Forest (DBF), Grasslands (GRA), Croplands (CRO). The error bars are the standard errors of the respective moment, depending on the sample size.*

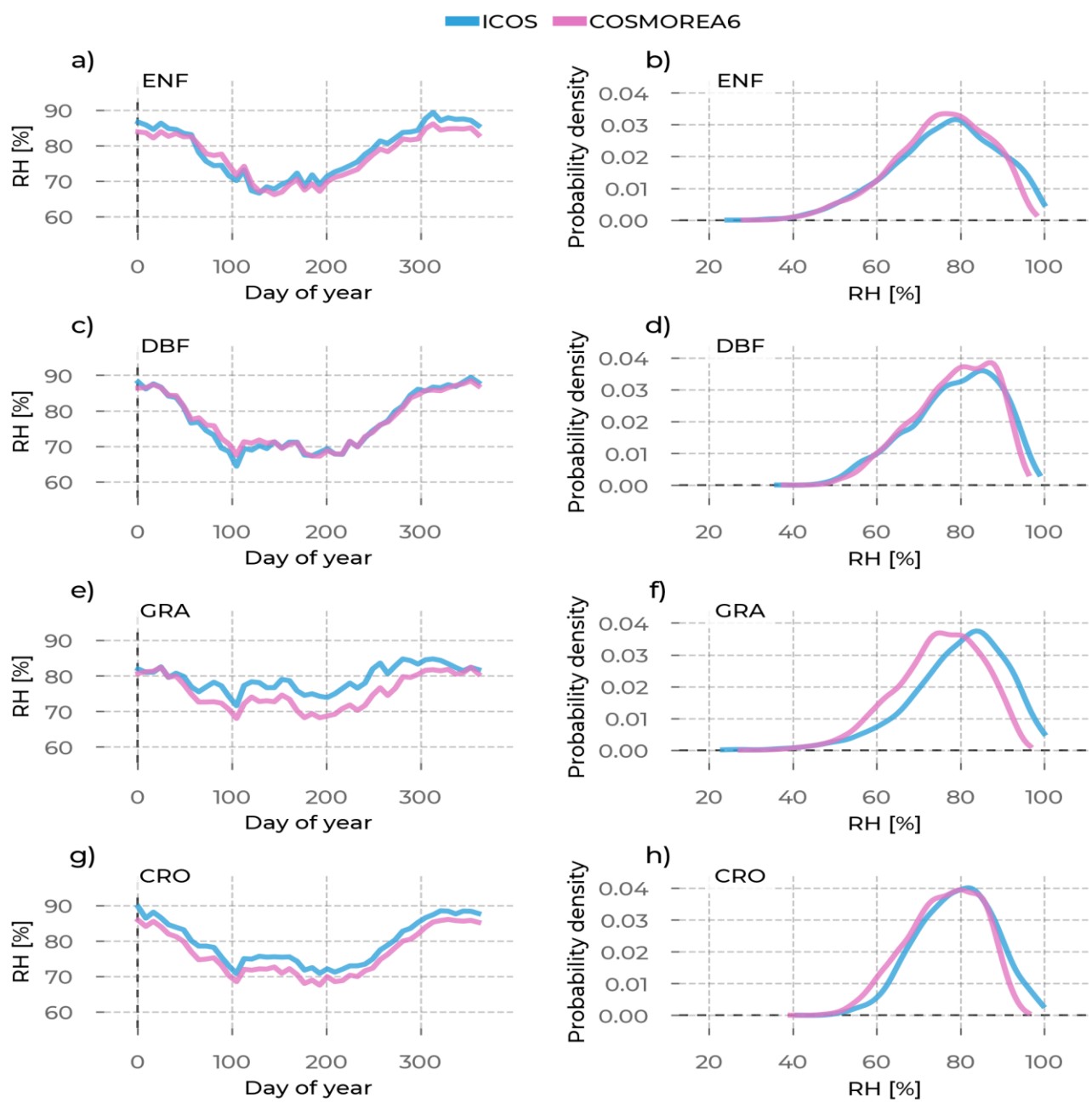

*Figure S13: In the left column are the yearly relative humidity (RH) evolutions averaged across stations belonging to one PFT (rows). We differentiate the data source by color (ICOS observations: blue, CLM5$_{grid}$: red, CLM5$_{PFT}$: yellow, GLASS: green, ERA5L: brown, GLEAM: purple). The probability density curves for all RH values from*

*stations belonging to the selected PFT are in the right column. Each row shows these plots for one PFT: Evergreen Needleleaf Forest (ENF), Deciduous Broadleaf Forest (DBF), Grasslands (GRA), and Croplands (CRO).*

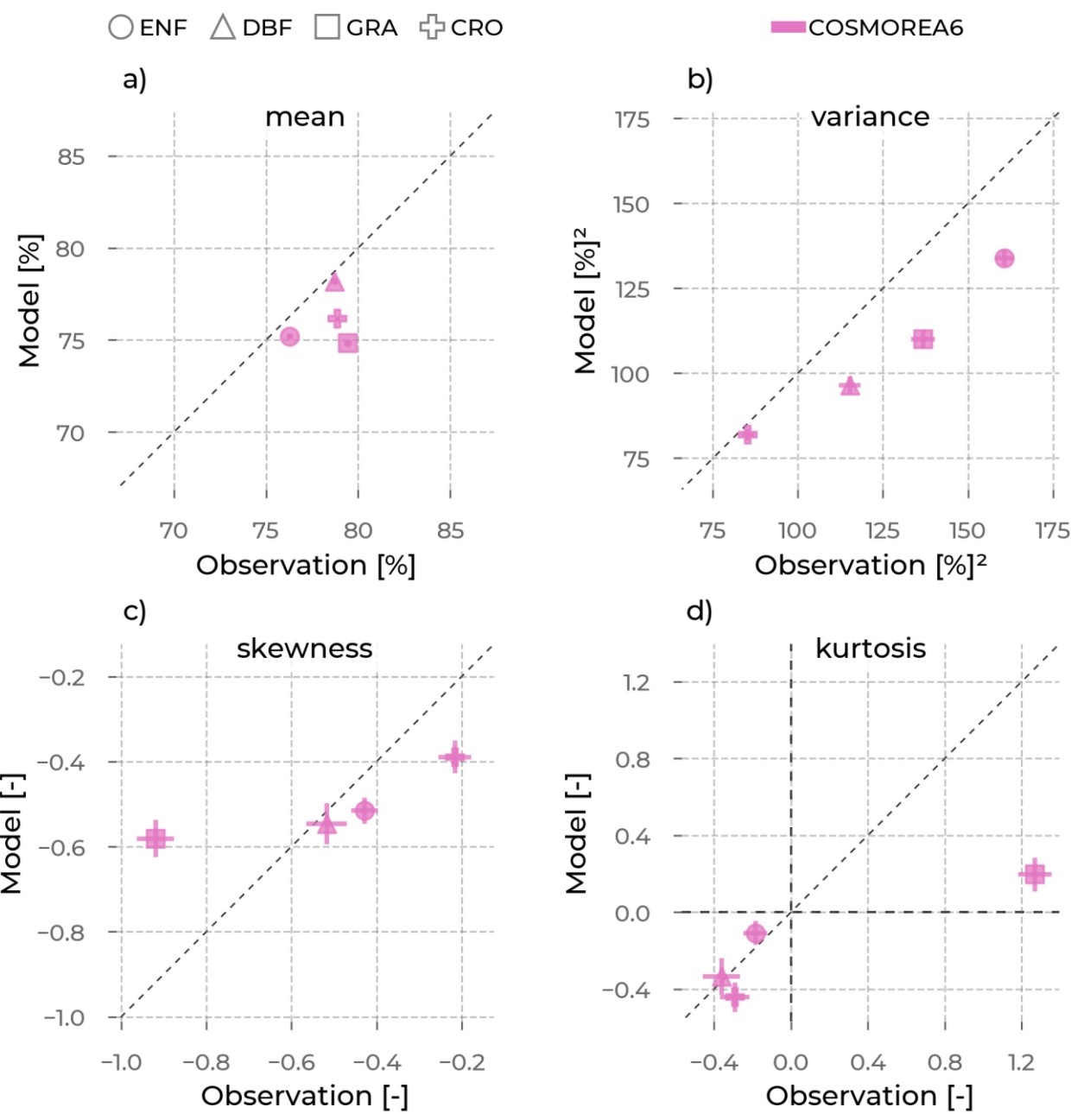

*Figure S14: The mean (a), variance (b), skewness (c), and excess kurtosis (d) of the RH distributions (visualized in Figure S13) from the models (color, y-axis), as opposed to the corresponding values from observations (x-axis) aggregated for each PFT (marker type): Evergreen Needleleaf Forest (ENF), Deciduous Broadleaf Forest (DBF),*

*Grasslands (GRA), Croplands (CRO). The error bars are the standard errors of the respective moment, depending on the sample size.*

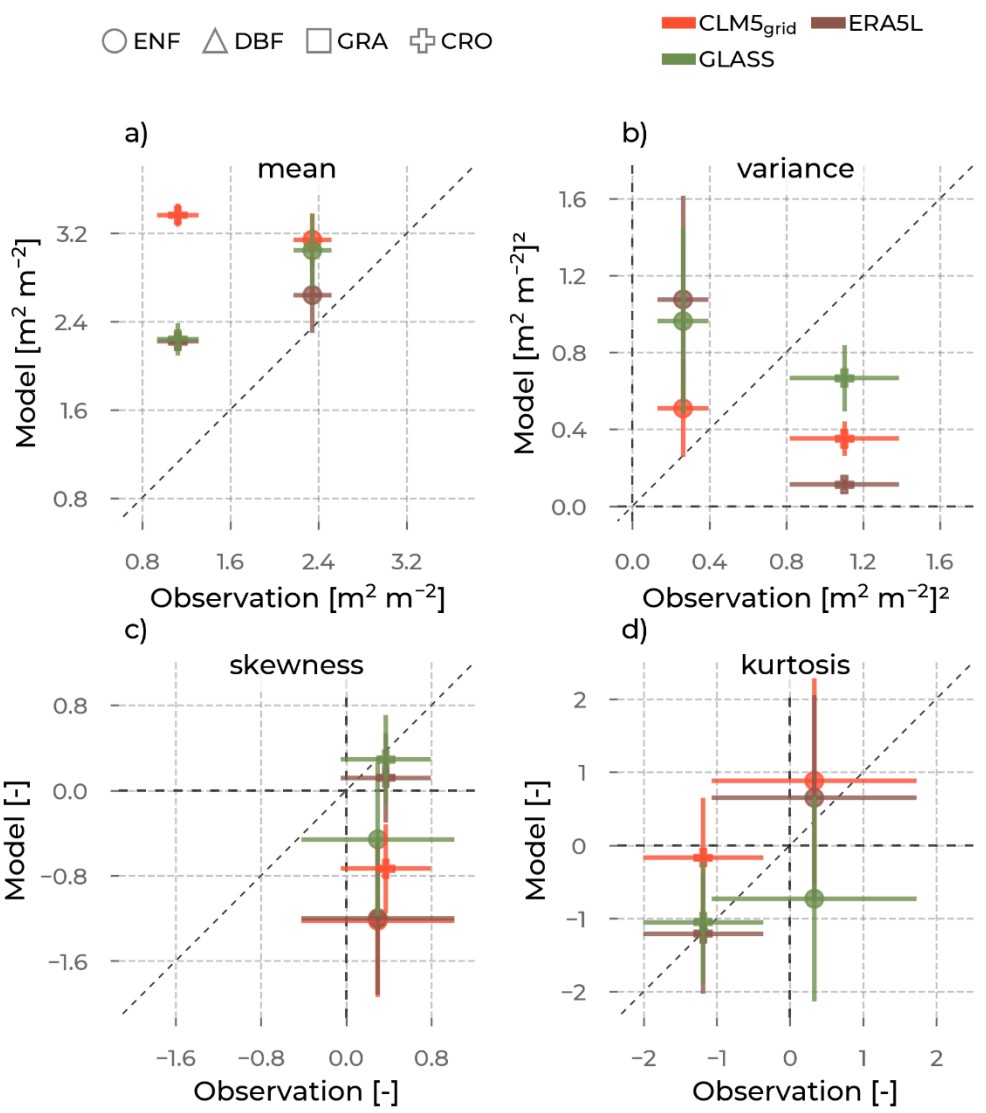


*Figure S15: The mean (a), variance (b), skewness (c), and excess kurtosis (d) of the leaf area index (LAI) distributions from the models (color, y-axis), as opposed to the corresponding values from observations (x-axis) aggregated for each plant functional type (marker type): Evergreen Needleleaf Forest (ENF), Deciduous Broadleaf Forest (DBF), Grasslands (GRA), Croplands (CRO). The error bars are the standard errors of the respective moment, depending on the sample size.*