# Peer review of "Systematic underestimation of type-specific ecosystem process variability in the Community Land Model v5 over Europe"

_EGUsphere, 2024_

## Author Comment (AC1)

**Author responses are embedded below the specific referee comment in green color.**

**Response to Anonymous Referee #1**

This is an interesting study that compares several gridded datasets of land energy, water and carbon fluxes with in situ observations over Europe. Simulations from the CLM5 land surface model are examined in more detail using two categories of simulations, one for each plant functional type and another aggregated to the grid cell level. It is found that CLM5 tends to underestimate the variability of water and carbon fluxes.

A list of possible reasons is given in the Discussion section, but nothing is said about the representation of leaf area index (LAI) by CLM5 and how a misrepresentation of the LAI seasonal cycle and interannual variability could affect model performance. In the introduction, the authors give a very broad definition of phenology without ever mentioning LAI. In reality, LAI is certainly more directly related to phenology than any other variable considered in this work. Moreover, LAI strongly controls land surface fluxes and the evaporative fraction. How is LAI represented in CLM5? Because LAI responds to environmental conditions, it can exhibit large interannual variability. Failure to represent this variability would reduce the ability of the model to represent land surface fluxes.

We are thankful for the referee's comments. We agree that the leaf area index (LAI) is an essential indicator of phenology and ecosystem function and co-varies with carbon uptake (GPP) and evapotranspiration (ET). We did not initially include LAI in our study for two main reasons:

1) In-situ LAI measurements are typically low-frequency, only providing a handful (at best) of data points per year. Therefore, the high-resolved GPP observations are more informative and robust for higher-order statistical analyses.

2) LAI is calculated from the leaf carbon (a partition from GPP), and it controls the upscaling of the carbon uptake from the individual plant to the canopy. Because of this tight relationship between GPP and LAI, we assumed that the analysis of only one variable is sufficiently informative.

However, as the referee rightly pointed out, exploring the phenology and variability of LAI could indeed be beneficial. Therefore, in our revised manuscript, we will comprehensively include the evaluation of CLM5 LAI and in-situ measurements.

Recommendation: major revisions.

Particular comments:
- L. 105-106 (warm winter 2020): Explain why it is called "warm winter".

This curated data-set supported research regarding the record warm winter 2020 in Europe. We will include this information in the revised manuscript.

- L. 143-144 (single soil column): This means that PFT-scale simulations are influenced by other PFTs. This weakens the rationale for PFT-scale simulations. It should be noticed than in other models, each PFT has its own soil column within a grid cell. This should be mentioned in the Discussion section.

We disagree with the referee. PFTs sharing a single soil column compete for water, potentially introducing water stress for less competitive PFTs. PFTs sharing a soil column better represent real-world conditions in heterogenous and water-scarce ecosystems — where competition for water does occur — than PFTs having separate soil columns. Thus, PFT-scale ecosystem processes should compare better to in-situ measurements, strengthening the rationale for PFT-scale evaluations. We will include a respective section in the discussion of the revised manuscript.

- L. 151: How does soil moisture affect stomatal conductance? Given the scope of this work, this should be clearly and completely explained.

A significant soil moisture deficit introduces water stress and down-regulates the stomatal conductance, carbon uptake and transpiration accordingly. We will include this aspect in detail in the Method section of a revised manuscript.

- L. 227 (warm winter 2020): For which time period are data available in the WARM-WINTER 2020 dataset? Only the 2019-2020 winter?

This varies on a site-basis. While the longest available time-series are from 1996, some sites provide only data for a single year. We only consider the simulation data where the observations are also available at the respective location. Please refer to the Supplementary Table S1 for the number of 8-daily data points available for each station. In a revised manuscript, we will include the available years for each site in this table.

- L. 263 (1995-2018): Clarify the link to the WARM-WINTER 2020 data set.

See the reply above. We will include the respective information also in this section in the revised manuscript.

- L. 300: Fig. 1c is not readable as many symbols overlap. This could be improved.

Indeed. We will make the scatter points more distinguishable in a revised manuscript.

- L. 326 (Table 2): units are missing ; what is the meaning of the symbol of column 2, lines 6 and 11?

Sorry for that inconvenience. The units will be included in the revised manuscript. We will also change the symbol indicating the average (Ø).

- L. 340 (Table 3): units are missing ; what is the meaning of the symbol of column 2, lines 6 and 11?

Please see the reply above. We will include the units here, too. The symbol is in some regions commonly used to indicate the average.

- L. 356 (Fig. 2): For a given PFT, is it a mean value across sites?

Correct. It is a mean value across the sites belonging to the respective PFT (and across the years available at those sites). We will include a clearer explanation in the caption in the revised manuscript.

**Author responses are embedded below the specific referee comment in green color.**

**Response to Anonymous Referee #2**

CLM5 ET and GPP are compared to ICOS sites in Europe, with RMSE and percent bias metrics. Model ET is often closer to the observations than remote sensing data, but model GPP is underestimated, particularly in deciduous forests.

Generally, the methods in this study seem robust, sources of uncertainty are carefully considered (Section 4), and the aims of the study are worthwhile. I certainly agree with the recommendations, especially with respect to optimizing PFT parameters and co-location of biodiversity and other data with the ICOS sites. The RMSE and bias metrics are well explained and appropriate.

Thanks, we appreciate the recognition of our study objectives and methods.

However, this study would be more accessible to a broader readership if metrics re the phenology and data distributions were better explained, with far less text given to describing the many details of the results and more to interpretation. Data-model comparisons (including for seasonal effects) should be quantified where possible, rather than just assessed by eye.

We agree that quantified indices of phenology would be more accessible than the descriptions in written text. In the revised manuscript, we will provide values for key aspects of the phenology curve (inflection points and modes), and shorten the written descriptions substantially.

The authors present the RMSE and bias results in tables 2 and 3, and also in the text (with some mistakes; e.g. in Section 3.2). Please consider displaying these results in a single diagram, such as a modified Taylor diagram.

Again, we agree and will provide a Taylor diagram in a revised manuscript.

Seasonal effects are shown in Figures 2 and 3, but they are then only discussed qualitatively in the text. There is no attempt to quantify differences (or variability) between model and observed peak ET or GPP timings. For example, model vs observed phase lag or estimated day of max ET or GPP (calculations clearly explained, with appropriate error bars) could be plotted and assessed. In any case, it would be helpful if the second "hypothesis" at the end of the introduction states how goodness-of-fit for phenology is to be quantified. Likewise, for the third "hypothesis", briefly state how the variability will be quantified.

Thanks. Similar to the quantification of modes and inflection points mentioned in the reply above, we will quantify the time difference between these key points in the simulations and the observations in a revised manuscript to support the hypothesis outline in the introduction.

The introduction states that the statistical distributions can help "contextualize" model drought responses, but there is no analysis or discussion about interpretation of the higher moments, responses to drought, or the apparent bimodality seen in Figures 4 and 6 in this article. Do the ICOS data suggest drought conditions at any time at any station? If not, are there any other climate-related

factors that could be discussed or quantified here? Please analyze/quantify/discuss drought, or another factor appearing in the ICOS data. In any case, it would be useful to know how drought (or other factor) affects skewness and kurtosis, and more broadly, what these moments will actually tell us or why we should care about them. For example, would we use the kurtosis to indicate changes in the frequency of extreme values, given kurtosis is a measure of the "heaviness" of the tails of a distribution?

We thank the referee for the critical remark. Investigating drought (e.g., soil moisture deficit) and a drought signal in the carbon uptake or evapotranspiration is highly complex due to the differences in the drought response functionality. For example, plant water stress might occur due to different magnitudes of water deficit in the soil, on different aggregation time scales of the water deficit, and with variable lead time (lag) when propagating from the soil to the vegetation. Given that, investigating whether and when a drought signal is present in the ecosystem processes is out of scope for this study. However, in a revised manuscript, we will shortly discuss how characteristics of the distribution moments, particularly the skewness and kurtosis, could evaluate the representation of observed extreme events in the model data.

Specific remarks

Abstract
The second sentence sounds odd. CLM5 quantifies fluxes and estimates the carbon and water budgets, potentially allowing for a better understanding of how climate change impacts ecosystems.

We will improve this sentence accordingly in the revised manuscript.

Line 30: reanalyses of what?

We will include information on the included reanalysis data in the abstract in the revised manuscript.

Figures and tables

Figure 1c (map of flux towers): Please show the extent of the CLM5 grid (1544x1592 gridcells), perhaps by using a different/lighter grey or white outside the grid area. Please state the number of stations shown in the caption.

The extent of the European CLM5 model corresponds with the complete shown map box. We will include the requested information in the image caption in the revised manuscript.

Figures 2 and 3 (seasonality curves): Define the ICOS, GLASS and model acronyms, and clarify that these curves are means and standard deviations of data covering X-X years during the period 1995 through 2018. It is difficult to see the ICOS curves in some of the panels; please bring them to the foreground in these plots to make them more obvious.

We will adapt the figure according to your suggestions in the revised manuscript.

Figures 4 and 6 (statistical distributions): It is rather difficult to see alignment of the main peaks in some of the panels, which is a point of discussion in the text.

We will quantify the alignment and shifts of modes of the distributions and improve the graph in the revised manuscript.

Figures 5 and 7 (moments): Clarify that these are moments from the distributions shown in figures 4 and 6. The kurtosis appears an "excess" kurtosis, given the normal distribution has kurtosis=3. Please clarify.

Thanks. We will include the requested information in the revised manuscript.

Tables 2 and 3: Ideally, the model and remote sensing acronyms should be defined in the caption; this may be more important than those of the PFTs which are defined in the previous figure and table. Please also explain PFT $\varnothing$ in the final rows for the RMSE and PBIAS sections.

Again, the requested information will be included in the revised manuscript. The $\varnothing$ stands for the average in some regions, but will be written out in the revision.

2.1.2 Setup of the European CLM5
Line 178: did you mean sub models for ice rather than "stub" models for ice?

No. We refer to a stub model as a method that represents a system compartment simplistically.

2.2.1 Station data

Line 229: Table S1 lists a single PFT for each ICOS station; please clarify here that this the dominant PFT as indicated in the last sentence of sec 2.3 ending line 267.

We will correct the information in the revised manuscript accordingly.

Line 234: Please state how many of the 73 stations were kept after wetlands, mixed forest, shrublands and indeterminate-land-cover stations were excluded; I assume 42 (the sum from Table 1).

Correct, 42 stations were retained for the analyses. We will include the information in the text of the revised manuscript.

3.2 General model performance
Line 333 bottom of page 16: The absolute value of PBIAS is smaller for CLM5PFT than for CLM5grid but the actual PBIAS is lower, being more negative. Please clarify; at least replace the word "lower" with "smaller".

We will correct the wording in the revised manuscript.

Line 336: In Table 2, ERA5L and GLASS RMSEs are largest for ENF and DBF as stated, but their RMSEs are lower than those of the CLM for GRA, rather than "similarly" as stated. Their PBIAS values are also much closer to zero than those of CLM5.

Thanks. We will correct the sentence in the revised manuscript.

Section 3.3.1 ET
Line 371: Please refer back to Table 2 when discussing the PBIAS and RMSE for ET. "Conversely" is better than "Oppositely"; the latter sounds weird.

We will improve the wording in the revision.

After this point, I stopped attempting to compare the text to the tables and figures. Rather than summarizing key points of a story, the text rambles on far too much about almost every detail of the results, and it is not always easy to see those details in the figures.

This will be improved in the revised version. We will provide quantified values that were described in the text before, shorten the text, and only summarize the most important points in the text.

---

## Author Response (AR1)

**Author responses are embedded below the specific referee comment in green color. Respective changes in the manuscript are highlighted in blue color below each comment.**

**Response to Anonymous Referee #1**

This is an interesting study that compares several gridded datasets of land energy, water and carbon fluxes with in situ observations over Europe. Simulations from the CLM5 land surface model are examined in more detail using two categories of simulations, one for each plant functional type and another aggregated to the grid cell level. It is found that CLM5 tends to underestimate the variability of water and carbon fluxes.

We appreciate the referee's insightful comments and revised the manuscript accordingly. Based on the referees' suggestions, we provide a thoroughly revised manuscript, and the main changes include the following:

1) Analysis of LAI to support the interpretation of the results for ET and GPP.
2) Taylor diagrams for a straightforward interpretation of the evaluation indices between the models.
3) A quantitative analysis of the shift between observed and simulated phenology curves.

Note that minor updates in the used libraries and code led to minor changes in the evaluation indices and distributions. However, this did not alter the interpretation of the results or any relative order between the observation and models in any of the analyses.

A list of possible reasons is given in the Discussion section, but nothing is said about the representation of leaf area index (LAI) by CLM5 and how a misrepresentation of the LAI seasonal cycle and interannual variability could affect model performance. In the introduction, the authors give a very broad definition of phenology without ever mentioning LAI. In reality, LAI is certainly more directly related to phenology than any other variable considered in this work. Moreover, LAI strongly controls land surface fluxes and the evaporative fraction. How is LAI represented in CLM5? Because LAI responds to environmental conditions, it can exhibit large interannual variability. Failure to represent this variability would reduce the ability of the model to represent land surface fluxes.

We acknowledge that the leaf area index (LAI) is a crucial indicator of phenology and ecosystem function, and it co-varies with carbon uptake (GPP) and evapotranspiration (ET). We did not initially include LAI in our study for three main reasons:

1) In-situ LAI measurements are typically low-frequency, providing only a handful (at best) of data points per year. Therefore, we focused on the high-resolved GPP data, which we believe are more informative and robust for higher-order statistical analyses.
2) In CLM5-BGC, LAI is calculated from the leaf carbon (a partition from GPP) and controls the upscaling of the carbon uptake from the individual plant to the canopy. Because of this tight relationship between GPP and LAI, we assumed that the analysis of only one variable is sufficiently informative.
3) LAI measurements are done over given vegetation stands for each site, which might not exactly coincide with the eddy covariance tower footprint, where the measured GPP and ET originate.

The mismatch in vegetation composition, soil characteristics, and microclimate limits the ability to infer relationships and explain co-variation between these variables.

However, as the referee rightly pointed out, exploring LAI's phenology and variability could be beneficial. Therefore, the supplement of our revised manuscript includes the evaluation of LAI from CLM5-BGC, GLASS remote sensing (MODIS-based), and ERA5-Land with in-situ measurements. ICOS data sets include LAI measurements. However, as pointed out, these are not as systematic, frequent, and harmonized as data from the eddy-covariance towers. Moreover, ICOS measurements started only more recently from 2017, limiting the possible evaluation period to two years (2017 and 2018). Lastly, only needleleaf evergreen forest (ENF) and cropland (CRO) sites have available data for this period. In short, the evaluation was performed for ENF and CRO in 2017 and 2018. Figure S15 shows the comparison of the distribution moments for each of these plant functional types with their respective uncertainty:

[Figure]

Figure S15: The mean (a), variance (b), skewness (c), and kurtosis (d) of the leaf area index (LAI) distributions from the models (color, y-axis), as opposed to the corresponding values from observations (x-axis) aggregated for each plant functional type (marker type): Evergreen Needleleaf Forest (ENF), Deciduous Broadleaf Forest (DBF), Grasslands (GRA), Croplands (CRO). The error bars are the standard errors of the respective moment, depending on the sample size.

We further discuss the implications of this analysis in lines 672 – 694:

"Given the hydraulic role of vegetation leaves in controlling transpiration, there is a tight relationship between ET, GPP, and LAI. In CLM5-BGC, the assimilated carbon by GPP gets further partitioned to respiration and the carbon storage in the plant organs, i.e., leaves, roots, and stems. Furthermore, the leaf carbon then controls the development and state of the vegetation leaves and, thus, the LAI. On the other hand, LAI controls GPP by determining the upscaling factors from leaf photosynthesis to the canopy, thereby driving canopy conductance. Unfortunately, no large-scale LAI in-situ measurements and no CLM5PFT simulated LAI are available, and comparisons between CLM5grid LAI and reanalysis or remote sensing LAI suffer from known biases in the latter and yield no further context on our evaluation based on ground truth information. We adhered to an LAI evaluation of CLM5 with sparse but systematic ICOS measurements, ERA5L reanalysis, and GLASS based on MODIS (Supplementary Figure S15). Notably, the ICOS LAI measurements are only available for two years of the study period (2017 to 2018) and are limited to ENF and CRO sites. Additionally, LAI measurements' expensive and time-intensive nature restricts the time resolution to a few measurement points per year. As a result, the data points for comparison are few, and the uncertainties are larger (noticeable larger error bars in Supplementary Figure S15). Another caveat is the potential mismatch of the land surface representation between the EC tower footprint (ET and GPP measurements) and the area covered by the LAI measurement campaigns. However, some key findings from this analysis are still robust. For example, all models overestimate LAI at ENF and CRO sites (Supplementary Figure 14 a), contrasting the results of GPP and ET. The variance in ENF sites is much more significant in GLASS and ERA5L than in CLM5grid, which is closest to the observations. The higher-order moments are more uncertain because of the small number of data points. The contrasting results, especially between the LAI and GPP PFT-level averages, suggest that processes and parameters connecting the assimilated carbon to the leaf area, depending on the environmental conditions, must be revisited. However, we make an even stronger case for systematic, long-term, and high-resolution LAI in-situ measurements (for example, using drones (Bates et al. 2021)), which would support a more robust and diverse evaluation of the simulations of this essential variable."

Recommendation: major revisions.

Particular comments:
- L. 105-106 (warm winter 2020): Explain why it is called "warm winter".

This curated data set supported research regarding Europe's record-warm winter of 2020. We included this information in the revised manuscript in the lines 104 – 106:

"The Integrated Carbon Observation System (ICOS) provides the WARM-WINTER-2020 data (Warm Winter 2020 Team and ICOS Ecosystem Thematic Centre, 2022), which includes Eddy Covariance measurements over a dense network of over 70 sites in Europe, and was curated named after and to support research on the effect of the warm winter of the year 2020 on the terrestrial carbon fluxes."

- L. 143-144 (single soil column): This means that PFT-scale simulations are influenced by other PFTs. This weakens the rationale for PFT-scale simulations. It should be noticed than in other models, each PFT has its own soil column within a grid cell. This should be mentioned in the Discussion section.

We agree with the referee that PFTs sharing a single soil column compete for water, potentially introducing water stress for less competitive PFTs. PFTs sharing a soil column better represent real-world conditions in heterogeneous and water-scarce ecosystems — where competition for water does occur — than PFTs having separate soil columns. Thus, PFT-scale ecosystem processes should compare better to in-situ measurements, strengthening the rationale for PFT-scale evaluations that potentially include competition for resources. We included respective information in the revised manuscript in the lines 144 – 148:

"The vegetation is grouped into PFTs (Lawrence and Chase, 2007), which are distinguished through leaf habit (evergreen or deciduous), morphology (needle and broad leaves, grass, and shrubs), and the bioclimate of the grid cell location (boreal, temperate, and tropical). While competition for soil moisture includes interactions among different PFTs, this is closer to natural conditions than separated soil columns and encourages evaluations on the PFT scale."

- L. 151: How does soil moisture affect stomatal conductance? Given the scope of this work, this should be clearly and completely explained.

A significant soil moisture deficit introduces water stress and down-regulates the stomatal conductance, carbon uptake, and transpiration. We included this aspect in the Method section of the revised manuscript, in lines 174 – 176:

"If T cannot meet the atmospheric water demand because of a soil moisture shortage, CLM5-BGC introduces water stress and attenuates $g_s$ based on that transpiration deficit factor. Through decreased $g_s$, water stress also regulates the photosynthesis, A."

- L. 227 (warm winter 2020): For which time period are data available in the WARM-WINTER 2020 dataset? Only the 2019-2020 winter?

This varies on a site and variable basis. While the most extended available time series began in 1996, some sites only provide one year of data that overlaps our study period (1995 – 2018). We only consider the simulation data where the observations are available at the respective location. Please refer to Supplementary Table S1 for the 8-daily data points available for each station. In the revised manuscript, that table also includes the years within the study period where data is available for each site. See the excerpt (first six entries) below:

| ID | country | PFT | lat | lon | years | lat (cell) | lon (cell) | N (ET) | N (GPP) |
|---|---|---|---|---|---|---|---|---|---|
| BE-Bra | Belgium | ENF | 51.31 | 4.52 | 1996 – 2018 | 51.29 | 4.51 | 503 | 670 |
| BE-Dor | Belgium | GRA | 50.31 | 4.97 | 2011 – 2018 | 50.31 | 4.96 | 0 | 270 |
| BE-Lcr | Belgium | DBF | 51.11 | 3.85 | - | 51.10 | 3.85 | 0 | 0 |
| BE-Lon | Belgium | CRO | 50.55 | 4.75 | 2004 – 2018 | 50.57 | 4.76 | 440 | 476 |

| CH-Cha | Switzerland | GRA | 47.21 | 8.41 | 2005 – 2018 | 47.21 | 8.43 | 386 | 459 |
| CH-Dav | Switzerland | ENF | 46.82 | 9.86 | 1997 – 2018 | 46.80 | 9.84 | 546 | 866 |

- L. 263 (1995-2018): Clarify the link to the WARM-WINTER 2020 data set.

We have added further information on the post-processing and the link to the ICOS WARM-WINTER reference data in the revised manuscript in lines 274 – 283:

"First, the remote sensing and reanalysis data are bilinearly remapped to the 3 km European CORDEX grid and interpolated to 8-daily means for 1995 - 2018. The ICOS observation time series are interpolated to 8-daily means for each station whose data availability overlaps with our study period. Then, we extracted the CLM5grid, GLASS, ERA5L, and GLEAM data from the grid cell closest to the location of each selected ICOS station. Further, we select the time series in CLM5PFT that coincides with that grid cell and the station's PFT. Importantly, we focus only on the four predominant PFTs represented in the ICOS network: Evergreen Needleleaf Forest (ENF), Deciduous Broadleaf Forest (DBF), Grasslands (GRA), and Croplands (CRO), as outlined in Table 1. Finally, the periods where station data is absent or of bad quality (determined by the corresponding measurement or gap-filling quality flag in the ICOS data) are discarded from the simulations to ensure we are comparing the same set of conditions."

- L. 300: Fig. 1c is not readable as many symbols overlap. This could be improved.

Indeed. We improved the visibility of the marker points in the Figure 1 map by choosing hollow marker types. The individual site locations are now more visible in the figure in the revised manuscript, Figure 1:

[Figure]

*Figure 1: The share of represented plant functional types (color: Evergreen Needleleaf Forest (ENF), Deciduous Broadleaf Forest (DBF), Grasslands (GRA), and Croplands (CRO)) in a) the ICOS station network used in subsequent analyses and b) the corresponding grid cells in our European CLM5 setup. In c) is a map showing the locations of the ICOS stations, with the marker type indicating their PFT and the color of the marker indicating their hydro-climate (adapted from Jafari et al. (2018)) based on the mean annual precipitation from the COSMO-Reanalysis 6.*

- L. 326 (Table 2): units are missing ; what is the meaning of the symbol of column 2, lines 6 and 11?

Sorry for the inconvenience. The units are included now in the revised manuscript. In some regions, the symbol "∅" is commonly used to indicate the mean value. We changed it to "mean," these tables are now available in the Supplementary Material as Tables S4 and S5. In the main text, we now provide modified Taylor diagrams to assess the model performances more easily:

[revised manuscript text omitted]

- L. 340 (Table 3): units are missing ; what is the meaning of the symbol of column 2, lines 6 and 11?

Sorry for the inconvenience. The units are included now in the revised manuscript. In some regions, the symbol "∅" is commonly used to indicate the mean value. We changed it to "mean". This updated Table S5 is in the Supplementary material of the revised manuscript:

[revised manuscript text omitted]

**Author responses are embedded below the specific referee comment in green color.**

**Response to Anonymous Referee #2**

CLM5 ET and GPP are compared to ICOS sites in Europe, with RMSE and percent bias metrics. Model ET is often closer to the observations than remote sensing data, but model GPP is underestimated, particularly in deciduous forests.

Generally, the methods in this study seem robust, sources of uncertainty are carefully considered (Section 4), and the aims of the study are worthwhile. I certainly agree with the recommendations, especially with respect to optimizing PFT parameters and co-location of biodiversity and other data with the ICOS sites. The RMSE and bias metrics are well explained and appropriate.

We appreciate the referee's insightful comments and revised the manuscript accordingly. Based on the referees' suggestions, we provide a thoroughly revised manuscript, and the main changes include the following:

1) Analysis of LAI to support the interpretation of the results for ET and GPP.
2) Taylor diagrams for a straightforward interpretation of the evaluation indices between the models.
3) A quantitative analysis of the shift between observed and simulated phenology curves.

Note that minor updates in the used libraries and code led to minor changes in the evaluation indices and distributions. However, this did not alter the interpretation of the results or any relative order between the observation and models in any of the analyses.

However, this study would be more accessible to a broader readership if metrics re the phenology and data distributions were better explained, with far less text given to describing the many details of the results and more to interpretation. Data-model comparisons (including for seasonal effects) should be quantified where possible, rather than just assessed by eye.

We agree that quantified phenology indices would be more accessible than the descriptions in written text. In the revised manuscript, we provide specific quantifications of the shift of critical phenology events for each plant functional type (the summer peaks and the beginning and the end of the growing season indicated by the infliction point) between the observations and the models and consider their uncertainties. We also substantially shortened the written descriptions. The shifts in phenological events are depicted in Figures 5 and 7:

[revised manuscript text omitted]

The authors present the RMSE and bias results in tables 2 and 3, and also in the text (with some mistakes; e.g. in Section 3.2). Please consider displaying these results in a single diagram, such as a modified Taylor diagram.

We corrected the mistakes, moved Tables 2 and 3 to the Supplementary Material (now Supplementary Tables S4 and S5) and provided the modified Taylor diagrams in the revised manuscript as Figures 2 and 3:

[Figure]

*Figure 2: Modified Taylor diagrams with observations from the Integrated Carbon Observation System (ICOS) of evapotranspiration as reference (black markers) and showing model performances between the years 1996 – 2018 (years varying by station; see Supplementary Table S1. Data sources by color: Community Land Model v5 (CLM5), CLM5$_{grid}$: red, CLM5$_{PFT}$: yellow, Global Land Surface Satellite (GLASS): green, European Center for Medium-Range Weather Forecasts Reanalysis 5 – Land (ERA5L): brown, Global Land Evaporation Amsterdam Model (GLEAM): purple). Each diagram shows these plots for one plant functional type. Upper left: Evergreen Needleleaf Forest (ENF, circles), upper right: Deciduous Broadleaf Forest (DBF, triangles), lower left: Grasslands (GRA, squares), and lower right: Croplands (CRO, crosses). The azimuth angle indicates the Pearson correlation with the ICOS data, the radial distance is the standard deviation, and the semicircles centered at the reference standard deviation show the root mean square error (RMSE). The size of each marker indicates the percent bias (PBIAS).*

[Figure]

*Figure 3: Modified Taylor diagrams with observations from the Integrated Carbon Observation System (ICOS) of gross primary production as reference (black markers) and showing model performances between the years 1996 – 2018 (years varying by station; see Supplementary Table S1. Data sources by color: Community Land Model v5 (CLM5), CLM5$_{grid}$: red, CLM5$_{PFT}$: yellow, Global Land Surface Satellite (GLASS): green). Each diagram shows these plots for one plant functional type. Upper left: Evergreen Needleleaf Forest (ENF, circles), upper right: Deciduous Broadleaf Forest (DBF, triangles), lower left: Grasslands (GRA, squares), and lower right: Croplands (CRO, crosses). The azimuth angle indicates the Pearson correlation with the ICOS data, the radial distance is the standard deviation, and the semicircles centered at the reference standard deviation show the root mean square error (RMSE). The size of each marker indicates the percent bias (PBIAS).*

Seasonal effects are shown in Figures 2 and 3, but they are then only discussed qualitatively in the text. There is no attempt to quantify differences (or variability) between model and observed peak ET or GPP timings. For example, model vs observed phase lag or estimated day of max ET or GPP (calculations clearly explained, with appropriate error bars) could be plotted and assessed. In any case, it would be helpful if the second "hypothesis" at the end of the introduction states how

goodness-of-fit for phenology is to be quantified. Likewise, for the third "hypothesis", briefly state how the variability will be quantified.

As shown above, we quantified the timings of crucial phenology events for each plant functional type and their standard deviation across respective sites to quantify their variability. We then compare the timings of these events in Figures 5 and 7 of the revised manuscript:

[Figure]

Figure 5: Mean shifts in ET phenological events (the start of the growing season, peak, and the end of the growing season) between the Integrated Carbon Observation System (ICOS) observations (solid black line) and the models (by color: Community Land Model v5 (CLM5), CLM5$_{grid}$: red, CLM5$_{PFT}$: yellow, Global Land Surface Satellite (GLASS): green, European Center for Medium-Range Weather Forecasts Reanalysis 5 Land (ERA5L): brown, Global Land Evaporation Amsterdam Model (GLEAM): purple), among sites belonging to one plant functional type: Evergreen Needleleaf Forest (ENF), Deciduous Broadleaf Forest (DBF), Grasslands (GRA), and Croplands (CRO). On the x-axis is the day of the year of the event. Error bars in grey correspond to the standard deviation of the day of the event in the models across sites of one plant functional type, and the error bars in black correspond to the standard deviation across the respective observations.

[Figure]

*Figure 7: Mean shifts in GPP phenological events (the start of the growing season, peak, and the end of the growing season) between the Integrated Carbon Observation System (ICOS) observations (solid black line) and the models (by color: Community Land Model v5 (CLM5), CLM5$_{grid}$: red, CLM5$_{PFT}$: yellow, Global Land Surface Satellite (GLASS): green), among sites belonging to one plant functional type: Evergreen Needleleaf Forest (ENF), Deciduous Broadleaf Forest (DBF), Grasslands (GRA), and Croplands (CRO). On the x-axis is the day of the year of the event. Error bars in grey correspond to the standard deviation of the day of the event in the models across sites of one plant functional type, and the error bars in black correspond to the standard deviation of the respective observations.*

The updated hypotheses in the revised manuscript in lines 133 – 142 now include information on how they would be tested quantitatively:

1. There is a lower systematic bias, and the simulation is closer to the observations by the PFT scale than the grid-scale CLM5 outputs, remote sensing, and reanalysis data.

2. The remotely sensed and modeled data approximate critical events in the phenologies of ET and GPP within the standard deviation of the ICOS measurements for sites of one PFT. However, this ability varies between PFTs.

3. The remotely sensed and modeled ET and GPP data distributions show a lower range among the moments within the PFT groups than the ICOS measurements.

The introduction states that the statistical distributions can help "contextualize" model drought responses, but there is no analysis or discussion about interpretation of the higher moments, responses to drought, or the apparent bimodality seen in Figures 4 and 6 in this article. Do the ICOS data suggest drought conditions at any time at any station? If not, are there any other climate-related factors that could be discussed or quantified here? Please analyze/quantify/discuss drought, or another factor appearing in the ICOS data. In any case, it would be useful to know how drought (or other factors) affects skewness and kurtosis, and more broadly, what these moments will actually tell us or why we should care about them. For example, would we use the kurtosis to indicate changes in the frequency of extreme values, given kurtosis is a measure of the "heaviness" of the tails of a distribution?

We thank the referee for the critical remark. Investigating drought (e.g., soil moisture deficit) and a drought signal in the carbon uptake or evapotranspiration is highly complex due to the differences in the drought response functionality. For example, plant water stress might occur due to different magnitudes of water deficit in the soil, on different aggregation time scales of the water deficit, and with variable lead time (lag) when propagating from the soil to the vegetation. Given that, investigating whether and when a drought signal is present in the ecosystem processes is out of scope for this study. Nevertheless, in the revised manuscript, we briefly discuss how characteristics of the distribution moments, particularly the skewness and kurtosis, could evaluate the representation of observed extreme events in the model data in lines 726 – 751:

"4.5 Distribution moments and droughts
Investigating the influence of drought on the analyses, or generally the ability of the models to simulate drought and the vegetation response, is complex due to the differences in drought response function-ality. For instance, plant water stress might occur due to different magnitudes of water deficit in the soil, on different aggregation time scales, and with a variable lag to the water deficit. A future study will investigate the PFT-scale drought responses from the model and how the drought propagates through the eco-hydrological sphere and compare it to observations. However, drought frequency, duration, and severity affect the shapes of the distribution of the precipitation and, eventually, the ecosystem processes. Thus, we briefly discuss possible insights into their drought responses.
Importantly, the skewness and excess kurtosis moments, which inform about the characteristics of the distribution tails (relativity between the tails and the general tailedness, respectively) of precipitation (Guo, 2022), and vegetation states and function (Kanavi et al., 2020; Liu et al., 2022; Cooley et al., 2022), are influenced by dry conditions, depending on their frequency, duration and severity. We found a low variability in the skewness and excess kurtosis of the precipitation used to force our CLM5 simulations (Figure S10 c and d), specifically a significantly lower skewness and excess kurtosis at ENF and DBF sites. A lower positive skewness than the observations means that the distribution is less skewed towards lower values, and a lower positive excess kurtosis than the observations indicates generally larger tails. A possible interpretation of these differences in the distribution moments is that the atmospheric forcings show more frequent, longer, and more severe extreme precipitation events, while the ICOS measurements are more concentrated around their mean. While the propagation of these extreme events could be complex and non-linear, we generally found the same results (lower skewness and smaller absolute excess kurtosis) for the simulated distributions of ET and GPP for almost all models and PFTs (Figures 9 c and d and 11 c and d), suggesting a more direct relationship. However, because of the possible non-linearity and the influence of other factors, the detailed relationship between these findings and the ability of CLM5 to simulate ecosystem drought responses must be examined in future studies. In any case, the missing

accuracy in representing higher distribution moments in the atmospheric forcings and in land surface models must be considered in studies using these to investigate drought."

Specific remarks

Abstract
The second sentence sounds odd. CLM5 quantifies fluxes and estimates the carbon and water budgets, potentially allowing for a better understanding of how climate change impacts ecosystems.

We improved this sentence accordingly in the revised manuscript in lines 19 – 22:

"Land surface models such as the Community Land Model v5 (CLM5) quantify these fluxes, estimate the state of carbon budgets and water resources, and contribute to a better understanding of climate change's impact on ecosystems."

Line 30: reanalyses of what?

We referred to reanalyzed data of GPP and ET (e.g., ET from ERA5-Land and GLEAM). We adapted the sentence for a better understanding in the revised manuscript in lines 29 – 31:

"Furthermore, the simulated ET and GPP distribution moments across PFTs in $CLM5_{grid}$ and $CLM5_{PFT}$ and their reanalyzed and remotely sensed counterparts indicate an underestimated spatiotemporal variability compared to the observations across Europe."

Figures and tables

Figure 1c (map of flux towers): Please show the extent of the CLM5 grid (1544x1592 gridcells), perhaps by using a different/lighter grey or white outside the grid area. Please state the number of stations shown in the caption.

The extent of the European CLM5 model corresponds to the complete map box shown. We included the requested information in the last sentence of the image caption in the revised manuscript:

*Figure 1: The share of represented plant functional types (color: Evergreen Needleleaf Forest (ENF), Deciduous Broadleaf Forest (DBF), Grasslands (GRA), and Croplands (CRO)) in a) the ICOS station network used in subsequent analyses and b) the corresponding grid cells in our European CLM5 setup. In c) is a map showing the locations of the ICOS stations, with the marker type indicating their PFT and the color of the marker indicating their hydro-climate (adapted from Jafari et al. (2018)) based on the mean annual precipitation from the COSMO-Reanalysis 6. Our 3 km European CLM5 simulation domain corresponds to the entire map box in c).*

Figures 2 and 3 (seasonality curves): Define the ICOS, GLASS and model acronyms, and clarify that these curves are means and standard deviations of data covering X-X years during the period 1995 through 2018. It is difficult to see the ICOS curves in some of the panels; please bring them to the foreground in these plots to make them more obvious.

Thanks. We adapted ICOS data in the figures to bold black color to make them more visible in all figures throughout the manuscript. We also stated that the years included in the analysis vary by station, and refer to Supplementary Table S1. As an example, this is Figure 4 in the revised manuscript (previously Figure 2):

[Figure]

*Figure 4: In the left column are the yearly evapotranspiration (ET) evolutions averaged across stations belonging to one plant functional type (rows: Evergreen Needleleaf Forest (ENF), Deciduous Broadleaf Forest (DBF), Grasslands (GRA), and Croplands (CRO)) and across the years (available years vary per station, see Supplementary Table S1). We differentiate the data source by color (Integrated Carbon Observation System (ICOS) observations: black, Community Land Model v5 (CLM5), CLM5$_{grid}$: red, CLM5$_{PFT}$: yellow, Global Land Surface Satellite (GLASS): green, European Center for Medium-Range Weather Forecasting Reanalysis 5 – Land (ERA5L): brown, Global Land Evaporation Amsterdam Model (GLEAM): purple). The corresponding standard deviations across the sites and across the years are plotted in the right column to measure the spread around this mean.*

Figures 4 and 6 (statistical distributions): It is rather difficult to see alignment of the main peaks in some of the panels, which is a point of discussion in the text.

We improved the visibility of the statistical distributions by making the ICOS observations line bold and black, and the colored model lines thinner. Please see Figures 8 and 10 in the revised manuscript:

[revised manuscript text omitted]

Figures 5 and 7 (moments): Clarify that these are moments from the distributions shown in figures 4 and 6. The kurtosis appears an "excess" kurtosis, given the normal distribution has kurtosis=3. Please clarify.

Thanks. In the captions of each Figure showing distribution moments, we clarified that they refer to the distributions shown in the corresponding distribution Figures. Further, we included the information that, indeed, we are presenting the "excess" kurtosis throughout the revised manuscript.

Tables 2 and 3: Ideally, the model and remote sensing acronyms should be defined in the caption; this may be more important than those of the PFTs which are defined in the previous figure and table. Please also explain PFT ∅ in the final rows for the RMSE and PBIAS sections.

The ∅ stands for the average in some regions. Because we included the modified Taylor plots as Figure 2 and Figure 3 now in the main text of the revised manuscript, we moved the Tables with the evaluation

indices (previously Tables 2 and 3) to the Supplementary material. We added the data source full names in the Table captions and wrote out "mean" in the corresponding row indices in the revised manuscript, Supplementary Tables S4 and S5:

*Table S4: The evapotranspiration (ET) root mean square error (RMSE) indicates the general model approximations and the percent bias (PBIAS), demonstrating systematic bias of the models (Community Land Model v5 (CLM5) on grid-scale (CLM5$_{grid}$), CLM5 on PFT scale (CLM5$_{PFT}$), from the European Center of Medium-Range Weather Forecasts Renalysis 5 - Land (ERA5-Land), the Global Land Surface Satellite (GLASS), and the Global Land Evaporation Amsterdam Model (GLEAM)) to the observations. Each value corresponds to a group of stations representing the same plant functional type (PFT; Evergreen Needleleaf Forest (ENF), Deciduous Broadleaf Forest (DBF), Grasslands (GRA), and Croplands (CRO)). The amount of data points (N) for each PFT is also indicated.*

|  | PFT | N | CLM5$_{grid}$ | CLM5$_{PFT}$ | ERA5L | GLASS | GLEAM |
|---|---|---|---|---|---|---|---|
| RMSE [mm day$^{-1}$] | ENF | 5038 | 0.71 | 0.72 | 0.84 | 0.83 | 0.67 |
| | DBF | 1663 | 0.56 | 0.62 | 0.73 | 0.70 | 0.56 |
| | GRA | 2859 | 0.65 | 0.85 | 0.60 | 0.57 | 0.59 |
| | CRO | 3690 | 0.72 | 1.00 | 0.88 | 0.86 | 0.63 |
| | mean | 3285 | 0.66 | 0.80 | 0.76 | 0.74 | 0.61 |
| PBIAS [%] | ENF | 5038 | -20.57 | -15.42 | 21.86 | 13.32 | 15.43 |
| | DBF | 1663 | -9.90 | -0.54 | 44.55 | 29.74 | 16.24 |
| | GRA | 2859 | -18.62 | -13.94 | 3.14 | 2.63 | 2.41 |
| | CRO | 3690 | -3.24 | 11.20 | 44.99 | 27.30 | 7.58 |
| | mean | 3285 | -13.08 | -18.70 | 28.64 | 18.25 | 10.42 |

*Table S5: The gross primary production (GPP) root mean square error (RMSE) indicates the general model approximation and the percent bias (PBIAS), demonstrating systematic bias of the models (Community Land Model v5 (CLM5) on grid-scale (CLM5$_{grid}$), CLM5 on PFT scale (CLM5$_{PFT}$), from the European Center of Medium-Range Weather Forecasts Renalysis 5 Land (ERA5-Land), the Global Land Surface Satellite (GLASS), and the Global Land Evaporation Amsterdam Model (GLEAM)) to the observations. Each value corresponds to a group of stations representing the same plant functional type (PFT: Evergreen Needleleaf Forest (ENF), Deciduous Broadleaf Forest (DBF), Grasslands (GRA), and Croplands (CRO)). The amount of data points (N) for each PFT is also indicated.*

|  | PFT | N | CLM5$_{grid}$ | CLM5$_{PFT}$ | GLASS |
|---|---|---|---|---|---|
| **RMSE [g C day$^{-1}$]** | ENF | 5976 | 2.25 | 2.44 | 1.75 |
|  | DBF | 2473 | 3.71 | 3.35 | 2.81 |
|  | GRA | 2838 | 3.14 | 3.01 | 2.63 |
|  | CRO | 3607 | 3.85 | 4.21 | 3.55 |
|  | mean | 3723.5 | 3.24 | 3.25 | 2.69 |
| **PBIAS [%]** | ENF | 5976 | -26.00 | -7.7 | -14.53 |
|  | DBF | 2473 | -38.88 | -43.76 | -24.51 |
|  | GRA | 2838 | -30.73 | -25.5 | -21.34 |
|  | CRO | 3607 | -14.99 | -1.48 | -6.29 |
|  | mean | 3723.5 | -27.65 | -19.61 | -16.67 |

2.1.2 Setup of the European CLM5

Line 178: did you mean sub models for ice rather than "stub" models for ice?

No. We refer to a stub model as a method that represents a system compartment simplistically.

2.2.1 Station data

Line 229: Table S1 lists a single PFT for each ICOS station; please clarify here that this the dominant PFT as indicated in the last sentence of sec 2.3 ending line 267.

The last sentence in sec. 2.3 instead referred to the predominance of these four PFTs (ENF, DBF, GRA, and CRO) in the station network rather than a single station's footprint. We tried making this point clearer in sec. 2.3 of the revised manuscript in lines 274 – 281:

"Further, we select the time series in CLM5$_{PFT}$ that coincides with that grid cell and the station's predominant PFT. Importantly, we focus only on the four predominant PFTs represented in the entire ICOS station network: Evergreen Needleleaf Forest (ENF), Deciduous Broadleaf Forest (DBF), Grasslands (GRA), and Croplands (CRO), as outlined in Table 1."

We included a clarification in the revised manuscript in lines 233 – 235:

"Note that the land cover type indicated by the ICOS site metadata and represented in the measurements refers to the predominant PFT in the footprint of the eddy covariance station."

And in the caption of Table S1:

"Table S1: A list of ICOS stations, their land cover, coordinates, years of data availability for our study period (1995 – 2018), the coordinates of the corresponding grid cell of the 3 km European Coordinated Regional Climate Downscaling Experiment (CORDEX) grid used in our simulations, and the number of 8-daily data points available for the analyses for evapotranspiration (ET) and gross primary production (GPP). Note that stations that do not belong to the plant functional types (PFT) of evergreen needleleaf forest (ENF), deciduous broadleaf forest (DBF), grasslands (GRA), and croplands (CRO) were omitted, and some included sites did not have data corresponding with the study period, thus having a count of 0 data points. See Section 2.2.1. The indicated PFT is the predominant PFT in the footprint of the ICOS eddy covariance towers. Stations, where the land cover was not directly indicated in the metadata sites were also left out in our analyses."

Line 234: Please state how many of the 73 stations were kept after wetlands, mixed forest, shrublands and indeterminate-land-cover stations were excluded; I assume 42 (the sum from Table 1).

Correct, 42 stations were retained for the analyses. We included this information in the revised manuscript lines 238 – 240:

The analyses also excluded stations whose land cover type was not included in metadata sites (e.g., DEIMS-SDR https://deims.org), leaving a total of 42 stations for our analyses."

3.2 General model performance
Line 333 bottom of page 16: The absolute value of PBIAS is smaller for CLM5PFT than for CLM5grid but the actual PBIAS is lower, being more negative. Please clarify; at least replace the word "lower" with "smaller".

We corrected the wording in the revised manuscript in the lines 365 – 366:

"In Table 2, we list the performance indices for ET and the number of 8-daily time-steps across the corresponding stations that went into their calculation. CLM5$_{PFT}$ has a higher RMSE and a smaller absolute PBIAS than CLM5$_{grid}$ for ET across PFTs, except in CRO."

Line 336: In Table 2, ERA5L and GLASS RMSEs are largest for ENF and DBF as stated, but their RMSEs are lower than those of the CLM for GRA, rather than "similarly" as stated. Their PBIAS values are also much closer to zero than those of CLM5.

Thanks. We corrected the sentence in the revised manuscript in lines 368 – 369:

"ERA5L and GLASS show more significant deviations from the ICOS ET observations at ENF and DBF than CLM5PFT and CLMgrid and have smaller RMSE values at GRA."

Section 3.3.1 ET

Line 371: Please refer back to Table 2 when discussing the PBIAS and RMSE for ET. "Conversely" is better than "Oppositely"; the latter sounds weird.

We generally improved the wording of the Results section and considered this suggestion. However, this sentence does not appear in the revised manuscript, as we now refer more to the modified Taylor diagrams than the Tables.

After this point, I stopped attempting to compare the text to the tables and figures. Rather than summarizing key points of a story, the text rambles on far too much about almost every detail of the results, and it is not always easy to see those details in the figures.

The Results section was drastically improved in readability by removing descriptive text, improving the wording, and concisely pointing out the main findings of each figure and analysis. Please check the much shorter and rewritten Results section in the revised manuscript.